# Understanding LLM Behaviors via Compression: Data Generation, Knowledge Acquisition and Scaling Laws

**Zhixuan Pan**[1,*]    **Shaowen Wang**[1,*]    **Pengfei Liao**[2]    **Jian Li**[1,†]

[1]Institute for Interdisciplinary Information Sciences, Tsinghua University
[2]School of Computer Science and Engineering, Beihang University
panzx24@mails.tsinghua.edu.cn, wangsw23@mails.tsinghua.edu.cn
liaopf22@buaa.edu.cn, lapordge@gmail.com

## Abstract

Large Language Models (LLMs) have demonstrated remarkable capabilities across numerous tasks, yet principled explanations for their underlying mechanisms and several phenomena, such as scaling laws, hallucinations, and related behaviors, remain elusive. In this work, we revisit the classical relationship between compression and prediction, grounded in Kolmogorov complexity and Shannon information theory, to provide deeper insights into LLM behaviors. By leveraging the Kolmogorov Structure Function and interpreting LLM compression as a two-part coding process, we offer a detailed view of how LLMs acquire and store information across increasing model and data scales – from pervasive syntactic patterns to progressively rarer knowledge elements. Motivated by this theoretical perspective and natural assumptions inspired by Heap's and Zipf's laws, we introduce a simplified yet representative hierarchical data-generation framework called the Syntax-Knowledge model. Under the Bayesian setting, we show that prediction and compression within this model naturally lead to diverse learning and scaling behaviors of LLMs. In particular, our theoretical analysis offers intuitive and principled explanations for both data and model scaling laws, the dynamics of knowledge acquisition during training and fine-tuning, factual knowledge hallucinations in LLMs. The experimental results validate our theoretical predictions.

## 1   Introduction

Large Language Models (LLMs) have emerged as one of the most influential breakthroughs in modern artificial intelligence, achieving impressive performance across a multitude of tasks, ranging from fluent text generation, translations, and summarization to answering factual queries, and even performing complex reasoning. Despite these remarkable achievements, a theoretical understanding of what enables LLMs to generalize so effectively remains limited. Traditional learning theory frameworks have not yet fully explained why certain scaling laws hold, why in-context learning emerges, or when and why hallucinations arise in the output of these models.

One promising lens for gaining deeper insight into LLM behavior is the intrinsic connection between prediction and compression. Kolmogorov complexity and Shannon information theory have long established that optimal prediction of a data sequence is intimately tied to the most efficient compression of that sequence (see e.g., [13, 47, 59, 34]). From this perspective, a predictive model, particularly a large language model (LLM), can be viewed as a practical approximation of the Kolmogorov compressor (see e.g., [69, 16, 48]) of training data. See Appendix C and Appendix B for more details.

---

[*]Equal contribution.
[†]Corresponding author.

Following this line of thought, we build upon the *Kolmogorov structure function* [47, 61] and interpret LLM training as constructing a *two-part (data-to-model) code* for the training data. The first part (the model compressor part) corresponds to the LLM itself, which adjusts its parameters to learn patterns and structural regularities for more efficient compression. The second part (the data part) is the compressed code of the data, generated by using the LLM as the compressor. [3] As already indicated by prior work on the structure function (see e.g., [71, 61, 46]), a model of low complexity captures only the most prominent regularities in the data, whereas allowing model of higher complexity captures more nuanced structures. In the context of compressing language corpus under capacity constraints, the most efficient LLM-based compressor should initially focuses on compressing frequently recurring regularities such as syntactic patterns, then integrates relatively common knowledge, and eventually handles increasingly rare knowledge elements. Any "residual" (such as factual knowledge exceeding the model's capacity or unpredictable noise) must be left out of the model and explicitly encoded in the second part. See Appendix C for formal definition of Kolmogorov structure function and Figure 5(a) for a schematic illustration.

Motivated by the insight from the study of Kolmogorov Structure Function and natural assumptions inspired by *Heap's law* [30] and *Zipf's law* [76, 77] (see Appendix A for more details), we propose the Syntax-Knowledge model, a (simplified) hierarchical data-generative framework that decomposes the generation process into two components (see Figure 5(b)). The first component, a *parametric Syntax Model*, captures the syntactic structures of language, allowing for random syntactic variations. The second component, a *Knowledge Model*, encodes relevant world knowledge and is modeled using the nonparametric *Pitman-Yor Chinese Restaurant Process*. This choice reflects the growing nature of human knowledge and captures the fact that certain pieces of information occur disproportionately more frequently than others in the data. By examining how efficiently data generated by this model can be compressed (specifically, by minimizing perplexity, or equivalently coding redundancy), we clarify the learning behaviors of LLMs: highlighting that syntax model is learned first at a faster rate, and the (factual) knowledge elements are acquired according to the order of their frequency. Furthermore, we theoretically demonstrate how model performance scales with both data size and model capacity, thus providing intuitive explanations for the scaling laws observed in real-world LLM training. Our contributions are summarized as follows:

1. **Kolmogorov Structure Function Perspective.** By viewing LLM training as a two-part coding process in the Kolmogorov Structure Function, we present a principled framework for viewing LLMs as compressors that distinguish between structural regularities of varying frequencies and residual randomness. See Appendix C.

2. **A Non-Parametric Hierarchical Data Generation Model.** In Section 3, we introduce the **Syntax-Knowledge model**, a hierarchical generative framework that separates language syntax (captured by a parametric model) from (factual) knowledge (represented by a non-parametric *Pitman-Yor Chinese Restaurant Process*). This design naturally accommodates the growing nature and power-law distributions of knowledge elements, motivated by Heap's law [30] and Zipf's laws [77], reflecting their disproportionate frequency in real-world data.

3. **Data Scaling Law.** Within a Bayesian framework, we show that perplexity minimization applied to data generated by our Syntax-Knowledge model naturally leads to data scaling laws observed in real world LLMs. In particular, the Bayesian redundancy is equal to the mutual information between the prior and the data, and we can bound the mutual information by $\widetilde{O}\left(C_{\mathrm{knw}}/N^{1-\alpha} + C_{\mathrm{syn}}/N\right)$, where $N$ is the size of the training data, $C_{\mathrm{knw}}$ and $C_{\mathrm{syn}}$ are constants depending on the knowledge model and syntax model respectively, and $\alpha$ is the discount parameter of the Pitman-Yor Chinese Restaurant Process employed in our knowledge model. See Section 4.2.

4. **Model Scaling Law.** In Section 4.3, we extend our theoretical models to account for model scaling laws, under a slightly different set of assumptions. As a consequence, we offer a more fine-grained understanding of model scaling by decomposing the test loss according to the frequency of knowledge elements. This allows us to accurately predict which knowledge elements LLMs can acquire under capacity constraints during training, and which ones they are more likely to hallucinate(even though the model may have encountered them many

---

[3]For details on using an LLM or any auto-regressive predictor to compress data via Arithmetic coding, see [62, 13, 16] or Appendix B.1.

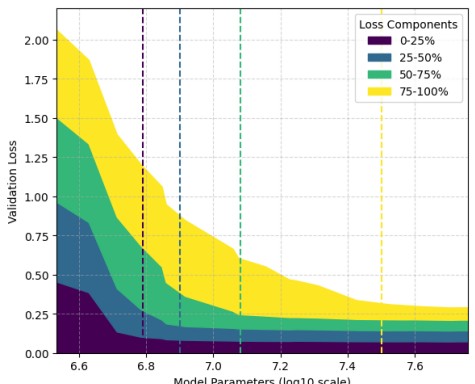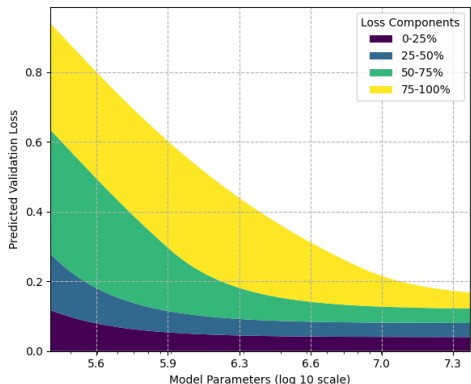

Figure 1: Decomposition of validation loss on knowledge tokens by frequency class, as model size increases. (a) Empirical results on a power-law-distributed dataset: knowledge tokens are grouped into four frequency classes (from most to least frequent) and colored accordingly. We observe the following trend: smaller models capture only the most frequent knowledge (the loss of the most frequent class decreases the first), while larger models gradually acquire less frequent knowledge. Each vertical dashed line marks the model size beyond which further loss reduction for a given frequency class becomes negligible, indicating the irreducible part of the loss. (b) Theoretical prediction of the same loss decomposition (the optimal solution of the constrained optimization problem (5) in Section 4.3) with irreducible loss part (i.e., the $H(P_\phi)$ term in (3)), which reproduces this frequency-dependent acquisition order and plateauing behavior.

times) as demonstrated in Figure 2. See Figure 1 for both our theoretical prediction and the corresponding experimental results on model scaling behaviors. [4]

## 2    Background

In this section, we review the necessary background on information theory, coding, perplexity minimization, and scaling laws in LLMs. Let $\mathbb{V}$ denote the set of all possible tokens. We denote the training corpus as $X_{1:N} := X_1, X_2, \ldots, X_N$, where each $X_i = x_1^{(i)} x_2^{(i)} \cdots x_{l_i}^{(i)}$ is a sentence represented as a sequence of tokens, with each token $x_j^{(i)} \in \mathbb{V}$.

**Perplexity Minimization in LLMs.**    In LLMs, the cross-entropy loss (or log-loss) serves as the metric for measuring how well the model predicts a given sequence of tokens. A model $\boldsymbol{M}$, which induces a predictive distribution $P_{\boldsymbol{M}}$, estimates the conditional probability of each token $x_t$ given the preceding context $x_{1:t-1}$ for each sentence. We assume that all sentences in the training corpus $X_{1:N}$ are independently drawn from the source distribution $P_{\phi_{\text{data}}}$. Given the training corpus $X_{1:N}$, the training objective is to minimize the *empirical averaged cross-entropy loss*:

$$\mathcal{L}(\boldsymbol{M}) = -\frac{1}{N} \log P_{\boldsymbol{M}}(X_{1:N}) = -\frac{1}{N} \sum_{i=1}^{N} \log P_{\boldsymbol{M}}(X_i) = \mathbb{E}_{X \sim \widehat{P}_\phi}[-\log P_{\boldsymbol{M}}(X)]$$

$$= H(\widehat{P}_\phi \| P_{\boldsymbol{M}}) = -\frac{1}{N} \sum_{i=1}^{N} \sum_{t} \log P_{\boldsymbol{M}}(x_t^{(i)} \mid x_{1:t-1}^{(i)}), \tag{1}$$

where $\widehat{P}_\phi$ is the empirical measure of the underlying source distribution $P_{\phi_{\text{data}}}$, and $H(P\|Q) = \mathbb{E}_{X \sim P}[-\log Q(X)]$ is the standard cross-entropy loss. The *perplexity* (PPL) is then defined as $\text{PPL} := \exp(\mathcal{L}(\boldsymbol{M}))$, capturing the effective "number of choices" the model has at each token.

---

[4]We follow the experimental setting of [2, 3], but use a power law distribution of individuals. The experimental details can be found in Appendix G.

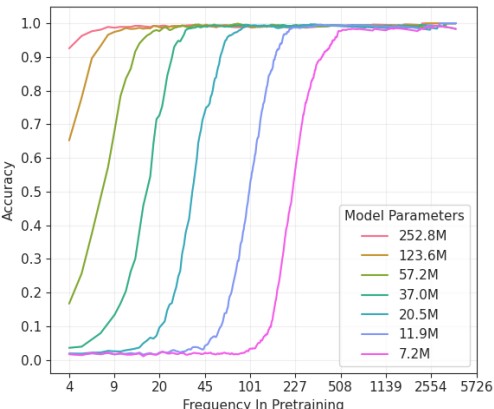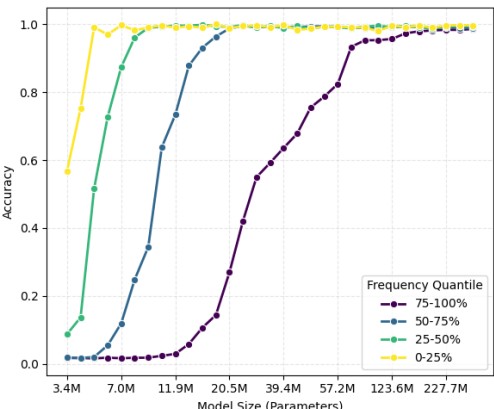

Figure 2: (a) Accuracy of sufficiently trained models with different sizes across varying input frequencies. When the frequency falls below a model-specific threshold, small models inevitably hallucinate and fail to learn the corresponding facts. (b) Accuracy of different frequency classes (split into four quantiles) under varying model sizes. As model size increases, the model progressively learns the more frequent data first, while infrequent data becomes learnable only at larger scales.

**Lossless Compression and Redundancy.** The goal of lossless compression is to encode a sequence of tokens $X = x_{1:n}$ (sampled from the source distribution $P_\phi$ parametrized by $\phi$) into a binary sequence $y_{1:m}$ of minimal *expected* length, such that $x_{1:n}$ can be recovered perfectly from $y_{1:m}$. We use a binary source code $c : \mathbb{V}^* \to \{0, 1\}^*$, which encodes each possible sequence $x_{1:n}$ into a binary codeword $c(x_{1:n})$ of length $\ell(c(x_{1:n}))$ bits. The objective is to minimize the expected code length

$$L(Q_c) := \mathbb{E}_{X \sim P_\phi}[\ell(c(X))] = \mathbb{E}_{X \sim P_\phi}[-\log Q_c(X)] = H(P_\phi \| Q_c). \tag{2}$$

where $Q_c(x) \propto 2^{-\ell(c(x))}$ is the *predictive probability corresponding to the code $c$*. According to Shannon's source coding theorem [64], the average length of any lossless code is bounded below by the entropy $H(P_\phi)$, and one can design an encoding scheme whose code length is $H(P_\phi) + O(1)$ if $P_\phi$ is known. However, when the source distribution $P_\phi$ is unknown, universal coding becomes necessary (see Appendix B.2 for more details). In this setting, the extra code length beyond $H(P_\phi)$, referred to as the *redundancy* of the code $c$, is defined as follows:

$$\mathsf{Red}(Q_c, P_\phi) = L(Q_c) - H(P_\phi) = H(P_\phi \| Q_c) - H(P_\phi) = D_{\mathsf{KL}}(P_\phi \| Q_c). \tag{3}$$

where $D_{\mathsf{KL}}(P_\phi \| Q_c) = \mathbb{E}_{x \sim P_\phi}[\log(P_\phi(x)/Q_c(x))]$ is the Kullback–Leibler (KL) divergence between source distribution $P_\phi$ and predicted distribution $Q_c$. Comparing equations (1) and (2) reveals that obtaining a predictive model that minimizes perplexity (or cross-entropy loss) is essentially equivalent to finding a code with minimal expected length. (See Appendix B.1 for further details on the *equivalence between prediction and compression*.) Moreover, one can see from (3) that $H(P_\phi)$ constitutes the irreducible part of the loss, corresponding to the minimum achievable code length under Shannon's source coding theorem. Therefore, the goal of minimizing the expected code length is essentially equivalent to minimizing the redundancy.

**Scaling Laws in LLMs.** The performance of LLMs, particularly the cross-entropy loss $\mathcal{L}$, has been observed to improve predictably with increases in model size, dataset size, and computational resources[42, 33]. These empirical relationships are known as *scaling laws*. A common formulation for the loss as a function of dataset size $D$ (e.g., number of tokens) and model size $M$ (e.g., number of parameters) $\mathcal{L}(D, M) \approx (D/D_0)^{-\alpha} + (M/M0)^{-\beta} + \varepsilon$. In these formulations, $D_0$ and $M_0$ represent characteristic scales for the dataset and model size respectively. The exponents $\alpha > 0$ and $\beta > 0$ determine how quickly the loss decreases as the dataset size and model size increase. The term $\varepsilon$ represents the *irreducible loss*, which is the minimum achievable loss that cannot be reduced by further scaling, potentially due to factors like the inherent entropy of the data.

## 3 A Hierarchical Data Generation Model

In this section, we propose a hierarchical data generation model, called the *syntax-knowledge* model. In this model, where each sentence in the training corpus is generated by a *syntax encoder* that encode

a (factual) knowledge element, sampled from the *knowledge model*. The syntax model (encoder) is parameterized by $\phi_{\text{syn}}$, the knowledge model is denoted as $\phi_{\text{knw}}$, and the entire data model is denoted as $\phi_{\text{data}} = \{\phi_{\text{syn}}, \phi_{\text{knw}}\}$.

In our model, the syntax model $\phi_{\text{syn}}$ (e.g., a probabilistic CFG, an English/code grammar, or even a template-based format) does not grow with the size of the dataset and can be modeled using a finite-dimensional parameterized model $\phi_{\text{syn}}$. On the other hand, the knowledge model employs a *non-parametric* stochastic process to account for two empirically observed phenomena: 1) the unbounded growth of factual information as datasets grow (mirroring Heap's Law in lexical growth patterns [30]), and 2) the long-tailed frequency distribution of factual occurrences, analogous to Zipf's law in natural language [76, 77].

Motivated by the above idea, we leverage the nonparametric *Pitman–Yor process (PYP)* [56] for modeling the knowledge model $\phi_{\text{knw}}$. A PYP is characterized by two real parameters, the *discount parameter* $0 \leq \alpha < 1$ and the *concentration parameter* $\beta > -\alpha$, and a base probability measure $\pi_{\text{knw}}$. We denote it as $\text{PYP}(\alpha, \beta, \pi_{\text{knw}})$. A sample from the Pitman–Yor process $\text{PYP}(\alpha, \beta, \pi_{\text{knw}})$ is a random probability measure $\phi_{\text{knw}} = \sum_{i=1}^{\infty} p_i \delta_{\phi_i}$, which is a discrete distribution with countably infinite atoms, where $p = (p_1, p_2, \ldots)$ are the weights generated by the Pitman–Yor Chinese Restaurant Process (PYCRP) (described below); each atom $\phi_i \sim \pi_{\text{knw}}$ is the $i$-th cluster parameter independently drawn from the base measure $\pi_{\text{knw}}$.

The weights $p = (p_1, p_2, \ldots)$ are generated by the Pitman–Yor Chinese Restaurant Process (PYCRP), denote as $p = (p_1, p_2, \ldots) \sim \text{PYCRP}(\alpha, \beta)$. PYCRP adopts a preferential attachment mechanism that naturally captures both the *sublinear scaling* of new factual discoveries and the *power-law distributed frequencies* of knowledge pieces (see Lemma H.2 in Appendix H.1). PYCRP works as follows: Imagining a restaurant where customers arrive sequentially, each choosing either to join an existing lively table or start their own. The first customer sits at a new table. Consider the $n$-th customer who just come to the restaurant. Suppose $N_k$ is the number of customers already at table $k$, and $K$ is the current number of occupied tables. The $n$-th customer either joins an existing table $k$ or starts a new table with the following probabilities:     The $n$-th customer joins an existing table $k$ with probability $(N_k - \alpha)/(n - 1 + \beta)$, or starts a new table with probability $(\beta + \alpha K)/(n - 1 + \beta)$. The weight $p = (p_1, p_2, \ldots) = \lim_{n \to \infty} (N_1/n, N_2/n, \cdots)$ are defined as the relative sizes of the tables as the number of customers $n \to \infty$. More details of PYMM can be found in Appendix D.1.

**Hierarchical Data Model:** In the Pitman–Yor Mixture Model, the Pitman–Yor Process serves as a prior defined over the set of mixture distributions. Recall that a sample from the Pitman–Yor Process is a random probability measure of the form $\phi_{\text{knw}} = \sum_{i=1}^{\infty} p_i \delta_{\phi_i}$, where $\{p_i\}$ are the mixture weights and $\{\phi_i\}$ are the atoms. This can be viewed as a mixture distribution in which the $i$-th cluster is chosen with probability $p_i$, and the corresponding model is parameterized by $\phi_i$, referred to as the $i$-th knowledge cluster (table).

The parameters $\phi_i$ are drawn from a base measure $\pi_{\text{knw}}$, which acts as a prior over the cluster parameters. We assume $\pi_{\text{knw}}$ is supported on a bounded parameter space $\mathbf{\Phi}_{\text{knw}} = \{\phi \in \mathbb{R}^{d_{\text{knw}}} : \|\phi\|_2 \leq 1\}$, ensuring that each $\phi_i \in \mathbf{\Phi}_{\text{knw}}$ for all $i \in \mathbb{N}^+$.

Now, it is ready to describe the Syntax-Knowledge model, which generates a sentence $X$ according to the following hierarchical framework:

1. We first independently sample the latent parameters of the knowledge and syntax models:
   $\phi_{\text{knw}} \sim \text{PYP}(\alpha, \beta, \pi_{\text{knw}}), \quad \phi_{\text{syn}} = \{\phi_{\text{syn}}^{(1)}, \phi_{\text{syn}}^{(2)}, \ldots, \phi_{\text{syn}}^{(n_s)}\}, \quad \phi_{\text{syn}}^{(i)} \overset{\text{i.i.d.}}{\sim} \pi_{\text{syn}}(\phi_{\text{syn}})$, where $\pi_{\text{syn}}$ is the prior distribution of $\phi_{\text{syn}}^{(i)}$ and supported on a bounded parameter space $\mathbf{\Phi}_{\text{syn}} = \{\phi \in \mathbb{R}^{d_{\text{syn}}} : \|\phi\|_2 \leq 1\}$. ensuring that each $\phi_{\text{syn}}^{(i)} \in \mathbf{\Phi}_{\text{syn}}$ for all $1 \leq i \leq n_s$. The value $n_s$ denotes the number of distinct syntax parameter vectors, indicating that different types of knowledge should be expressed through different syntactic patterns.

2. We generate the sentence from both $\phi_{\text{syn}}$ and $\phi_{\text{knw}}$. In fact, we first sample an (abstract) knowledge element $\boldsymbol{\kappa}$ (corresponding to a customer) from a knowledge cluster (corresponding to a table) in $\phi_{\text{knw}}$, use $\boldsymbol{\kappa}$ to determine which syntax $\phi_{\text{syn}}^{(i)}$ (e.g., which template or format) to use, and then use the syntax encoder to generate the corresponding sentence. See Appendix D for details and Figure 7 for the data model schematic.

# 4 Explaining Scaling Laws

## 4.1 A Bayesian Sequential Prediction Framework

In this section, we explain LLM scaling laws by adopting the Bayesian sequential prediction framework (also called online Bayesian coding game). This Bayesian setting has been studied extensively in information theory (especially related to universal coding), statistics and machine learning literature (see e.g., [11, 12] and more recent expositions [17, 37]). More details can be found in Appendix B.2.2.

Given the data-generating distribution $P_{\phi_{\text{data}}}$, the redundancy of the model $M$ (recall the definition from (3)) with respect to samples $X_{1:N}$ i.i.d. drawn from $P_{\phi_{\text{data}}}$ is:

$$\text{Red}_N(M, \phi_{\text{data}}) = \mathbb{E}_{X_{1:N} \sim P_{\phi_{\text{data}}}} \left[ -\log P_M(X_{1:N}) \right] - \mathbb{E}_{X_{1:N} \sim P_{\phi_{\text{data}}}} \left[ -\log P_{\phi_{\text{data}}}(X_{1:N}) \right] \quad (4)$$

$$= D_{\text{KL}}(P_{\phi_{\text{data}}}^N \,\|\, P_M),$$

where the first term represents the cumulative cross-entropy of the model $M$, and the second term corresponds to the irreducible entropy of the ground-truth data distribution $\phi_{\text{data}}$.

In Bayesian generative models, the data-generating parameter $\phi_{\text{data}}$ is treated as a random variable drawn from a prior distribution $\pi$. Let $\mathbf{\Phi}_{\text{data}}$ denote the parameter space. The *Bayesian redundancy* of a model $M$ is defined as the expected redundancy under the prior $\pi$:

$$\text{Red}_N(M, \mathbf{\Phi}_{\text{data}}) := \int_{\mathbf{\Phi}_{\text{data}}} \pi(\phi_{\text{data}}) \, \text{Red}_N(M, \phi_{\text{data}}) \, d\phi_{\text{data}}.$$

According to Lemma B.3 in Appendix B.2.2, the optimal Bayesian redundancy is given by:

$$\inf_M \int_{\mathbf{\Phi}_{\text{data}}} \pi(\phi_{\text{data}}) \text{Red}_N(M, \phi_{\text{data}}) d\phi_{\text{data}} = \int_{\mathbf{\Phi}_{\text{data}}} \pi(\phi_{\text{data}}) D_{\text{KL}}(P_{\phi_{\text{data}}}^N \| Q_\pi) d\phi_{\text{data}} = \mathbb{I}(X_{1:N}; \phi_{\text{data}}),$$

where $Q_\pi = \int P_{\phi_{\text{data}}}^N \pi(\phi_{\text{data}}) \, d\phi_{\text{data}}$ denotes the marginal distribution over $X_{1:N}$ obtained by first sampling $\phi_{\text{data}} \sim \pi$, and then drawing $X_{1:N} \stackrel{\text{i.i.d.}}{\sim} P_{\phi_{\text{data}}}$ conditioned on $\phi_{\text{data}}$.

## 4.2 Data Scaling Law (under the Bayesian framework)

Under the Bayesian sequential prediction framework, we derive an upper bound on the optimal Bayesian redundancy (which is equal to the mutual information $\mathbb{I}(X_{1:N}; \phi_{\text{data}})$ by Lemma B.3) of our hierarchical data model $\phi_{\text{data}}$.

We make the following natural assumption of the knowledge model: if the parameters of two knowledge clusters are close, the KL divergence between two induced distributions is small.

**Assumption 4.1.** The probability families $\mathbf{\Phi}_{\text{syn}}$ and $\mathbf{\Phi}_{\text{knw}}$ satisfy the following: there exist positive constants $L_{\text{knw}}$ and $L_{\text{syn}}$ such that for all $\phi_{\text{knw}}^{(1)}, \phi_{\text{knw}}^{(2)} \in \mathbf{\Phi}_{\text{knw}}$ and $\phi_{\text{syn}}^{(1)}, \phi_{\text{syn}}^{(2)} \in \mathbf{\Phi}_{\text{syn}}$, we have:

$$D_{\text{KL}}\left(P_{\phi_{\text{knw}}^{(1)}} \| P_{\phi_{\text{knw}}^{(2)}}\right) \le L_{\text{knw}} \|\phi_{\text{knw}}^{(1)} - \phi_{\text{knw}}^{(2)}\|^2, \quad D_{\text{KL}}\left(P_{\phi_{\text{syn}}^{(1)}} \| P_{\phi_{\text{syn}}^{(2)}}\right) \le L_{\text{syn}} \|\phi_{\text{syn}}^{(1)} - \phi_{\text{syn}}^{(2)}\|^2.$$

The constants $L_{\text{knw}}$ and $L_{\text{syn}}$ may depend on the concrete form of the parametrization of $P_{\phi_{\text{knw}}}$ and $P_{\phi_{\text{syn}}}$, and are typically related to the Fisher information (See Lemma H.6 and Remark H.7). However, the particular form of the parametrization is not important for our later development. The constants $L_{\text{knw}}$ and $L_{\text{syn}}$ appear in logarithmic order in the upcoming Theorem 4.2. For the sake of clarity, we omit logarithmic terms in the statement of Theorem 4.2. Under the Bayesian setting and the above assumption, we can derive the following upper bound of the optimal Bayesian redundancy.

**Theorem 4.2.** *Under the Bayesian sequential prediction framework and Assumption 4.1, the averaged optimal Bayesian redundancy (per sentence) of the hierarchical data model $\phi_{data}$ satisfies:*

$$\inf_M \frac{1}{N} \text{Red}_N(M, \mathbf{\Phi}_{data}) = \frac{1}{N} \mathbb{I}(X_{1:N}; \phi_{data}) = \widetilde{O}\left(\frac{d_{knw}}{N^{1-\alpha}} + \frac{n_s d_{syn}}{N}\right).$$

*where $d_{knw}$ and $d_{syn}$ are the parameter dimensions of the knowledge and syntax clusters, respectively, and $n_s$ is the number of distinct syntax clusters.*

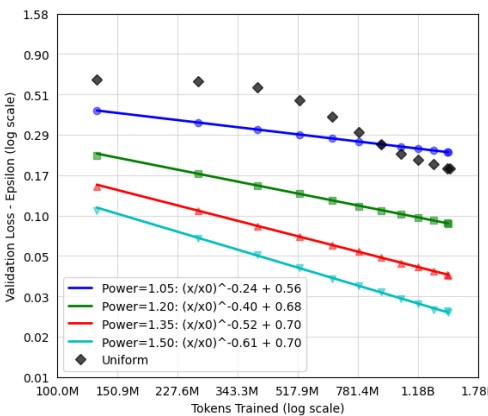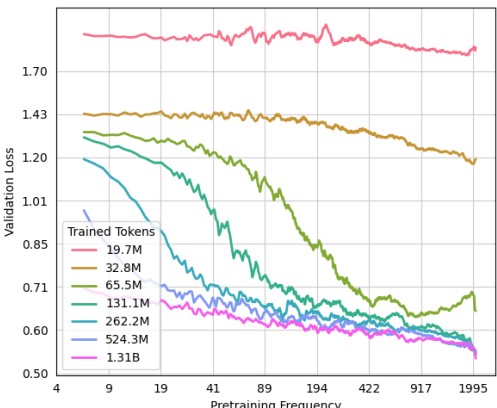

Figure 3: (a) Validation loss as a function of training data size. Models trained on data sampled from pretrained knowledge under various power-law distributions (i.e., $p(i) \sim (x + b)^{\text{power}}$) show clear power-law scaling of loss with data size, while uniform sampling does not. A more skewed data distribution leads to faster loss decay. (b) Loss decomposition by data frequency class: high-frequency data is learned earlier, while lower-frequency data is acquired later during training.

Using (3), we can obtain the following decomposition of the optimal Bayesian cross-entropy loss.

**Corollary 4.3.** *Suppose $\pi$ is the prior of $\phi_{data}$. Under the same setting as Theorem 4.2, the averaged optimal Bayesian loss (per sentence) can be bounded as:*

$$\inf_{\boldsymbol{M}} \frac{1}{N} \mathop{\mathbb{E}}_{\phi_{data} \sim \pi} \mathop{\mathbb{E}}_{X_{1:N} \sim P_{\phi_{data}}} \left[ - \log P_{\boldsymbol{M}}(X_{1:N}) \right] = \widetilde{O}\left( \frac{d_{knw}}{N^{1-\alpha}} + \frac{n_s d_{syn}}{N} \right) + \frac{1}{N} H(X_{1:N} | \boldsymbol{\Phi}_{data}).$$

*where $H(X_{1:N} | \boldsymbol{\Phi}_{data}) = \mathbb{E}_{\phi_{data} \sim \pi}[H(X_{1:N} \mid \phi_{data})]$ is the irreducible part of the loss.*

Note that the optimal Bayesian redundancy bound in Theorem 4.2 consists of two distinct terms, corresponding respectively to the syntax and knowledge models. These two models exhibit significantly different learning behaviors: the redundancy for the syntax model decreases rapidly at a rate $\tilde{O}(N^{-1})$, whereas the redundancy for the knowledge model decreases more slowly at a rate $\tilde{O}(N^{\alpha-1})$. These differences highlight two distinct training phases: an early stage dominated by syntax redundancy reduction, and a later stage dominated by knowledge redundancy reduction.

Such a behavior is quite intuitive: Initially, the model primarily learns syntactic structures, since frequently recurring syntactic patterns yield substantial and immediate reductions in redundancy (and thus test cross-entropy loss). Moreover, as the syntax model can be captured by a parametric model, its redundancy decreases rapidly at the parametric learning rate $\widetilde{O}(N^{-1})$. This is also consistent with standard Bayesian results for parametric models [60, 12]. As training progresses, the model gradually incorporates factual knowledge; however, knowledge elements with lower frequencies receive fewer training examples, resulting in a slower reduction in redundancy.

Finally, we note that a few recent studies have also derived the data scaling law from the power-law data distribution in various stylized settings [35, 53, 8]. See Appendix A for more discussions.

**Experimental Validations:** We validate Theorem 4.2 through experiments using datasets sampled from pretrained knowledge following various power-law distributions as well as a uniform distribution (See the experimental details in Appendix G). In Figure 3(a), we observe that models trained with power-law sampled data exhibit a strong power-law relationship between validation loss and data size, accurately captured by the regression form $\text{loss} = (x/x_0)^{-\alpha} + \epsilon$. The fitted exponent $\alpha$ increases in magnitude as the data distribution becomes more skewed, indicating faster loss decay, which is consistent with our conclusion in Theorem 4.2. In contrast, models trained with uniformly sampled data deviate significantly from this power-law behavior. We note our theory does not cover such uniform distribution, as it cannot arise from our data modeling based on the PYCRP.

Figure 3(b) further decomposes validation loss by data frequency class over training steps. We find that during the initial training steps (from the topmost curve to the 2nd topmost curve), the loss reduces for data of all frequencies (although the loss reduction for higher frequency data is slightly larger). This indicates the syntax learning phase. Later, we observe that high-frequency data achieves

more rapid loss reduction early in training, while medium- and low-frequency data improve later. This confirms a frequency-dependent learning dynamic: the model first captures more common patterns and then gradually incorporates rarer knowledge as training progresses, which is consistent from the insight gained from Kolmogorov structure function (see Appendix C).

## 4.3 Model Scaling Law

In this section, we derive a power-law characterization for the model scaling law. Our theoretical results here are derived under slightly simplified assumptions (but retain essential insights). Specifically, we focus exclusively on the knowledge model and omit the syntax model, motivated by our earlier findings in Theorem 4.2, where we showed (and empirical observations confirm) that the syntax model is learned at a significantly faster rate.

We continue modeling the knowledge model $\phi_{\text{knw}} = \sum_{i=1}^{\infty} p_i \delta_{\phi_i}$ as an infinite mixture model but now make the following assumption on the mixing probabilities $p_i$.

**Assumption 4.4.** For the mixture model $\phi_{\text{knw}} = \sum_{i=1}^{\infty} p_i \delta_{\phi_i}$, the mixing probabilities $p_i$ follow a power-law distribution: $p_i = \zeta(1/\alpha)^{-1} i^{-1/\alpha}$, where $\zeta(1/\alpha) = \sum_{i=1}^{\infty} i^{-1/\alpha}$. [5]

In this section, we consider the optimal redundancy achievable by an omniscient model $M_C^*$ under the constraint $C$. That is, we may construct the model $M_C^*$ from the true data distribution $\phi_{\text{data}}$. The only constraint of $M_C^*$ is that the model capacity is at most $C$ bits. Due to the finite capacity, the model $M_C^*$ cannot memorize all the data nor the true distribution $P_{\phi_{\text{data}}}$, hence must apply lossy compression to the true model. In particular, We denote by $\mathcal{D}_i(m_i)$ the redundancy incurred by answering questions from knowledge cluster $\phi_i$, and $m_i$ denotes the constraint of the mutual information between the model and $\phi_i$ (one may think it as the memory allocated to compress $\phi_i$). The minimal achievable redundancy under a given memory constraint can be characterized by a distortion-rate function, and we make the following assumption of $\mathcal{D}_i(m_i)$. For the formal definition of $\mathcal{D}_i(m_i)$ and the justification of the assumption, see Appendix E.2.

**Assumption 4.5.** We assume that the distortion-rate function $\mathcal{D}_i(R)$ satisfies
1. There exists $c_3, c_4$ such that for all $i \in \mathcal{N}^+, R \leq c_3$, the distortion-rate function $\mathcal{D}_i(R) \geq c_4$.

2. There exists $c_{\max}, b_{\max}$ such that for all $i \in \mathcal{N}^+$, the distortion rate function $\mathcal{D}_i(R) \leq c_{\max} b_{\max}^{-R}$.

The redundancy minimization problem can be formulated as the following optimization problem:

$$\text{minimize} \quad \mathbb{E}_i[\mathcal{D}_i(m_i)] = \sum_{i=1}^{\infty} p_i \mathcal{D}_i(m_i), \tag{5}$$

$$\text{subject to} \quad \mathbb{I}(\Phi_0; M_C^*) \leq C, \quad m_i = \mathbb{I}(\phi_i; M_C^*) \geq 0 \text{ for all } i \in \mathbb{N}^+,$$

where $\Phi_0 = (\phi_1, \phi_2, \ldots)$. If we further assume that $\phi_i \perp \Phi_{-i} \mid M_C^*$ where $\Phi_{-i} = (\phi_1, \ldots, \phi_{i-1}, \phi_{i+1}, \ldots)$, i.e., $\phi_i$ is conditionally independent of the remaining cluster parameters given the model $M_C^*$. In other words, conditional on the model $M_C^*$, knowing $\phi_i$ does not provide any additional information about $\phi_j$ for $j \neq i$. Under this natural assumption, the model capacity constraint can be simplified as: $\mathbb{I}(\Phi_0; M_C^*) = \sum_{i=1}^{\infty} \mathbb{I}(\phi_i; M_C^*) \leq C$. See Appendix E.2 for the derivation. Let $\text{Red}_M(C)$ denote the optimal value of the optimization problem in (5), representing the minimal achievable redundancy under the model capacity constraint.

**Theorem 4.6.** *Under Assumption 4.5 and Assumption 4.4, the optimal value of the optimization problem under the given model size constraint satisfies:*

$$\text{Red}_M(C) = \Theta(C^{-1/\alpha+1}).$$

*Moreover, if we further assume that $D_k(R) = a_k b_k^{-R}$ for some constant $b_{\min} \leq b_k \leq b_{\max}, a_{\min} \leq a_k \leq a_{\max}$, the contribution of the $k$'s cluster is $p_k \mathcal{D}_k(m_k^*) = \Theta(\min\{k^{-1/\alpha}, C^{-1/\alpha}\})$, where $m_k^*$ is the solution of the optimization problem* (5).

**Experimental Validation:** In Figure 1(a), we show the empirical decomposition of validation loss by knowledge token frequency class as model size increases. Figure 1(b) presents the corresponding

---

[5]This choice of exponent $1/\alpha$ aligns with the asymptotic behavior of mixing weights in the Pitman–Yor Process $\text{PYP}(\alpha, \beta, \pi_{\text{knw}})$ (see Lemma H.2).

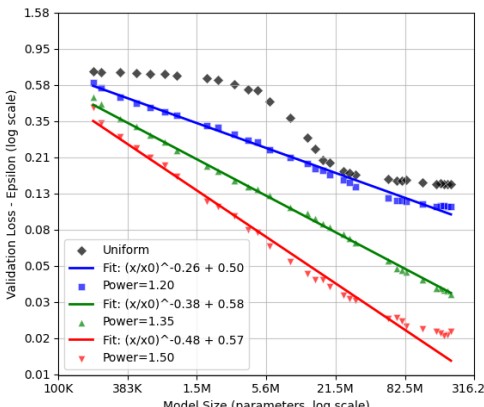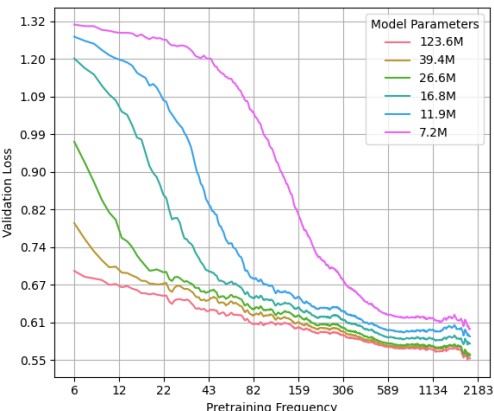

Figure 4: (a) Validation loss as a function of model size (excluding token embedding head). Regression analysis of model scaling: using the fitting form $loss = (x/x_0)^{-\alpha} + \epsilon$, we find that when the data is generated from a power-law distribution, the loss decreases with model size following a power-law trend. In contrast, the loss does not exhibit power-law scaling with model size when the data is generated from a uniform distribution. (b) Validation loss for different frequency classes: larger models are able to better fit lower-frequency (rarer) data, while for a fixed model size, more frequent data is learned more effectively.

theoretical prediction, derived from the optimal solution to the constrained optimization problem (5). The empirical results on power-law distributed data closely align with the theory: in both panels, high-frequency classes exhibit loss reduction with smaller models, while low-frequency classes only begin to improve as model capacity increases.

In Figure 4(a), we present the model scaling law across different data distributions. Consistent with our theoretical predictions, we observe that the more skewed the data distribution (i.e., the heavier the power-law tail), the larger the exponent in the fitted scaling law, indicating faster loss decay as model size increases. In contrast, *for data generated from a uniform distribution, the loss curve significantly deviates from a power law*: the loss remains nearly flat until the model reaches a critical capacity threshold, after which it drops sharply, and then plateaus again. This pattern arises for the following reason: initially, none of the individual items stand out and are fully learned, resulting in a flat loss. As the model grows, it begins to capture a subset of properties for most individuals, causing a rapid drop. Once all such properties have been learned, the loss flattens out again. A more detailed analysis of the learning dynamics for each property is presented in Appendix G.2. We also note that Allen-Zhu & Li [3] investigated factual knowledge acquisition using the bioS dataset consisting of uniformly distributed individuals, and one of their main findings is that the amount of acquired knowledge (in bits) scales linearly with the model size. However, they did not examine the scaling behavior in terms of cross-entropy loss, which follows a very different scaling than power law (Figure 4(a)).

Comparing the learning curves of uniformly distributed data with those of power-law-distributed data (across both data scaling and model scaling) suggests that *it can be advantageous to have power-law-distributed data*, because the model can gradually learn knowledge in the order of frequency, which is more effective than the uniform case, where no one element stands out and the model lacks guidance on what to prioritize. Exploring how adjusting the frequency or mixing ratio of data can enhance (or accelerate) learning performance is an intriguing direction for future research (see [27] for a recent study in this direction).

Our theorem also implies that for a fixed model capacity $C$, if a knowledge element appears with a frequency below a certain threshold, the model will choose not to learn it, despite that the model may have seen it many times during training. This aligns with our experimental findings: hallucination tends to occur when the total amount of knowledge exceeds the model's capacity, specifically, when higher-frequency knowledge already saturates the model's capacity, lower-frequency elements are ignored. As shown in Figure 2, for a 7.2M-sized model, knowledge that occurs fewer than 508 times is consistently hallucinated, regardless of the number of pretraining epochs.

## 5 Concluding Remarks

There are several exciting directions to extend this work. First, while we focused on factual knowledge, real-world datasets encompass more diverse forms of knowledge. It would be interesting to integrate more knowledge structures, as well as compositional reasoning and inference mechanisms into our theoretical model. Another interesting challenge is understanding how LLMs can approximate universal predictors (e.g., the Solomonoff predictor, which is also based on Kolmogorov complexity) within practical computational constraints (see, e.g., [25, 46]). Bridging these theoretical frameworks with real-world LLMs could deepen our understanding of their behaviors and pave the way for developing models that are more controllable and more reliable.

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

# A    Discussions and Related Work

**Prediction and Compression:** The link between prediction and compression is fundamental in both Shannon's probabilistic information theory and Kolmogorov's algorithmic information theory, forming the basis for efficient encoding and decoding of data [13, 50, 47]. In particular, the better one can predict the distribution of next symbol, the better one can compress the data (via arithmatic code) and vise versa [62, 58, 16, 48]. The connection of Kolmogorov's theory to LLMs and artificial intelligence was outlined in Hutter's theory of universal intelligence [34] and Ilya Sutskever's talk at Simons institute [69]. [16] advocate viewing language prediction as a compression problem and show that modern LLMs serve as powerful general-purpose compressors, outperforming traditional text compression tools such as gzip.

**Heap's, Zipf's Laws and Generative Models of Power Laws:** Heap's Law [30] an empirical relationship stating that the vocabulary size grows sublinearly ($\propto N^{\beta}$, for some $\beta$ between 0.4 and 0.7) with the size of a corpus $N$, revealing that new words appear at a diminishing rate. Zipf's Law [77], meanwhile, states that $f(r)$, the frequency of the $r$th frequent word, $\propto \frac{1}{r^{\alpha}}$, leading to a heavily skewed distribution dominated by leading terms. These complementary observations expose the disproportionate frequency of the words, especially "long tail" of rare words, and these long-tail effects are observed not only in linguistics but also in phenomena like city populations, animal populations, website traffic, etc.

One foundational theoretical model that produces power-law patterns is the classic Simon's model [66], which posits that each new element is more likely to replicate already-popular elements, thus creating a "rich-get-richer" effect. Bayesian nonparametric models, such as the Chinese Restaurant Process (CRP) and its generalization, the Pitman–Yor CRP (PYCRP), can also yield power-law distributions while benefiting from the property of exchangeability (the probability of any particular clustering remains unchanged by the order in which data points are observed).

**Syntax-Knowledge Modeling of Language:** There is a body of literature that explicitly separates or conceptually distinguishes syntax and knowledge models within language models. Here we only mention a few representative ones. [18] proposed a RNN-based model that learn syntactic structures (in the form of parse trees) alongside the generation of words. They did not explicitly incorporate a separate knowledge model.[23] models data as comprising disentangled two-type features, such as syntax and semantics. Then they analyze the Transformer's learning process, demonstrating that a two-stage dynamic emerges where the syntax component is learned first, which aligns with our Theorem 4.2. [44] proposed Grammar Variational Autoencoder combining variational autoencoders (VAEs) with formal grammars to generate syntactically valid structured data, explicitly separates the syntactic and semantic elements. [43] proposed the neural Abstract Meaning Representation model that, in some pipelines, first generates a syntactic skeleton, then integrates semantic content from AMR.

**Scaling Laws:** The study of neural scaling laws began with the observation that the population loss of trained deep neural networks follows a power-law relationship with respect to dataset size and model parameters. Early work by [63] introduced a joint error function that captured these dependencies, laying the groundwork for empirical analyses. Henighan et al.[31] subsequently expanded scaling laws to a broader range of architectures and tasks, while Kaplan et al.[42] demonstrated their robustness at vastly larger scales, showing that the loss scales as $L \propto (N/N_{\min})^{-\alpha_N}(D/D_{\min})^{-\alpha_D}$ where $N$ is the number of parameters, $D$ is the dataset size, and $\alpha_N, \alpha_D$ are scaling exponents. The Chinchilla study [33] later refined this framework by fitting their scaling law, and then identifying the compute-optimal frontier, showing that many prior models were undertrained, and demonstrating that scaling both parameters and data yields superior performance under fixed compute budgets. On the theoretical front, Bahri et al.[5] distinguished between variance-limited and resolution-limited regimes, identifying four distinct scaling behaviors. Sharma et al.[65] then linked scaling exponents to the intrinsic dimension of data manifolds, highlighting the role of data geometry in performance. More recently, Havrilla et al.[29] employed statistical and approximation theory to explain transformer scaling laws for low-dimensional data, and Daliri et al.[14] established convergence guarantees for 1-bit quantized networks, extending scaling principles to extreme weight-precision settings. Several works have also aimed to explain the origin of power-law behavior observed in scaling laws. Hutter [35] analyzed data scaling law under a stylized setting and showed that when the data distribution follows a power law, the resulting loss curve follows a power-law decay, rather than the common

$1/N$ rate. Subsequent works [53, 8] extended the setting in [35] and include aspects of model scaling, but still assumed a form for the loss on each data category, without considering the specifics of the learning algorithm. Maloney et al. [52] derived a power-law form for data scaling of the loss using random matrix theory, linking it to the power-law structure in the spectral properties of the data distribution. As in earlier studies, we also derive that power-law data distribution is the primary driver of the scaling law in our theoretical model. However, Our work differ in the following crucial aspects. First, we explicitly link LLMs to compression, and the scaling laws to the Kolmogorov structure function, a powerful view that can incorporate more complicated forms of language (rather than just facts). Second, we enrich the data-generation process with a syntax component, separating from underlying knowledge and yielding a more realistic model of language generation. Third, leveraging the coding/compression view, we rigorously quantify the redundancy for both Bayes-optimal models or any learning algorithm under mild regularity assumptions.

**Bayesian Mixture Code:** A common approach to constructing a universal code is to employ the *Bayesian mixture code*. It is well known that this Bayesian mixture code minimizes the Bayes risk/ redundancy [1] and the Bayes or minimax risk/redundancy can be characterized by the mutual information $I(\theta; X)$, which is closely tied to channel capacity [21, 13] (see also [17]); in the fixed-dimensional case, the worst (capacity-achieving) prior is the Jeffreys prior [12]. Recently, Jeon et al. [38] derived a Bayes risk/redundancy upper bound for the family of deep transformers and proposed a Bayesian meta-learning model to explain in-context learning. [6] Our data-generation model draws inspiration from their meta-learning model but differs in two key aspects: we explicitly separate the syntax model from the knowledge model, and we model the growing and power-law nature of knowledge. Jeon and Roy [37] proposed to employ an infinitely wide neural network as a nonparametric data-generation model to explain scaling laws, which is similar to our modeling in spirit. However, their theory predicts that the loss scales as $\widetilde{O}(1/M)$ and $\widetilde{O}(1/N)$, where $M$ denotes the model size and $N$ the data size, which does not capture the power-law scaling behavior observed in LLM practice (with exponents less than 1).

**Knowledge Acquisition:** Researchers have explored multiple mechanisms for external knowledge acquisition in LLMs, including pretraining on massive datasets [9], which further reveals a power-law relationship between training steps and the forgetting of memorization and generalization of factual knowledge, and shows that LLMs trained with duplicated training data are more prone to forgetting. In addition, recent studies find that continued pretraining contributes little to factual recall unless overfitting occurs [32], whereas targeted knowledge augmentation—via paraphrasing and diverse retrieval contexts—can more reliably inject domain-specific knowledge into LLMs for RAG applications [7]. Mallen et al. [51] also suggests that LLMs struggle to memorize less popular, long-tail facts even at large scales, and that retrieval augmentation can outperform much larger unassisted models while also improving efficiency. Recently, Gu et al. [27] demonstrated that knowledge acquisition can exhibit phase transitions with respect to data mixing and model size. In particular, their findings indicate that data with a mixing ratio below a certain threshold (depending on the model size) cannot be effectively learned, a result that is consistent in spirit with our model scaling law (see the proof of Theorem 4.6, in which there is a similar threshold). Furthermore, Ou et al. [54] examined the internal representation of knowledge by introducing the concept of "knowledge circuits," while addressing challenges related to ensuring factual accuracy and mitigating biases in scaled models. In addition, Allen-Zhu & Li [3] provide a quantitative perspective by estimating the number of knowledge bits a model can store, showing that LLMs are capable of storing up to 2 bits of factual knowledge per parameter and analyzing how architectural choices, training regimes, and data properties influence this capacity. Lu et al. [49] also discussed the relationship between fact knowledge capacity and model size and training epochs, finding they exhibit a linear and negative exponential law relationship.

**Cause of Hallucination:** While we focus on hallucinations arising from limited model capacity, prior work has identified a variety of causes. These include erroneous, outdated, or domain-incomplete data in the pretraining corpus [74]; biased data distributions [45]; instruction fine-tuning on unfamiliar or underrepresented data [41]; and knowledge shadowing [75], where dominant knowledge within the model suppresses less prominent information, leading to the generation of fabricated or inaccurate

---

[6]Although their work did not mention universal coding explicitly, their main result can be interpreted in terms of universal coding and redundancy.

details. [40] propose a distributional model (on facts) and prove that hallucinations must occur at a certain rate if the model satisfies a statistical calibration condition.

**Solomonoff's Universal Predictor:** We would like to mention that Solomonoff introduced a universal Bayesian mixture over all *Turing-computable* predictors [67, 68], known as the *Solomonoff predictor*, which is a major inspiration of the design of our knowledge model. It is known that Solomonoff's predictor achieves universally optimal prediction and compression rates in expectation for any computable sequence-generation process [57]. The connection to LLMs and meta-learning is also alluded in [16, 25]. However, this predictor remains a purely theoretical model and is not computable in practice. Exploring how modern LLMs might approximate the Solomonoff predictor (or a constrained version of it) is an intriguing direction for future research.

# B  Prediction and Compression

## B.1  LLMs as Compressors

For completeness, we provide a brief introduction to *arithmetic coding* and explain how Large Language Models (LLMs) can serve as lossless data compressors using this method. More details can be found in, e.g., [62, 13, 16, 48].

Suppose we have an autoregressive LLM $M$ that predicts the next-token probability $P_M(x_n \mid x_{1:n-1})$. Arithmetic coding is a popular method for lossless data compression, encoding each symbol (or token) based on its predicted probability. Specifically, arithmetic coding represents a data sequence as an interval within the real line between 0 and 1, sequentially refining this interval based on conditional probabilities. The process is as follows:

1. **Initialization:** Start with the interval $[0, 1)$.
2. **Subdivision:** Divide the current interval into subintervals, each proportional to the probabilities assigned to symbols in the alphabet.
3. **Encoding tokens:** For each symbol in the input sequence, refine the current interval to the corresponding subinterval associated with that symbol.
4. **Output:** After processing the entire sequence, select the shortest binary number (base 2) that lies within the final interval as the encoded output.

**Example B.1.** Consider an alphabet with symbols $A$, $B$, and $C$, having probabilities 0.5, 0.3, and 0.2, respectively. The initial interval $[0, 1)$ is divided as follows:

$$A : [0, 0.5), \quad B : [0.5, 0.8), \quad C : [0.8, 1).$$

To encode the message "$AB$", start with the interval $[0, 1)$. First, narrow it to $A$'s range $[0, 0.5)$. When processing the second symbol $B$, subdivide the interval $[0, 0.5)$ according to the conditional probabilities:

$$A : [0, 0.25), \quad B : [0.25, 0.4), \quad C : [0.4, 0.5).$$

Thus, the interval for "$AB$" is $[0.25, 0.4)$. We select the number 0.25, whose binary representation is 0.01.

The efficiency of arithmetic coding directly relates to the predictive accuracy of the underlying LLM. Formally, we have the following proposition [62] (see also [13, Ch. 13]).

**Proposition B.2.** *Let $P_M$ be the probability distribution predicted by an LLM $M$ for a data sequence $x_{1:n}$. Using arithmetic coding, the code length $L(x_{1:n})$ required to encode the sequence $x_{1:n}$ satisfies:*

$$L(x_{1:n}) \le -\log P_M(x_{1:n}) + O(1) = -\sum_{i=1}^{n} \log P_M(x_i \mid x_{1:i-1}) + O(1).$$

Consequently, viewing an LLM $M$ as a compressor, the total description length of the data $x$ includes two parts: the complexity $K(M)$ required to describe the model itself (architecture and parameters), and the data encoding length $L(x)$, bounded as in Proposition B.2.

[16, 48] showed that modern LLMs (such as Chinchilla 70B) can serve as powerful general-purpose compressors, significantly outperforming traditional text compression tools such as gzip and LZMA2.

## B.2 Universal Coding and The Coding Game

In this appendix, we briefly introduce the concepts of universal coding and the coding game and how these concepts are connected to perplexity minimization in LLMs.

### B.2.1 Universal Coding

The celebrate Shannon's source coding theorem ([64]) establishes a fundamental limit on achievable code rates by stating that for any code, the average code length $L$ is at least $H(P)$, where $H(\phi) = H(P) = \mathbb{E}_{x \sim P}\big[-\log_2 P(x)\big]$ is the Shannon entropy of the source distribution $P$. Moreover, if we know the source distribution $P$, we can encode the source message with average code length approaching $H(P)$.

However, real-world data sources often cannot be fully characterized by a single, known distribution. Designing coding schemes that adapt effectively to unknown distributions is known as *universal coding*. Universal coding seeks a single coding scheme $c$ that achieves near-optimal performance in terms of average code length over every distribution in a given family $\Theta$ (see e.g. [13, Ch. 13]). The additional cost compared to the entropy $H(P)$ is called the *redundancy* of the code $c$. suppose $\ell(c(x))$ is the length of the code $c(x)$ and $Q_c(x) = 2^{-\ell(c(x))}$ is defined to be the predictive probability corresponding to code $c$ ($Q_c$ is a valid probability distribution by Kraft inequality). The redundancy of code $c$ (or distribution $Q_c$) is formally defined as follows:

$$\mathsf{Red}(Q_c, P) = \mathbb{E}_{x \sim P}\big[\ell(c(x))\big] - H(P) = \mathbb{E}_{x \sim P}\big[-\log Q_c(x)\big] - H(P) \qquad (6)$$
$$= H(P \,\|\, Q_c) - H(P) = D_{\mathsf{KL}}(P \,\|\, Q_c).$$

Hence, finding an efficient universal code that minimizes the redundancy is equivalent to finding a predictive probability that minimizes the cross-entropy. A central goal in the study of universal coding is to design efficient universal coding schemes that can achieve sublinear redundancy ($\mathsf{Red} = o(n)$ for sequence of length $n$). Efficient universal coding schemes and tight redundancy bounds have been studied extensively in the information theory literature for a wide variety of distribution families (see, for example, [13, 59, 12, 78, 15, 4, 19]).

### B.2.2 A Coding Game and the Bayesian (Mixture) Strategy

In this subsection, we introduce a coding game that has many connections to online learning, information theory and Bayesian statistics. We mostly follow the exposition in [17].

Consider the following online Bayesian coding game. Suppose $P_\theta$ is a distribution indexed by $\theta$. Here the set of indices is $\Theta$ and the prior distribution of $\theta$ is $\pi(\theta)$ defined over $\Theta$. The player's goal is model the distribution $P_\theta$ as well as possible using a distribution $Q$. The nature chooses the distribution $P_\theta$ (according to the prior) and sample $n$ random variables $x_i \sim P_\theta$ sequentially. At step $i$, the player observes the history $x_{1:i-1}$, chooses the distribution $Q(x_i \mid x_{1:i-1})$, and suffers the log loss $-\log Q(x_i \mid x_{1:i-1})$. The overall objective is to minimize the Bayesian log-loss as follow:

$$\inf_Q \int_\Theta \pi(\theta) \mathop{\mathbb{E}}_{x_{1:n} \sim P_\theta} \left[\log \frac{1}{Q(x_{1:n})}\right] \mathrm{d}\theta = \inf_Q \int_\Theta \pi(\theta) \sum_{i=1}^n \mathop{\mathbb{E}}_{P_\theta} \left[\log \frac{1}{Q(x_i \mid x_{1:i-1})}\right] \mathrm{d}\theta.$$

In view of Proposition B.2, the above objective can also be thought as minimizing the Bayesian code length (hence the name of the game).

By Shannon's source coding theorem, we can see that the average code length cannot be smaller than

$$\int_\Theta \pi(\theta) \mathop{\mathbb{E}}_{x_{1:n} \sim P_\theta^n} \left[\log \frac{1}{P_\theta^n(x_{1:n})}\right] \mathrm{d}\theta = \int_\Theta \pi(\theta) H(P_\theta^n) \mathrm{d}\theta.$$

Here the superscript $n$ in $P_\theta^n$ indicates the distribution over a sequence of $n$ random variables. Subtracting this lower bound, the objective becomes minimizing the *Bayesian redundancy*:

$$\inf_Q \int_\Theta \pi(\theta) \mathop{\mathbb{E}}_{x_{1:n} \sim P_\theta} \left[\log \frac{1}{Q^n(x_{1:n})} - \log \frac{1}{P_\theta^n(x_{1:n})}\right] \mathrm{d}\theta$$
$$= \inf_Q \int_\Theta \pi(\theta) D_{\mathsf{KL}}(P_\theta^n \| Q^n) \mathrm{d}\theta = \inf_Q \int_\Theta \pi(\theta) \mathsf{Red}(Q^n, P_\theta^n) \mathrm{d}\theta \stackrel{\triangle}{=} \inf_Q \mathsf{Red}_n(Q, \Theta) \qquad (7)$$

**Bayesian Strategy:** For a probability family $\{P_\theta\}_{\theta \in \Theta}$, a common approach to constructing a universal code is to employ the *Bayesian strategy* (also called *Bayesian mixture code*). We place some prior distribution $\pi$ defined over $\Theta$, and consider the Bayesian mixture measure defined as

$$Q_\pi^n(x_{1:n}) = \int_\Theta P_\theta^n(x_{1:n})\, \pi(\theta)\, d\theta.$$

At time step $i$, the Bayesian strategy which uses the Bayes (posterior) estimator

$$Q_\pi(x_k = x \mid x_{1:k-1}) = \frac{Q_\pi^k(x_{1:k})}{Q_\pi^{k-1}(x_{1:k-1})}$$

as the next-token predictor. It is a classic result that this Bayesian mixture code minimizes the Bayesian redundancy $\mathsf{Red}_n(Q, \Theta) = \int_\Theta D_{\mathsf{KL}}(P_\theta \,\|\, Q)\, \pi(\theta)\, d\theta$ [1]. In fact, the Bayes redundancy can be characterized by the mutual information $I(\theta; X)$, which is closely tied to channel capacity [21, 12, 13]. See also [17, Ch. 19] and [38]. For reader's convenience, we summarize the results as the following lemma.

**Lemma B.3** ([12, 17, 37])**.** *The minimum Bayesian redundancy is attained by the Bayesian mixture code $Q_\pi$, and is equal to the mutual information between random variable $\theta$ (from the prior $\pi$ over $\Theta$) and the data $x_{1:n}$.*

$$\inf_Q \mathsf{Red}_n(Q, \Theta) = \inf_Q \int_\Theta \pi(\theta) D_{\mathsf{KL}}(P_\theta^n \| Q^n) d\theta = \int_\Theta \pi(\theta) D_{\mathsf{KL}}(P_\theta^n \| Q_\pi^n) d\theta = I(x_{1:n}; \theta).$$

*Here, $\theta \in \Theta$ is sampled from the prior $\pi$, and $x_{1:n}$ are sampled from $P_\theta^n$.*

*Proof.* We provide a simple proof for completeness. The proof is given in the single-observation case. The exact same argument applies in the $n$-observation case, by replacing $x$ by $x_{1:n}$. We first show that minimum Bayesian redundancy is attained by the Bayesian mixture code $Q_\pi$. For any distribution $Q$, we can see that

$$
\begin{aligned}
\mathsf{Red}_n(Q_\pi, \Theta) - \mathsf{Red}_n(Q, \Theta) &= \int_\Theta \pi(\theta) \left[ D_{\mathsf{KL}}(P_\theta \,\|\, Q_\pi) - D_{\mathsf{KL}}(P_\theta \,\|\, Q) \right] d\theta \\
&= \int_\Theta \pi(\theta) \sum_x P_\theta(x) \ln \frac{Q(x)}{Q_\pi(x)}\, d\theta = \sum_x \left[ \ln \frac{Q(x)}{Q_\pi(x)} \right] \int_\Theta \pi(\theta)\, P_\theta(x)\, d\theta \\
&= \sum_x Q_\pi(x) \ln \frac{Q(x)}{Q_\pi(x)} = -D_{\mathsf{KL}}(Q_\pi \,\|\, Q) \;\leq\; 0.
\end{aligned}
$$

The second part is simply rewriting the the definition of mutual information, as follows:

$$
\begin{aligned}
I(X; \theta) &= \mathbb{E}_{(\theta, X)}\left[ \ln \frac{P(X|\theta)}{P(X)} \right] = \mathbb{E}_{\theta \sim \pi, X \sim P_\theta}\left[ \ln \frac{P_\theta(X)}{\int_\Theta \pi(\theta')\, P_{\theta'}(X)\, d\theta'} \right] \\
&= \int_\Theta \pi(\theta) \int_X P_\theta(X) \ln \frac{P_\theta(X)}{Q_\pi(X)}\, dX\, d\theta \\
&= \int_\Theta \pi(\theta)\, D_{\mathsf{KL}}\left( P_\theta \,\middle\|\, Q_\pi \right) d\theta \;=\; \mathsf{Red}(Q_\pi, \Theta).
\end{aligned}
$$

This proves the lemma. $\qquad\square$

## C  Kolmogorov Structure Function and LLMs

### C.1  Basic Concepts

In this section, we briefly introduce the concepts of Kolmogorov complexity $K(\cdot)$ and Kolmogorov structure function $h_X(\cdot)$ (see the classic book [47] for more details), and how these concepts are connected to LLMs. Although $K(\cdot)$ and $h_X(\cdot)$ are not computable, their properties directly motivate our theoretical modeling in latter sections.

**Kolmogorov complexity:** The Kolmogorov complexity $K(X)$ of a string $X$ is defined as the length of the shortest binary program $p$ that outputs $X$ when run on a universal Turing machine $U$. Formally:

$K_U(X) := \min_p\{|p| : U(p) = X\}$ where $U$ is a fixed universal Turing machine, $p$ is a binary program and $|p|$ denotes the length of the program $p$ in bits. When there is no confusion, we omit the subscript $U$. The Kolmogorov complexity $K(X)$ measures the information content of $X$ and is sometimes referred to as the *algorithmic complexity* of $X$. $K(X)$ has been considered by many as *the "ultimate" notion for the compression* of $X$ [7] and modern LLMs can be thought as approximations of the Kolmogorov compressor (e.g., Ilya Sutskever's talk [69]).

While Shannon entropy $H(P_\phi)$ defines the fundamental limit for the expected code length of sequences drawn from a source $P_\phi$, $K(X)$ provides the ultimate, non-probabilistic compression limit for a specific sequence $X$. The two concepts are deeply linked: for a sequence $X$ that is typical according to a source $P_\phi$, its Kolmogorov complexity $K(X)$ is asymptotically close to its Shannon code length $-\log P_\phi(X)$. Consequently, the expected Kolmogorov complexity over the source, $\mathbb{E}_{X \sim P_\phi}[K(X)]$ approximates the Shannon entropy $H(P_\phi)$[26].

**Two-part code description:** One can describe the data $X$ by a two-part description: the model description, and the *data-to-model* code describing $X$ using the model. In particular, for any given lossless compressor $\boldsymbol{M}$, one can see that

$$K(X) \le K(\boldsymbol{M}) + L_{\boldsymbol{M}}(X) + O(1), \tag{8}$$

where $L_{\boldsymbol{M}}(X)$ represents the compressed length of $X$ when using a lossless compressor $\boldsymbol{M}$. When viewing an LLM as the compressor, the two-part description of the data $X$ consists of $K(\boldsymbol{M})$, the description of the LLM (the architecture and the parameters), and the data-to-model code of length $L_{\boldsymbol{M}}(X)$. Specifically, for an LLM that predicts the probability $P_{\boldsymbol{M}}(x_n \mid x_{1:n-1})$ of the next token auto-regressively, the code length $L_{\boldsymbol{M}}(X)$ can be bounded by $-\log P_{\boldsymbol{M}}(X) + O(1)$ since one can encode $X$ token by token using this many bits (via Arithmetic coding [62, 16]). See Appendix B.1 for details.

Then we briefly review Kolmogorov structure function and related concepts. See [47, Ch 5.5] or [71, 61] for more details.

We treat a set $S$ containing $X$ as a model describing $X$. According to the two-part description framework introduced earlier, one can describe every binary string $X$ by a two-part description: The model description in the form of a finite set $S$ that contains $X$, and the data-to-model code describing $X$ given $S$ (using $K(X \mid S)$ bits).

In Kolmogorov complexity, we say that a binary string $X$ of length $n$ is *random* if its Kolmogorov complexity $K(X) = n \pm O(1)$ (i.e., it is impossible to compress $X$ significantly). Extending this idea, one can define the *randomness deficiency* of $X$ with respect to a set $S$ (such that $X \in S$) as

$$\delta(X \mid S) \dot{=} \log |S| - K(X \mid S).$$

**Definition C.1 ("Best Fit" function).** It is also called the *minimal randomness deficiency* function, which is defined as:
$$\beta_X(\alpha) = \min_S \{\delta(X \mid S) : S \ni X; K(S) \le \alpha.\}$$

We say $X$ is a *typical* or *random* element of a finite set $S$, or $S$ is a fitting model for $X$, if $X \in S$ and the randomness deficiency $\delta(X \mid S) = O(1)$. Basically, it says that to describe $X$ in $S$, one essentially needs $\log |S| \pm O(1)$ bits (meaning $X$ is not very special in $S$). This definition parallels the concept of typical set in information theory [13, Ch. 3]. If $\delta(X \mid S)$ is small enough, $X$ satisfies *all properties* of low complexity with high probability for the elements of $S$ (see [71] for the details).

**Definition C.2 (Kolmogorov structure function).** Formally, it is defined as:
$$h_X(\alpha) = \min_S \{\log |S| : S \ni X; K(S) \le \alpha,\}$$

for any $\alpha > 0$. The set $S$ can be viewed as a candidate "typical set" for the data $X$. One may understand $S$ as the *model* of $X$ (the complexity of $S$, should be at most $\alpha$) and the term $\log|S|$ measures how many bits are needed to single out $X$ within $S$.

From the two-part description, it is easy to see that $K(X) \le K(S) + \log |S| + O(1)$. Hence, we have that
$$K(X) \le \alpha + h_X(\alpha) + O(1).$$

---

[7]Kolmogorov complexity is *universal* in the following sense: If $U$ is a universal TM, for any other TM $A$, there is a constant $c_A$ such that $K_U(X) \le K_A(X) + c_A$ for all string $X$. See [47].

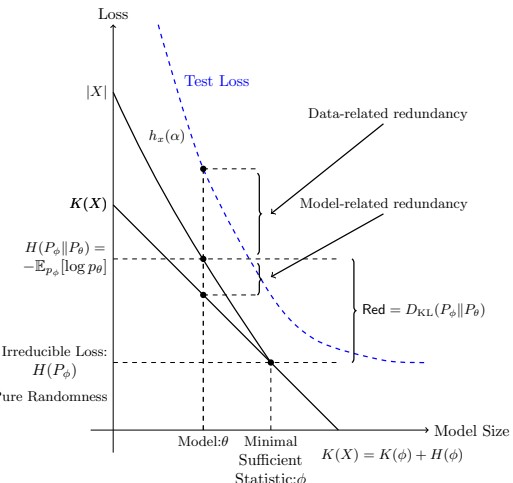

Figure 5: Kolmogorov Structure Function View of LLMs: The x-axis represents model size, while the y-axis represents the loss of the model, corresponding to code length of data given the model. The anti-diagonal solid straight line is the sufficiency line ($x + y = K(X)$), which is the lower bound of the code length of all possible two-part codes. The upper solid curve represents $h_X(\alpha)$. The dashed blue curve is the test loss of some LLM. See more details about the test loss curve in Figure (b) in Figure 6.

Hence, $h_X(\alpha)$ cannot be below the *sufficiency line* $L : L(\alpha) + \alpha = K(X)$ (by more than an additive constant). For those $\alpha$'s such that $K(X) \leq \alpha + h_X(\alpha) + O(1)$, we say the corresponding model *sufficient statistics*, and the smallest such $\alpha$ the *minimum sufficient statistics* [20]. The notion of sufficient statistics here parallels the same notion in the probabilistic setting: a sufficient statistic for the data captures all essential and relevant information and suffices for answering any downstream questions. Any additional information beyond the sufficient statistic is treated as "random noise" with respect to these tasks. For instance, the exact phrasing of a fact may be irrelevant for any downstream tasks unless it is so widely quoted that the particular wording appears frequently in the data, in which case it belongs to the the sufficient statistic, rather than the noise.

In this paper, we use the probabilistic extension of the above definition (see [26, 71]). As Kolmogorov himself pointed out, the finite set model class is equivalent (up to small additive terms) to the model class of probability density functions [71, 20] If $X$ is genuinely "typical" under the model distribution $P_M$ (parametrized by $M$), then $\log|S|$ parallels the negative log-likelihood, $-\log P_M(X)$ (which is the code length of $X$ under model $M$, ignoring the integer constraint). Hence, we define

$$h_X(\alpha) = \min_M\{-\log P_M(X) : K(P_M) \leq \alpha.\}$$

The shape of the structure function $h_X(\alpha)$ has been studied by several authors [71, 61, 47]. A major results in this line of research is that $h_X(\alpha)$ can essentially assume all possible shapes as along as it satisfies monotonicity, $h_X(0) = |X|$, $h_X(K(X)) = 0$ and above the sufficiency line (up to logarithmic additive terms).

### C.2 Kolmogorov Structure Functions and LLMs

Recall from Eq. (1) that $-\log P_M(X)$ is the empirical cross-entropy loss, which is also code length of data $X$ using predictive model $M$(up to an additive constant). See Figure 5 and Figure 6 for a schematic diagram of Kolmogorov structure function in the context of LLMs (or any probabilistic language models). We explain several key aspects that motivate our theoretical modeling in this paper.

1. **Kolmogorov Structure Function and Scaling Laws**: The Kolmogorov Structure function measures the performance of an optimal compressor under a given model-size constraint. It provides deep insights into the structure of the data from the compression perspective as the complexity of compressor increases. Such relationship directly mirrors the *model scaling laws* observed in the context of LLMs, where the training cross-entropy loss

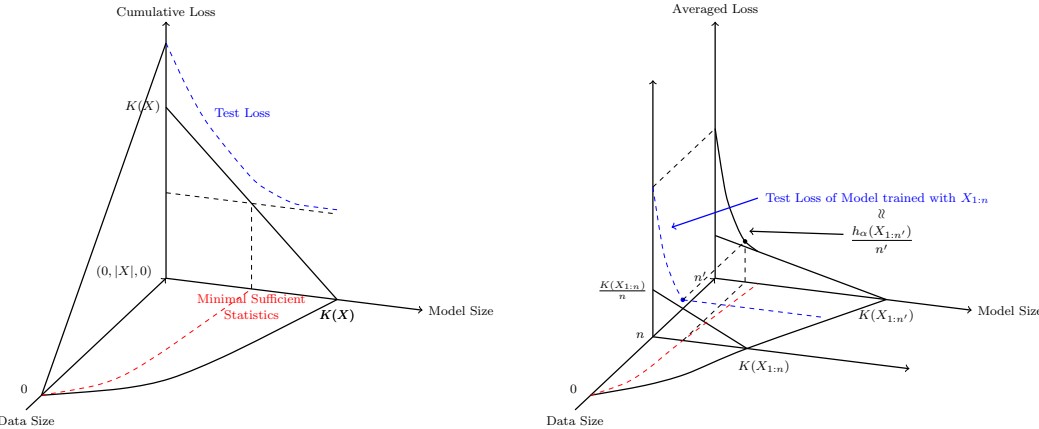

Figure 6: The red dashed curve in Data Size-Model Size place represents the minimal sufficient statistic, the blue dashed curve in Model Size-Loss place represents the test loss. (a) An overview of the relationship among data size, model size, and loss. (b) An additional illustration of test loss curve in Model Size-Loss place. We replace the cumulative loss with the average loss on the loss axis to better align with the setting of scaling laws. The test loss of the model trained with $X_{1:n}$ is approximately given by $\frac{h_\alpha(X_{1:n'})}{n'}$, where $\alpha$ denotes the minimal sufficient statistics of $X_{1:n}$ and $n' \gg n$ so that the average loss on $X_{1:n'}$ serves as a good approximation of the test loss.

(or perplexity) decreases as parameter count size grow [42, 33], and $h_X(\alpha)$ can be viewed a *theoretical lower bound* of the empirical scaling laws (for all possible LLM architectures).

2. **Structure of the Data:** From the perspective of the Kolmogorov structure function $h_X(\alpha)$, the structure of data $X$ (the regularities and randomness in $X$) is gradually revealed as the model size $\alpha$ (or complexity) increases. Initially, with relatively small $\alpha$, a simple model captures the most pervasive regularities – such as syntax in a linguistic dataset – because such patterns recur across every sentences (and therefore yield the greatest immediate reduction in coding cost or cross-entropy loss). As $\alpha$ grows, the model gains enough capacity to encode less frequent regularities, including widely shared factual knowledge that appears relatively frequently in the data, albeit less frequently than the universal syntax. Next, it captures rarer forms of knowledge that appear in smaller subsets of the data. Finally, for very large $\alpha$ exceeding the *minimal sufficient statistics*, the model starts to encode "random noise". We note that similar high-level ideas appeared in the study of the shape of the structure function [71, 61, 47].

3. **Minimal Sufficient Statistics and Randomness:** The *irreducible test loss* (e.g., the entropy term in (3) or (4)) corresponds to the lowest possible test loss, ideally achieved at the *minimal sufficient statistics*. Beyond this point, any further drop in *training* loss merely reflects memorizing only pure randomness, such as different re-phrasings of the same fact. In our data generation model (see Section 3), we adopt a probabilistic syntax model to capture such pure randomness in the language.

4. **Redundancy and Scaling Laws:** A central quantity in Figure 5 is the *redundancy* of the code, given by the KL divergence $D_{\mathsf{KL}}(P_{\phi_{\mathrm{data}}} \| P_M) = H(P_{\phi_{\mathrm{data}}} \| P_M) - H(P_{\phi_{\mathrm{data}}})$, where $P_M$ denotes the distribution predicted by the model and $P_{\phi_{\mathrm{data}}}$ the true data distribution. Hence, characterizing the redundancy curve $D_{\mathsf{KL}}(P_{\phi_{\mathrm{data}}} \| P_M)$ (varying the description length of the model $M$) provides the idealized model scaling law.

5. **Growing Knowledge Models:** Imagine that the data (consisting of sentences) has been generated sequentially. As the data size increases, it is natural to assume that the minimal sufficient statistics also grows, reflecting the expansion of "world knowledge" (e.g., newly discovered species, proteins, facts etc.). See Figure 6 in the appendix for a three-dimensional diagram (incorporating the $K(X)$ axis) to illustrate the shape of $h_X(\alpha)$ with increasing data sizes. This insight motivates our subsequent data model, in which the knowledge model grows as the data size (modeled using a *nonparametric* model).

To better understand the power-law behavior observed in empirical scaling laws of LLMs, an instructive question to consider is: *What structure in the data $X$ causes the redundancy to follow*

*a power-law shape?* Indeed, as pointed out by [35], different data structures can lead to different learning curves. To build intuition, first consider the example below that naturally yield a staircase shape.

**Example C.3.** Suppose $x$ is a binary string composed of three segments: 1. A segment of alternating bits (e.g., '010101...' repeated many times). 2. A segment with a more subtle pattern (e.g., inserting a '1' in every prime-numbered position). 3. A purely random segment.

At low model complexities $\alpha$, the best model only captures the obvious alternating pattern in the first segment, causing the first drop in $\log |S|$. Beyond a certain $\alpha$ threshold (the size of a Turing machine capable of generating prime numbers), the model learns the more nuanced pattern, resulting in a second drop. Beyond this point, $h_X(\alpha)$ meets the sufficiency line, and further increases in $\alpha$ merely allow the model to memorize the random segment almost bit by bit. □

Then we intuitively answer the question *What structure in the data X causes the redundancy to follow a power-law shape?*. Figure 5 illustrates the Kolmogorov structure function in a classical, finite setting. In this setting, we consider a finite-length sequence $X$ and the model code length for $X$ in its two-part code description is also finite. However, natural language exhibits a property where knowledge scales with data, compelling us to consider the case when model code length grows with the data size. To build intuition, we consider a simplified scenario that satisfies the above condition: data generated by an infinite-dimensional mixture model. We also assume the components of this model are encoded independently (i.e., the code length for a set of components is approximately the sum of their individual code lengths) for simplicity. We explore the intuition for model scaling in the asymptotic regime where data size grows infinitely large relative to the model size. Given a model size sufficient to encode $M$ components, the redundancy induced by this model size constraint corresponds to the sum of the tail mixture probabilities (i.e., the sum of all mixture probabilities excluding the $M$ largest). If the mixture probabilities of this model with infinite dimensions follow a power law, then the resulting redundancy (the sum of a tail that also follows a power law) also follows a power law. Conversely, if the sum of the tail probabilities follows a power law, we can also deduce that the mixture probabilities themselves follow a power law. This bidirectional relationship provides the intuition for our model scaling theory in Appendix E.2, and it can also serve as a theoretical bridge connecting empirically observed phenomena in language, such as Zipf's Law and Heaps' Law, with the model scaling observed in large language models.

We also note that Kolmogorov structure function is closely related to the rate-distortion function in Shannon's information theory. Specifically, the expected Kolmogorov structure function can be shown to be approximately equal to the rate-distortion function, see e.g., [26]. In our study of model scaling law in Appendix E.2, we leverage the concept of rate-distortion function as well.

As illustrated above, the Kolmogorov structure function provides insights into model scaling. In Figure 6, we extend this concept to incorporate the data scaling. Following the intuition provided earlier for the question *What structure in the data X causes the redundancy to follow a power-law shape?*, we consider that the data is generated from an infinite-dimensional power-law mixture model. In this case, the minimal sufficient statistics with respect to the model size also exhibit a power law, as shown in the left panel of Figure 6.

The right panel of Figure 6 illustrates the relationship between the Kolmogorov structure function and the test loss of an LLM training with finite data. Crucially, the test loss represents the expected loss of a trained model on unseen data, effectively an average loss over an infinitely large test set. In the right panel, this is approximated using a test set of size $n'$ much larger than the training set size $n$ (i.e., $n' \gg n$). To maintain consistency with this approximation, the y-axis in the right panel of Figure 6 represents average loss($h_\alpha(X_{1:n})/n$) rather than cumulative loss($h_\alpha(X_{1:n})$).

Intuitively, guided by the two-part code description and Kolmogorov structure function, the behavior of the test loss as a function of model size can be segmented into two phases around the minimal sufficient statistics. The connection is as follows: Before the model capacity reaches the minimal sufficient statistics for the training data $X_{1:n}$, the optimal test loss corresponds to the average structure function evaluated on a much larger dataset $X_{1:n'}$. As model size increases in this region, the test loss decreases because the model is capturing more of the data's true, generalizable structure. However, once the model capacity surpasses the minimal sufficient statistics, it begins to overfit by memorizing pure noise specific to the training set $X_{1:n}$. At this point, further increases in model size no longer improve generalization, causing the optimal test loss to plateau.

Therefore, the optimal redundancy achievable given the training data $X$ is equivalent to the redundancy obtained with infinite data, but under a model size constraint determined by the minimal sufficient statistics of $X$. We have previously established that for infinite data, the redundancy exhibits a power-law relationship with the model size constraint. Furthermore, as we show in the left panel of Figure 6 and demonstrated before, the size of the data ($|X|$) and the size of its minimal sufficient statistics also follow a power-law relationship. Combining these two factors implies that the optimal redundancy for training data $X$ also follows a power-law relationship with respect to the size of $X$.

## D    Details of the Syntax-Knowledge Model

In this section, we propose a hierarchical data generation model, called the *syntax-knowledge* model. In this model, where each sentence in the training set is generated by a *syntax encoder* that encode a (factual) knowledge element, sampled from the *knowledge model*. In our model, the syntax model (e.g., a probabilistic CFG or English grammar) does not grow indefinitely with the size of the dataset. Therefore, we assume that the distribution of the syntax model can be parametrized using a finite-dimensional parameter $\phi_{\text{syn}}$. On the other hand, the knowledge model employs a *non-parametric* stochastic process to account for two empirically observed phenomena: 1) the unbounded growth of factual information as datasets grow (mirroring Heap's Law in lexical growth patterns [30]), and 2) the long-tailed frequency distribution of factual occurrences, analogous to Zipf's law in natural language [76, 77].

To operationalize this above idea, we leverage the *Pitman–Yor Mixture Model(PYMM)* [56] for modeling the knowledge model. The preferential attachment mechanism in Pitman-Yor Process naturally captures both the *sublinear scaling* of new factual discoveries and the power-law distributed frequencies of knowledge pieces.

**Notations:** We denote the conditional probability $p(x \mid \phi)$ as $p_\phi(x)$, where $\phi$ is the latent variable that determines this probability distribution. The syntax model is parameterized by $\phi_{\text{syn}}$, the knowledge model is denoted as $\phi_{\text{knw}}$, and the entire data model is denoted as $\phi_{\text{data}} = \{\phi_{\text{syn}}, \phi_{\text{knw}}\}$.

The syntactic elements generated by the syntax model are denoted by $\boldsymbol{\xi}_{1:N}$, and the knowledge elements generated by the knowledge model are denoted by $\boldsymbol{\kappa}_{1:N}$. The corpus consists of sentences $X_{1:N}$, where each sentence $X_i$ is generated by the syntax model from a syntax–knowledge pair $(\boldsymbol{\xi}_i, \boldsymbol{\kappa}_i)$. Now, we provide the details below.

### D.1    The Non-Parametric Knowledge Model

We model the knowledge model as a Pitman-Yor Mixture Model (PYMM), which is a *nonparametric Bayesian* model. We first introduce the Pitman-Yor Process, from which the PYMM is constructed.

**Pitman-Yor Process:** The *Pitman-Yor Process (PYP)*, also known as the *Pitman-Yor two-parameter Poisson–Dirichlet process*, extends the Dirichlet Process (DP) and the Chinese Restaurant Process [56]. It has been applied to model a growing number of topics in the literature of topic modeling [70, 22]. A Pitman–Yor Process (PYP) is characterized by two real-valued parameters: the *discount parameter* $0 \leq \alpha < 1$ and the *concentration parameter* $\beta > -\alpha$, along with a base probability measure $\pi_{\text{knw}}$. We denote the process as $\text{PYP}(\alpha, \beta, \pi_{\text{knw}})$.

A sample from the Pitman–Yor process $\text{PYP}(\alpha, \beta, \pi_{\text{knw}})$ is a random probability measure $\phi_{\text{knw}} = \sum_{i=1}^{\infty} p_i \delta_{\phi_i}$, which is a discrete distribution with countably infinite atoms, where:

- Each atom $\phi_i \sim \pi_{\text{knw}}$ is the $i$-th cluster parameter, independently drawn from the base measure $\pi_{\text{knw}}$;

- $\delta_{\phi_i}$ denotes the Dirac delta measure centered at $\phi_i$;

- $p = (p_1, p_2, \ldots)$ are the weights generated by the Pitman–Yor Chinese Restaurant Process (PYCRP) (described below).

Imagining a restaurant where customers arrive sequentially, each choosing either to join an existing lively table or start their own, resulting in a naturally evolving clustering structure.

- The first customer sits at a new table.

- Suppose $N_k$ is the number of customers already at table $k$, and $K$ is the current number of occupied tables. The $n$-th customer either joins an existing table $k$ or starts a new table with the following probabilities:

$$\text{For the } n\text{-th customer:} \quad \begin{cases} \dfrac{N_k - \alpha}{n - 1 + \beta}, & \text{if joining an existing table } k, \\ \dfrac{\beta + \alpha K}{n - 1 + \beta}, & \text{if starting a new table,} \end{cases}$$

As the number of customers $n \to \infty$, the relative sizes of the tables (i.e., the proportion of customers at each table) converge in distribution to a random probability vector([55],Lemma H.1), denoted as:

$$p = (p_1, p_2, \dots) = \lim_{n \to \infty} (N_1/n, N_2/n, \cdots) \sim \mathsf{PYCRP}(\alpha, \beta)$$

This probability vector $p = (p_1, p_2, \dots)$ asymptotically exhibits a power-law distribution(see Lemma H.2).

**Pitman–Yor Mixture Model (PYMM):** In the Pitman–Yor Mixture Model, the Pitman–Yor Process $\mathsf{PYP}(\alpha, \beta, \pi_{\mathrm{knw}})$ serves as a prior over the space of mixture distributions. In particular, a random probability measure $\phi_{\mathrm{knw}} = \sum_{i=1}^{\infty} p_i \delta_{\phi_i}$, sampled from $\mathsf{PYP}(\alpha, \beta, \pi_{\mathrm{knw}})$, can be interpreted as a discrete mixture model: to generate a knowledge element from $\phi_{\mathrm{knw}}$, one first selects index $i \in \mathbb{N}$ with probability $p_i$, then draws a sample from the conditional distribution $P_{\phi_i}$, where $\phi_i \sim \pi_{\mathrm{knw}}$ is the $i$-th cluster parameter.

The knowledge elements generated by the knowledge model are abstract representations that capture the factual component underlying a sentence. We assume that the support of the knowledge element lies in a discrete set $\mathbb{K}$ (i.e., each knowledge element $\kappa \in \mathbb{K}$). Although the cardinality $|\mathbb{K}|$ may be potentially very large, we assume that $\log |\mathbb{K}|$ is quite small (since $\log |\mathbb{K}|$ is roughly the number of bits or tokens required to encode the knowledge element), and particularly much smaller compared to $N^\alpha$ for any $\alpha > 0$.

We assume that the base measure $\pi_{\mathrm{knw}}$, from which the knowledge cluster parameters $\phi_i$ are drawn, is supported on the bounded parameter space $\Phi_{\mathrm{knw}} = \{\phi_{\mathrm{knw}} \in \mathbb{R}^{d_{\mathrm{knw}}} : \|\phi_{\mathrm{knw}}\|_2 \leq 1\}$. Given the parameter spaces $\Phi_{\mathrm{knw}}$, we define corresponding parametric families of probability distributions: $\mathcal{P}_{\Phi_{\mathrm{knw}}} = \{P(\cdot \mid \phi_{\mathrm{knw}}) : \phi_{\mathrm{knw}} \in \Phi_{\mathrm{knw}}\}$, where $P(\cdot \mid \phi_{\mathrm{knw}})$ denotes the conditional distribution of the knowledge model given parameter $\phi_{\mathrm{knw}}$.

## D.2 Parametric Syntax Model

The syntax model generates syntactic elements conditioned on knowledge elements. Each knowledge element $\kappa \in \mathbb{K}$ deterministically selects an index $i \in \{1, \dots, n_s\}$, corresponding to a latent variable $\phi_{\mathrm{syn}}^{(i)}$ that parameterizes the conditional syntax model.

Conditioning the generation of syntax on a knowledge element $\kappa$ is thus equivalent to conditioning on the corresponding latent variable:

$$P(\cdot \mid \phi_{\mathrm{syn}}, \kappa) = P(\cdot \mid \phi_{\mathrm{syn}}^{(i)}).$$

Accordingly, the overall syntax model is parameterized as:

$$\phi_{\mathrm{syn}} = \{\phi_{\mathrm{syn}}^{(1)}, \phi_{\mathrm{syn}}^{(2)}, \dots, \phi_{\mathrm{syn}}^{(n_s)}\}.$$

The syntactic elements are abstract representations capturing surface-level variation such as word choice (e.g., synonyms), phrase structure (e.g., active vs. passive voice), and other randomness that does not affect the underlying semantics.

We assume that the prior distribution $\pi_{\mathrm{syn}}$, from which the conditional syntax model parameters $\phi_{\mathrm{syn}}^{(i)}$ are drawn, is supported on the bounded parameter space $\Phi_{\mathrm{syn}} = \{\phi_{\mathrm{syn}} \in \mathbb{R}^{d_{\mathrm{syn}}} : \|\phi_{\mathrm{syn}}\|_2 \leq 1\}$ for all $1 \leq i \leq n_s$. Given the parameter spaces $\Phi_{\mathrm{syn}}$, we define corresponding parametric families of probability distributions:

$$\mathcal{P}_{\Phi_{\mathrm{syn}}} = \{P(\cdot \mid \phi_{\mathrm{syn}}) : \phi_{\mathrm{syn}} \in \Phi_{\mathrm{syn}}\},$$

where $P(\cdot \mid \phi_{\mathrm{syn}})$ denotes the conditional distribution of the syntax model given parameter $\phi_{\mathrm{syn}}$.

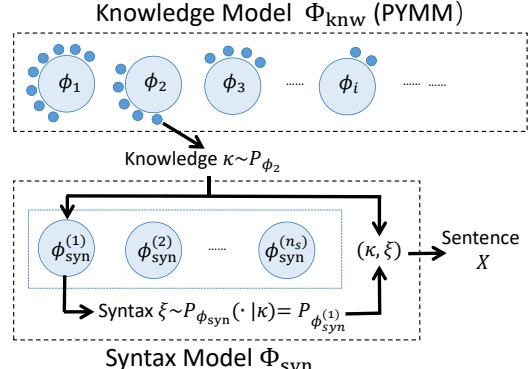

Figure 7: An illustration of the hierarchical Syntax-Knowledge data model

### D.3  Hierarchical Syntax-Knowledge Data Model

We propose that the Syntax–Knowledge model generates a sentence $X$ according to the following hierarchical Bayesian framework (see Figure 5):

1. Sample the latent variables for the knowledge and syntax models:

$$\phi_{\text{knw}} \sim \text{PYP}(\alpha, \beta, \pi_{\text{knw}}), \quad \phi_{\text{syn}} = \{\phi_{\text{syn}}^{(1)}, \phi_{\text{syn}}^{(2)}, \ldots, \phi_{\text{syn}}^{(n_s)}\}, \quad \phi_{\text{syn}}^{(i)} \overset{\text{i.i.d.}}{\sim} \pi_{\text{syn}}(\phi_{\text{syn}}).$$

2. Sample a knowledge element $\boldsymbol{\kappa}$:

$$\boldsymbol{\kappa} \sim P(\boldsymbol{\kappa} \mid \phi_{\text{knw}}).$$

3. Sample a syntactic element $\boldsymbol{\xi}$ conditioned on the sampled knowledge element $\boldsymbol{\kappa}$:

$$\boldsymbol{\xi} \sim P(\boldsymbol{\xi} \mid \phi_{\text{syn}}, \boldsymbol{\kappa}).$$

4. The syntax encoder generates the sentence $X$ from the syntax–knowledge pair $(\boldsymbol{\xi}, \boldsymbol{\kappa})$. We assume that the syntactic element $\boldsymbol{\xi}$ has already captured the randomness in the syntax, and there is a one-to-one mapping between the sentence $X$ and the syntax–knowledge pair $(\boldsymbol{\xi}, \boldsymbol{\kappa})$. Note that we do not assume the one-to-one mapping to be explicitly known or learnable. For the purpose of our theoretical analysis, it is sufficient to assume the existence of such a mapping, which ensures identifiability of the observed sentence with respect to its underlying syntax and knowledge elements.

By independently repeating steps 2, 3, 4 for $N$ times, we obtain a corpus $X_{1:N}$ consisting of $N$ i.i.d. sentences generated from the Syntax–Knowledge model.

We now illustrate how our data model generates data through a simple bioS dataset example. Note that our data model is not limited to such examples.

**Example D.1.** A concrete example is the generation of the bioS dataset designed in [3]. For example, a knowledge element $\boldsymbol{\kappa}$ encodes the factual knowledge about an individual: *[Person: Alice], [Event: Birth], [Date: January 1, 2000]*. A syntax element $\boldsymbol{\xi}$ corresponding to a sentence template. Conditioned on the knowledge element $\boldsymbol{\kappa}$, the syntax model chooses a syntax element $\boldsymbol{\xi}$, such as "[Subject] was born on [Date]". By composing $(\boldsymbol{\xi}, \boldsymbol{\kappa})$, the resulting sentence is: "Alice was born on January 1, 2000." Alternatively, the syntax model may select a different syntax element, "[Subject] entered the world on [Date]", yielding the sentence: "Alice entered the world on January 1, 2000." Note that the choice of syntax element is dependent on the knowledge element; for instance, if the knowledge element pertains to an animal, a different syntax element would be chosen.

## E  Explaining Scaling Laws

In Section 4.2, we established a data scaling law under the Bayesian setting for our Syntax-Knowledge model, showing that the optimal Bayesian redundancy decreases according to a power law with

respect to data size, with an exponent larger than $-1$. Given our earlier results from Section 4.2, we know the syntax model is learned at a significantly faster rate (also observed empirically), and thus the scaling law is primarily driven by the knowledge model. For simplicity, we ignore the syntax model in this section and focus exclusively on the knowledge model. In this section, we derive a similar power-law behavior under a slightly different set of assumptions. The key differentiating assumption is that knowledge is represented in a question-answering format, which ensures the identifiability of the mixture knowledge model. These assumptions greatly simplify the theoretical proof and allows us to derive both upper and lower bounds on redundancy, as long as the model satisfies Assumption E.3 (without requiring it to be an optimal Bayesian predictor). The central insight of this section is that the empirically observed scaling laws primarily stem from the power-law-distributed structure in the data. As a result, the assumptions made here do not affect the validity of our main conclusion. We also provide the omitted details from the model scaling law part (Section 4.3) in the main text.

We continue modeling the knowledge model $\phi_{\mathrm{knw}} = \sum_{i=1}^{\infty} p_i \delta_{\phi_i}$ as an infinite mixture model but now make the following assumption on the mixing probabilities $p_i$.

**Assumption E.1.** For the mixture model $\phi_{\mathrm{knw}} = \sum_{i=1}^{\infty} p_i \delta_{\phi_i}$, the mixing probabilities $p_i$ follow a power-law distribution: $p_i = \zeta(1/\alpha)^{-1} i^{-1/\alpha}$, where $\zeta(1/\alpha) = \sum_{i=1}^{\infty} i^{-1/\alpha}$ is the Riemann zeta function.

Note that the choice of exponent $1/\alpha$ aligns with the asymptotic behavior of mixing weights in the Pitman–Yor Process $\mathrm{PYP}(\alpha, \beta, \pi_{\mathrm{knw}})$ (see Lemma H.2).

In this section, we represent knowledge as question-answer pairs. We assume knowledge is given by pairs $(\psi, \omega)$, where $\psi$ denotes the question description, and $\omega$ is the answer of the question. Each knowledge cluster corresponds to a set of question descriptions $\Psi_i$, and generates knowledge pairs $(\psi, \omega) \sim P_{\phi_i}$ where $\psi \in \Psi_i$. We assume that the sets of question descriptions corresponding to different knowledge clusters are pairwise disjoint, i.e., $\Psi_i \cap \Psi_j = \emptyset$ for $i \neq j$.

Here, we are only concerned with whether the model can provide the corresponding answer given a specific question. Consequently, the redundancy of the model with respect to the $i$-th knowledge cluster is defined as follows:

**Definition E.2.** The redundancy of the model $M$ associated with the $i$-th knowledge cluster and conditioned on knowledge being drawn from this cluster, is defined as

$$\mathrm{Red}^{(i)} := \mathbb{E}_{q \sim P_{\phi_i}} \left[ D_{\mathsf{KL}} \left( P_{\phi_i}(\omega \mid \psi = q) \,\|\, P_M(\omega \mid \psi = q) \right) \right],$$

where $P_{\phi_i}$ denotes the data distribution induced by the cluster-specific parameter $\phi_i$, and $P_M$ is the predictive distribution defined by the model $M$.

The total expected redundancy of the model $M$ under the prior over knowledge clusters is then given by:

$$\mathrm{Red} := \sum_{i=1}^{\infty} p_i \mathbb{E}_{\phi_i \sim \pi_{\mathrm{knw}}} \left[ \mathrm{Red}^{(i)} \right].$$

### E.1 Data Scaling Laws

In this section, we derive the data scaling law under the assumption that there is no model capacity constraint. There are two key differences between this section and Section 4.2. First, we analyze the redundancy of a potentially suboptimal model under certain assumptions, whereas Section 4.2 focuses on the optimal case. Second, we study the test redundancy, in contrast to the Bayesian redundancy considered in Section 4.2, which corresponds to the averaged cumulative redundancy. Moreover, unlike Section 4.2, which provides only an upper bound, this section establishes both upper and lower bounds. The effect of dataset size on $\mathrm{Red}^{(i)}$ primarily arises from the number of occurrences of the question description $q \in \Psi_i$ in the training data. Accordingly, we express $\mathrm{Red}^{(i)}$ as $\mathrm{Red}_D(t_i)$, where $t_i$ is the number of such occurrences in the training data. The expected redundancy under data size constraint is then given by

$$\mathrm{Red}_D(N) = \mathbb{E}_{t_i} \left[ \sum_{i=1}^{\infty} p_i \mathbb{E}_{\phi_i \sim \pi_{\mathrm{knw}}} \left[ \mathrm{Red}_D^{(i)}(t_i) \right] \right],$$

where $t_i$ denotes a random variable representing the number of occurrences of the $i$-th knowledge cluster in a training dataset of size $N$.

Next, we present the assumptions on $\mathsf{Red}_D^{(i)}$ and derive the upper and lower bound for $\mathsf{Red}_D$ under these assumptions.

**Assumption E.3.** (I) The expected redundancy of unseen cluster of knowledge (i.e., $t_i = 0$) exceeds a certain constant $c_1$, i.e., $\mathbb{E}_{\phi_i \sim \pi_{\mathrm{knw}}}\left[\mathsf{Red}_D^{(i)}(0)\right] > c_1$ for all $k \in \mathcal{N}^+$.

(II) For all cluster of knowledge, there exists a constant $c_2$ such that $\mathbb{E}_{\phi_i \sim \pi_{\mathrm{knw}}}\left[\mathsf{Red}_D^{(i)}(t)\right] \leq \dfrac{c_2}{t+1}$ for all $i, t \in \mathcal{N}^+$.

Assumption (I) posits that the model exhibits constant redundancy when encountering unseen knowledge, which we consider reasonable. We now turn to the validity of Assumption (II). During language model training, the objective is to minimize the negative log-likelihood, which is equivalent to maximizing the likelihood function. When the model is viewed as a maximum likelihood estimator (MLE), the asymptotic normality of the MLE implies that the redundancy, in the asymptotic regime, is of the same order as specified in Assumption (II).

**Lemma E.4** (Asymptotic Normality of MLE). *Let $X_1, X_2, \ldots, X_n$ be $n$ i.i.d. samples drawn from a distribution $P_{\theta^*}$, and let $\hat{\theta}_n = \arg\max_\theta \ell(\theta)$ denote the maximum likelihood estimator. Then, as $n \to \infty$, the MLE satisfies*

$$\sqrt{n}(\hat{\theta}_n - \theta^*) \xrightarrow{d} \mathcal{N}(0, J(\theta^*)^{-1}),$$

*where $J(\theta^*)$ is the Fisher information matrix evaluated at $\theta^*$.*

Combining this conclusion with the result in Lemma H.6, we conclude that, under the asymptotic regime, the redundancy of the MLE matches the order assumed in Assumption (II). We assume that this order holds in all cases. Note that the assumption is stated with $1/(t+1)$ instead of $1/t$ to include the case where $t = 0$.

**Theorem E.5.** *Under (I) in Assumption E.3, the total expected redundancy $\mathsf{Red}_D$ satisfies*

$$\mathsf{Red}_D(N) = \Omega(N^{-1+\alpha}).$$

*Under (II) in Assumption E.3, the total expected redundancy $\mathsf{Red}_D$ satisfies*

$$\mathsf{Red}_D(N) = O(N^{-1+\alpha}).$$

*Proof of Theorem E.5.* Denote $f_i(t_i) = \mathbb{E}_{\phi_i \sim \pi_{\mathrm{knw}}}[\mathsf{Red}_D^{(i)}(t_i)]$.

We begin by proving the first part of the theorem.

$$
\begin{aligned}
\mathsf{Red}_D(N) &= \sum_{k=1}^{\infty} p_k \mathbb{E}_{t_k}[f_k(t_k)] \\
&\geq \sum_{k > N^\alpha} p_k \mathbb{E}_{t_k}[f_k(t_k)] \\
&\stackrel{(1)}{=} \sum_{k > N^\alpha} \sum_{l=0}^{N} p_k^{l+1}(1 - p_k)^{N-l} \binom{N}{l} f_k(l) \\
&\stackrel{(2)}{\geq} \sum_{k > N^\alpha} p_k(1 - p_k)^N f_k(0) \\
&\stackrel{(3)}{\geq} \frac{c_1}{\zeta(1/\alpha)} \sum_{k > N^\alpha} \frac{1}{k^{1/\alpha}} \left(1 - \frac{1}{\zeta(1/\alpha)N}\right)^N \\
&\geq \frac{c_1}{2e\zeta(1/\alpha)} \sum_{k > N^\alpha} \frac{1}{k^{1/\alpha}} \\
&= \Omega(N^{-1+\alpha}).
\end{aligned}
$$

(1) is obtained by expanding the expectation; (2) corresponds to taking the term with $l = 0$; and (3) is derived by substituting the value of $p_k = \dfrac{1}{k^{1/\alpha}\zeta(1/\alpha)}$.

Next, we prove the second part. For all $k \in \mathcal{N}^+$, we know that

$$
\begin{aligned}
p_k \mathbb{E}_{t_k}[f_k(t_k)] &\overset{(1)}{=} p_k \sum_{l=0}^{N} p_k^l (1-p_k)^{N-l} \binom{N}{l} f_k(l) \\
&= \sum_{l=0}^{N} p_k^{l+1}(1-p_k)^{N-l} \frac{l+1}{N+1} \binom{N+1}{l+1} f_k(l) \\
&\leq \frac{\max\limits_l (l+1) f_k(l)}{N+1} \sum_{l=0}^{N} p_k^{l+1}(1-p_k)^{N-l} \binom{N+1}{l+1} \\
&\overset{(2)}{\leq} \frac{\max\limits_l (l+1) f_k(l)}{N+1} \\
&\leq \frac{c_2}{N+1}.
\end{aligned}
$$

(1) is obtained by expanding the expectation, and (2) is derived by utilizing
$\sum\limits_{l=0}^{N} p_k^{l+1}(1-p_k)^{N-l}\binom{N+1}{l+1} \leq \sum\limits_{l=-1}^{N} p_k^{l+1}(1-p_k)^{N-l}\binom{N+1}{l+1} = 1.$

Therefore,

$$
\sum_{k \leq N^\alpha} p_k \mathbb{E}_{t_k}[f_k(t_k)] \leq \frac{c_2}{N+1} N^\alpha = O(N^{-1+\alpha}).
$$

For $k > N^\alpha$, we have

$$
\begin{aligned}
\sum_{k > N^\alpha} p_k \mathbb{E}_{t_k}[f_k(t_k)] &= \sum_{k > N^\alpha} \sum_{l=0}^{N} p_k^{l+1}(1-p_k)^{N-l} \binom{N}{l} f_k(l) \\
&\overset{(1)}{\leq} \sum_{k > N^\alpha} \sum_{l=0}^{N} \left(\frac{1}{\zeta(1/\alpha)k^{1/\alpha}}\right)^{l+1} \binom{N}{l} f_k(l) \\
&\overset{(2)}{\leq} \sum_{k > N^\alpha} \sum_{l=0}^{N} \left(\frac{1}{k^{1/\alpha}}\right)\left(\frac{1}{N}\right)^l \binom{N}{l} f_k(l) \\
&\overset{(3)}{\leq} \sum_{k > N^\alpha} \sum_{l=0}^{N} \left(\frac{1}{k^{1/\alpha}}\right) \frac{f_k(l)}{l!} \\
&= \sum_{l=0}^{N} \frac{f_k(l)}{l!} \sum_{k > N^\alpha} \frac{1}{k^{1/\alpha}} \\
&\leq c_2 \sum_{l=0}^{N} \frac{1}{(l+1)!} \sum_{k > N^{1/\alpha}} \frac{1}{k^{1/\alpha}} \\
&\leq c_2 e \sum_{k > N^\alpha} \frac{1}{k^{1/\alpha}} = O(N^{-1+\alpha}).
\end{aligned}
$$

(1) is obtained by substituting $p_k = \dfrac{1}{k^{1/\alpha}\zeta(1/\alpha)}$ and using the inequality $(1-p_k)^{N-l} \leq 1$; (2) follows from the fact that $\zeta(1/\alpha) > 1$ and $k > N^\alpha$; (3) uses the bound $\binom{N}{l} \leq \dfrac{N^l}{l!}$.

Therefore, we can see that

$$\mathsf{Red}_D(N) = \mathbb{E}_{k,t_k}\left[\sum_{k=1}^{\infty} p_k f_k(t_k)\right] \leq O(N^{-1+\alpha}).$$

This completes the proof. □

## E.2 Model Scaling Laws

In this section, we consider the optimal redundancy achievable by an omniscient model $\boldsymbol{M}_C^*$ under a model capacity constraint. (That is, the model $\boldsymbol{M}_C^*$ has access to the true data distribution $\phi_{\text{data}}$.) Due to the finite capacity, the model $\boldsymbol{M}_C^*$ must apply lossy compression to each $\phi_i$, resulting in a corresponding redundancy $\mathsf{Red}_M^{(i)}(m_i)$, where $m_i$ denotes the constraint of the mutual information between the model and $\phi_i$ (one may think it as the memory allocated to compress $\phi_i$). The minimal achievable redundancy under a given memory constraint conditioned on a given cluster of knowledge is characterized by the distortion-rate function:

$$\mathcal{D}_i(R) = \min_{\mathbb{I}(\phi_i;\boldsymbol{M}_C^*)\leq R} \mathbb{E}_{\phi_i \sim \pi_{\text{knw}}}\left[\mathbb{E}_{q\sim P_{\phi_i}}\left[D_{\mathsf{KL}}\left(P_{\phi_i}(\omega \mid \psi = q) \,\|\, P_{\boldsymbol{M}_C^*}(\omega \mid \psi = q)\right)\right]\right],$$

where $\mathbb{I}(\phi_i;\boldsymbol{M}_C^*)$ represents the information retained about $\phi_i$ after compression.

Furthermore, the omniscient model $\boldsymbol{M}_C^*$ allocates memory to each $\phi_i$ in a manner that minimizes the expected redundancy. Formally, the memory allocation corresponds to the solution of the following optimization problem:

$$\text{minimize} \quad \mathbb{E}_i[\mathcal{D}_i(m_i)] = \sum_{i=1}^{\infty} p_i \mathcal{D}_i(m_i),$$

$$\text{subject to} \quad \mathbb{I}(\Phi_0;\boldsymbol{M}_C^*) \leq C, \quad m_i = \mathbb{I}(\phi_i;\boldsymbol{M}_C^*) \geq 0 \text{ for all } i \in \mathbb{N}^+,$$

where $\Phi_0 = (\phi_1, \phi_2, \ldots)$ and $\mathbb{I}(\Phi_0;\boldsymbol{M}_C^*) \leq C$ represents the model capacity constraint.

If we further assume that $\phi_i \perp \Phi_{-i} \mid \boldsymbol{M}_C^*$, i.e., $\phi_i$ is conditionally independent of the remaining cluster parameters given the model $\boldsymbol{M}_C^*$, where $\Phi_{-i} = (\phi_1, \ldots, \phi_{i-1}, \phi_{i+1}, \ldots)$. In other words, conditional on the model $\boldsymbol{M}_C^*$, knowing $\phi_i$ does not provide any additional information about $\phi_j$ for $j \neq i$. Under this assumption, the model capacity constraint can be decomposed as follows:

$$\begin{aligned}
C \geq \mathbb{I}(\Phi_0;\boldsymbol{M}_C^*) &= \mathbb{I}(\Phi_{-1};\boldsymbol{M}_C^*) + \mathbb{I}(\phi_1;\boldsymbol{M}_C^* \mid \Phi_{-1}) \\
&= \mathbb{I}(\Phi_{-1};\boldsymbol{M}_C^*) + \mathbb{I}(\phi_1;\boldsymbol{M}_C^*) \\
&= \mathbb{I}(\Phi_{-2};\boldsymbol{M}_C^*) + \mathbb{I}(\phi_2;\boldsymbol{M}_C^*) + \mathbb{I}(\phi_1;\boldsymbol{M}_C^*) \\
&= \sum_{i=1}^{\infty} \mathbb{I}(\phi_i;\boldsymbol{M}_C^*).
\end{aligned}$$

Therefore, the optimization problem can be rewritten as:

$$\text{minimize} \quad \sum_{i=1}^{\infty} p_i \mathcal{D}_i(m_i), \tag{9}$$

$$\text{subject to} \quad \sum_{i=1}^{\infty} m_i \leq C, \quad m_i \geq 0 \text{ for all } i \in \mathbb{N}^+.$$

Let $\mathsf{Red}_M(C)$ denote the optimal value of the problem in (9), representing the minimal achievable redundancy under the model capacity constraint.

In general, the distortion-rate function is hard to express analytically. Here, we make the following assumptions.

**Assumption E.6.** We assume that the distortion-rate function $\mathcal{D}_i(R)$ satisfies

- there exists positive constant $c_3, c_4$ such that for all $i \in \mathcal{N}^+, R \leq c_3$, the distortion-rate function $\mathcal{D}_i(R) \geq c_4$,

- there exists positive constant $c_{\max}, b_{\max}$ such that for all $i \in \mathcal{N}^+$, the distortion rate function $\mathcal{D}_i(R) \leq c_{\max} b_{\max}^{-R}$.

The first assumption states that when the rate is small, the distortion cannot drop below a threshold, which is a natural property of rate-distortion theory for general sources.

The second assumptions can be formally justified within the framework of rate-distortion theory when the distortion function is the mean squared error (MSE) [13]. Although the distortion function in our setting is not MSE but rather the KL divergence between the probability distributions induced by the parameters, Lemma H.6 shows that the two distortion measures exhibit similar behavior in a local neighborhood.

- **Distortion Rate Function of Gaussian Distribution:** Let $X \sim \mathcal{N}(0, \sigma^2)$ be a zero-mean Gaussian source, and let the distortion measure be the mean squared error (MSE). The distortion-rate function $D(R)$ is defined as the minimum achievable distortion under a given rate $R$:

$$D(R) = \min_{p(\hat{x}|x):\mathbb{I}(X;\hat{X}) \leq R} \mathbb{E}[(X - \hat{X})^2]$$

  For the Gaussian source with MSE distortion, the closed-form solution is:

$$D(R) = \sigma^2 \cdot 2^{-2R}, \quad R \geq 0$$

- **Distortion-Rate Upper Bound via Maximum Entropy of Gaussian Distribution** Among all real-valued random variables with fixed variance, the Gaussian distribution achieves the maximum differential entropy:

$$h(X) \leq \frac{1}{2}\log(2\pi e \sigma^2),$$

  with equality if and only if $X \sim \mathcal{N}(0, \sigma^2)$. This implies that under a fixed distortion constraint and mean squared error (MSE) as the distortion measure, the Gaussian source requires the highest rate among all sources with the same variance.

  Therefore, the distortion-rate function of any real-valued source with variance $\sigma^2$ must satisfy:

$$D(R) \leq \sigma^2 \cdot 2^{-2R}, \quad R \geq 0,$$

  with equality achieved only when the source is Gaussian. This provides a universal lower bound on achievable distortion under MSE for a given rate $R$.

**Theorem E.7** (Resteatement of Theorem 4.6). *Under Assumption 4.5 and Assumption 4.4, the optimal value of the optimization problem under the given model size constraint satisfies:*

$$\mathsf{Red}_M(C) = \Theta(C^{-1/\alpha+1}).$$

*Moreover, if we further assume that $D_k(R) = a_k b_k^{-R}$ for some constant $b_{\min} \leq b_k \leq b_{\max}, a_{\min} \leq a_k \leq a_{\max}$, the contribution of the $k$-th cluster is*

$$p_k \mathcal{D}_k(m_k^*) = \Theta(\min\{k^{-1/\alpha}, C^{-1/\alpha}\}).$$

*where $m_k^*$ is the solution of the optimization problem.(9).*

*Proof of Theorem 4.6.* We first provide a lower bound for the optimal value. By Assumption 4.5, we have $D_k(m_k) \geq c_4$ whenever $m_k \leq c_3$. Since $\sum_{k=1}^{\infty} m_k \leq C$, there can be at most $\lfloor C/c_3 \rfloor$ indices $k$ such that $D_k(m_k) < c_4$. Due to the monotonicity of $p_k$, we know that

$$\sum_{k=1}^{\infty} p_k D_k(m_k) \geq \sum_{k > \lfloor C/c_3 \rfloor}^{\infty} p_k c_4 = \Omega(C^{-1/\alpha+1}). \tag{10}$$

Next, we solve the optimization problem (E.2) under assumption $D_k(R) = a_k b_k^{-R}$ using KKT condition. First consider the Lagrangian dual:

$$\mathcal{L}(k, \lambda, \mu) = \sum_{k=1}^{\infty} p_k \mathcal{D}_k(m_k) + \lambda \left( \sum_{k=1}^{\infty} m_k - C \right) - \sum_{k=1}^{\infty} \mu_k m_k.$$

We list the KKT conditions below:

$$\text{Stationarity:} \quad p_k \mathcal{D}'_k(m_k^*) + \lambda - \mu_k = 0, \quad \forall k \geq 1. \tag{11}$$

$$\text{Primal feasibility:} \quad \sum_{k=1}^{\infty} m_k^* \leq C, \quad m_k^* \geq 0, \quad \forall k \geq 1. \tag{12}$$

$$\text{Dual feasibility:} \quad \lambda \geq 0, \quad \mu_k \geq 0, \quad \forall k \geq 1. \tag{13}$$

$$\text{Complementary slackness:} \quad \lambda \left( \sum_{k=1}^{\infty} m_k^* - C \right) = 0, \quad \mu_k m_k^* = 0, \quad \forall k \geq 1. \tag{14}$$

Due to monotonicity, we know that $\sum_{k=1}^{\infty} m_k^* = C$. By the Stationary condition (11), we know that

$$\mathcal{D}'_k(m_k^*) = \frac{\mu_k - \lambda}{p_k}.$$

Due to the convexity of $\mathcal{D}_k$[13], we know that $\mathcal{D}'_k$ is monotonically increasing. Therefore, the inverse of $\mathcal{D}'_k$ exists, and we denote it as $g_k$.

For $m_k^* > 0$, we have $\mu_k = 0$, thus we know that

$$m_k^* = \max \left\{ g_k \left( \frac{-\lambda}{p_k} \right), 0 \right\}.$$

Next, we estimate the value of $\lambda$. Since we have assumed that $D_k(R) = a_k b_k^{-R}$ for some $b_{\min} \leq b_k \leq b_{\max}, a_{\min} \leq a_k \leq a_{\max}$, so there exists $r'_1, r'_2, d_1, d_2$ such that

$$-r'_1 \ln(d_1 x) \leq g_k(-x) \leq -r'_2 \ln(d_2 x).$$

Then, we know that

$$\max \left\{ 0, r_1 \ln \frac{d_1 k^{-1/\alpha}}{\lambda} \right\} \leq m_k^* \leq \max \left\{ 0, r_2 \ln \frac{d_2 k^{-1/\alpha}}{\lambda} \right\}.$$

where $r_1 = r'_1 \ln \zeta(1/\alpha), r_2 = r'_2 \ln \zeta(1/\alpha)$.

Assume $l_1$ is the maximum integer such that $d_1 l_1^{-1/\alpha} > \lambda$. We know that $d_1 (l_1 + 1)^{-1/\alpha} \leq \lambda < d_1 l_1^{-1/\alpha}$. Therefore, we have

$$C \geq \sum_{k=1}^{\infty} \max \left\{ 0, r_1 \ln \frac{d_1 k^{-1/\alpha}}{\lambda} \right\} = r_1 \sum_{k=1}^{l_1} \ln \frac{k^{-1/\alpha}}{\lambda}$$

$$\geq r_1 \sum_{k=1}^{l_1} \ln \frac{k^{-1/\alpha}}{l_1^{-1/\alpha}} = \frac{r_1}{\alpha} \left( \ln l_1^{l_1} - \ln l_1! \right).$$

By Stirling's formula, we know that $\ln l_1^{l_1} - \ln l_1! = \Theta(l_1)$. So we know that

$$C \geq \Theta(l_1) = \Theta(\lambda^{-1/\alpha}).$$

Similarly, assume $l_2$ is the maximum integer such that $d_2 l_2^{-1/\alpha} > \lambda$. We know that $d_2 (l_2 + 1)^{-1/\alpha} \leq \lambda < d_2 l_2^{-1/\alpha}$. Therefore, we have

$$C \leq \sum_{k=1}^{\infty} \max \left\{ 0, r_2 \ln \frac{d_2 k^{-1/\alpha}}{\lambda} \right\} = r_2 \sum_{k=1}^{l_2} \ln \frac{k^{-1/\alpha}}{\lambda}$$

$$\leq r_2 \sum_{k=1}^{l_2} \ln \frac{k^{-1/\alpha}}{(l_2 + 1)^{-1/\alpha}} = \frac{r_2}{\alpha} \left( \ln(l_2 + 1)^{l_2+1} - \ln l_2! \right).$$

By Stirling's formula, we know that $\ln l_2^{l_2} - \ln l_2! = \Theta(l_2)$. So we know that

$$C \geq \Theta(l_2) = \Theta(\lambda^{-1/\alpha}).$$

Since we know that $\Theta(l_1) = \Theta(\lambda^{-1/\alpha}) = \Theta(l_2)$, we have that

$$C = \Theta(l_1) = \Theta(l_2) = \Theta(\lambda^{-1/\alpha}).$$

For $k \leq l_1$, we have that

$$m_k^* \geq \max\left\{0, r_1 \ln \frac{d_1 k^{-1/\alpha}}{\lambda}\right\} \geq r_1 \ln \frac{d_1 k^{-1/\alpha}}{\lambda} > 0.$$

For $k > l_2$, we have that

$$m_k^* \leq \max\left\{0, r_2 \ln \frac{d_2 k^{-1/\alpha}}{\lambda}\right\} = 0.$$

Thus $m_k^* = 0$ for $k > l_2$. Hence, we have that

$$p_k \mathcal{D}_k(m_k^*) = \Theta(\min\{k^{-1/\alpha}, C^{-1/\alpha}\}).$$

Finally, we prove that $\mathsf{Red}_M(C) \geq \Theta(M^{-1/\alpha+1})$. Since we have assumed that $D_k(R) = a_k b_k^{-R}$ for some $b_{\min} \leq b_k \leq b_{\max}, a_{\min} \leq a_k \leq a_{\max}$, we know that there exists constant $d_3, d_4$ such that

$$d_3 \mathcal{D}'_k(m_k^*) \leq \mathcal{D}_k(m_k^*) \leq d_4 \mathcal{D}'_k(m_k^*)$$

Thus we know that

$$\sum_{k=1}^{\infty} p_k \mathcal{D}_k(m_k^*) \geq d_3 \sum_{k \leq l_2} \lambda + c_3 \sum_{k > l_2} p_k$$

$$= \Theta(\lambda l_2) + \Theta(l_2^{-1/\alpha+1}) = \Theta(M^{-1/\alpha+1}). \tag{15}$$

Similarly,

$$\sum_{k=1}^{\infty} p_k \mathcal{D}_k(m_k^*) \leq d_4 \sum_{k \leq l_1} \lambda + c_4 \sum_{k > l_1} p_k \tag{16}$$

$$= \Theta(\lambda l_1) + \Theta(l_1^{-1/\alpha+1}) = \Theta(C^{-1/\alpha+1}) \tag{17}$$

Combining Equation (15) and Equation (16) together, we have that

$$\sum_{k=1}^{\infty} p_k \mathcal{D}_k(m_k^*) = \Theta(C^{-1/\alpha+1}).$$

Note that the above solution is carried out under a strong assumption $D_k(R) = a_k b_k^{-R}$. Thus under Assumption 4.5, we have $\mathsf{Red}_M(C) \leq \Theta(C^{-1/\alpha+1})$. Combining this and (10), we have that

$$\mathsf{Red}_M(C) = \Theta(C^{-1/\alpha+1}).$$

$\square$

# F Fine-Tuning

Fine-tuning an LLM is typically used for two broad purposes: (1) instruction-following or (2) knowledge injection. Here, we focus on a standard instruction fine-tuning scenario. Consider, for instance, in our experiment, a model that has been pretrained on a bioS (multi+permute) dataset [3] and has already learned factual knowledge about specific individuals. During fine-tuning, our goal is to fine-tune the model to be able to answer the questions about these individuals, and thus the fine-tuning dataset consists of question–answer pairs with facts drawn from the same distribution as the pretraining data.

We view the generation of pretraining data and instruction fine-tuning data as a two-stage data generation process. The first stage, corresponding to the generation of pretraining data, follows the

data model introduced in Appendix D. In the second stage, i.e., the instruction fine-tuning stage, the knowledge model and the generation process of knowledge elements in the data remain unchanged; however, the syntactic elements are produced by a different syntax model (e.g., the question–answer format), which we denote by $\phi_{\text{ins}}$.

Let $X_{1:N}$ denote the pretraining corpus, which is generated by $(\phi_{\text{syn}}, \phi_{\text{knw}})$ and $X_{N+1:N+n}$ the instruction fine-tuning corpus, generated by $(\phi_{\text{ins}}, \phi_{\text{knw}})$. The full corpus is denoted by $X_{1:N+n}$. Following the same reasoning as in Section 4.2, the Bayesian redundancy of the full corpus $X_{1:N+n}$ is given by the mutual information

$$\mathbb{I}(X_{1:N+n}; \phi_{\text{syn}}, \phi_{\text{ins}}, \phi_{\text{knw}}).$$

In the following, we present a (somewhat heuristic) decomposition of the redundancy over the full corpus. The $\approx$ in the derivation below requires certain independence assumption, which may not hold exactly in real world, and bounding the approximation error is left as a future work.

As in our data generation process, there is a one-to-one mapping between a sentence $X$ and its syntax-knowledge element pair $(\boldsymbol{\xi}, \boldsymbol{\kappa})$. Hence, using the chain rule of mutual information, we can write

$$\mathbb{I}(X_{1:N+n}; \phi_{\text{syn}}, \phi_{\text{ins}}, \phi_{\text{knw}}) = \mathbb{I}(\boldsymbol{\kappa}_{1:N+n}; \phi_{\text{syn}}, \phi_{\text{ins}}, \phi_{\text{knw}}) + \mathbb{I}(\boldsymbol{\xi}_{1:N+n}; \phi_{\text{syn}}, \phi_{\text{ins}}, \phi_{\text{knw}} \mid \boldsymbol{\kappa}_{1:N+n})$$

Also note that the first and second terms in the RHS can be decomposed as

$$\mathbb{I}(\boldsymbol{\kappa}_{1:N+n}; \phi_{\text{syn}}, \phi_{\text{ins}}, \phi_{\text{knw}}) = \mathbb{I}(\boldsymbol{\kappa}_{1:N+n}; \phi_{\text{knw}}) + \mathbb{I}(\boldsymbol{\kappa}_{1:N+n}; \phi_{\text{syn}}, \phi_{\text{ins}} \mid \phi_{\text{knw}}),$$

$$\mathbb{I}(\boldsymbol{\xi}_{1:N+n}; \phi_{\text{syn}}, \phi_{\text{ins}}, \phi_{\text{knw}} \mid \boldsymbol{\kappa}_{1:N+n}) = \mathbb{I}(\boldsymbol{\xi}_{1:N+n}; \phi_{\text{syn}}, \phi_{\text{ins}} \mid \boldsymbol{\kappa}_{1:N+n}) + \mathbb{I}(\boldsymbol{\xi}_{1:N+n}; \phi_{\text{knw}} \mid \phi_{\text{syn}}, \phi_{\text{ins}}, \boldsymbol{\kappa}_{1:N+n}).$$

Note that $\mathbb{I}(\boldsymbol{\kappa}_{1:N+n}; \phi_{\text{syn}}, \phi_{\text{ins}} \mid \phi_{\text{knw}}) = 0$ since the knowledge elements are generated independently of the syntax model, and $\mathbb{I}(\boldsymbol{\xi}_{1:N+n}; \phi_{\text{knw}} \mid \phi_{\text{syn}}, \phi_{\text{ins}}, \boldsymbol{\kappa}_{1:N+n}) = 0$ since $\phi_{\text{knw}} - \boldsymbol{\kappa} - \boldsymbol{\xi}$ form a Markov chain. Hence, we can see that

$$
\begin{aligned}
\mathbb{I}(X_{1:N+n}; \phi_{\text{syn}}, \phi_{\text{ins}}, \phi_{\text{knw}}) &= \mathbb{I}(\boldsymbol{\kappa}_{1:N+n}; \phi_{\text{knw}}) + \mathbb{I}(\boldsymbol{\xi}_{1:N+n}; \phi_{\text{syn}}, \phi_{\text{ins}} \mid \boldsymbol{\kappa}_{1:N+n}) \\
&\overset{(1)}{\approx} \mathbb{I}(\boldsymbol{\kappa}_{1:N+n}; \phi_{\text{knw}}) + \mathbb{I}(\boldsymbol{\xi}_{1:N+n}; \phi_{\text{syn}}, \phi_{\text{ins}}) \\
&= \mathbb{I}(\boldsymbol{\kappa}_{1:N+n}; \phi_{\text{knw}}) + \mathbb{I}(\boldsymbol{\xi}_{1:N}; \phi_{\text{syn}}, \phi_{\text{ins}}) + \mathbb{I}(\boldsymbol{\xi}_{N+1:N+n}; \phi_{\text{syn}}, \phi_{\text{ins}} \mid \boldsymbol{\xi}_{1:N}) \\
&= \mathbb{I}(\boldsymbol{\kappa}_{1:N+n}; \phi_{\text{knw}}) + \mathbb{I}(\boldsymbol{\xi}_{1:N}; \phi_{\text{syn}}) + \mathbb{I}(\boldsymbol{\xi}_{1:N}; \phi_{\text{ins}} \mid \phi_{\text{syn}}) \\
&\quad + \mathbb{I}(\boldsymbol{\xi}_{N+1:N+n}; \phi_{\text{ins}} \mid \boldsymbol{\xi}_{1:N}) + \mathbb{I}(\boldsymbol{\xi}_{N+1:N+n}; \phi_{\text{syn}} \mid \phi_{\text{ins}}, \boldsymbol{\xi}_{1:N}) \\
&\overset{(2)}{\approx} \mathbb{I}(\boldsymbol{\kappa}_{1:N+n}; \phi_{\text{knw}}) + \mathbb{I}(\boldsymbol{\xi}_{1:N}; \phi_{\text{syn}}) + \mathbb{I}(\boldsymbol{\xi}_{N+1:N+n}; \phi_{\text{ins}})
\end{aligned}
$$

In the above, approximation $\overset{(1)}{\approx}$ relies on the assumption $\boldsymbol{\xi}_{1:N+n} \perp \boldsymbol{\kappa}_{1:N+n}$, and $\overset{(2)}{\approx}$ requires $\boldsymbol{\xi}_{N+1:N+n} \perp \boldsymbol{\xi}_{1:N}$ and $\boldsymbol{\xi}_{N+1:N+n} \perp \phi_{\text{syn}} \mid \phi_{\text{ins}}, \boldsymbol{\xi}_{1:N}$, both of which can be induced by the same assumption $\boldsymbol{\xi}_{1:N+n} \perp \boldsymbol{\kappa}_{1:N+n}$. Under the same assumption, the redundancy of the pretraining corpus can also be decomposed as follows:

$$
\begin{aligned}
\mathbb{I}(X_{1:N}; \phi_{\text{syn}}, \phi_{\text{knw}}) &= \mathbb{I}(\boldsymbol{\kappa}_{1:N}; \phi_{\text{syn}}, \phi_{\text{knw}}) + \mathbb{I}(\boldsymbol{\xi}_{1:N}; \phi_{\text{syn}}, \phi_{\text{knw}} \mid \boldsymbol{\kappa}_{1:N}) \\
&= \mathbb{I}(\boldsymbol{\kappa}_{1:N}; \phi_{\text{knw}}) + \mathbb{I}(\boldsymbol{\xi}_{1:N}; \phi_{\text{syn}} \mid \boldsymbol{\kappa}_{1:N}) \\
&\approx \mathbb{I}(\boldsymbol{\kappa}_{1:N}; \phi_{\text{knw}}) + \mathbb{I}(\boldsymbol{\xi}_{1:N}; \phi_{\text{syn}}).
\end{aligned}
$$

The redundancy of the instruction fine-tuning corpus can be written as the difference between the redundancy of the full corpus and the redundancy of the pretraining corpus:

$$\mathbb{I}(X_{1:N+n}; \phi_{\text{syn}}, \phi_{\text{ins}}, \phi_{\text{knw}}) - \mathbb{I}(X_{1:N}; \phi_{\text{syn}}, \phi_{\text{knw}}) \approx \underbrace{\mathbb{I}(\boldsymbol{\kappa}_{1:N+n}; \phi_{\text{knw}}) - \mathbb{I}(\boldsymbol{\kappa}_{1:N}; \phi_{\text{knw}})}_{\text{Red}_{\text{knw}}} + \underbrace{\mathbb{I}(\boldsymbol{\xi}_{N+1:N+n}; \phi_{\text{ins}})}_{\text{Red}_{\text{ins}}}$$

The two terms in the above equation correspond to the redundancy of knowledge elements $\text{Red}_{\text{knw}}$, and the redundancy of syntactic elements $\text{Red}_{\text{ins}}$ in the instruction fine-tuning corpus, respectively.

Following the same arguments in the proof of Theorem 4.2, we can see that $\mathbb{I}(\boldsymbol{\kappa}; \phi_{\text{knw}}) = \tilde{O}(N^\alpha)$, and hence for simplicity, we further assume that $\mathbb{I}(\boldsymbol{\kappa}; \phi_{\text{knw}}) = cN^\alpha$ for some constant $c > 0$. Then, the average redundancy of the instruction fine-tuning corpus satisfies:

$$
\begin{aligned}
\frac{1}{n}\text{Red}_{\text{knw}} &= \mathbb{I}(\boldsymbol{\kappa}_{1:N+n}; \phi_{\text{knw}}) - \mathbb{I}(\boldsymbol{\kappa}_{1:N}; \phi_{\text{knw}}) = \frac{(N+n)^\alpha - N^\alpha}{n} \overset{(1)}{=} O(N^{\alpha-1}), \\
\frac{1}{n}\text{Red}_{\text{ins}} &= \mathbb{I}(S_{N+1:N+n}; \phi_{\text{ins}}) = \tilde{O}(n^{-1}).
\end{aligned}
$$

where $\overset{(1)}{=}$ relies on the natural assumption that the size of pretrain corpus $N$ is much larger than the size of instruction fine-tuning corpus $n$. Similar to Theorem 4.2, the primary effect of fine-tuning in this scenario is that the model learns the new syntax to reduce the fine-tuning loss (i.e., redundancy) with fast rate $\tilde{O}(n^{-1})$, hence requiring relative fewer samples, while allowing the model to retain the factual knowledge it has already acquired ($N$ is large and the redundancy for the knowledge elements $O(N^{\alpha-1})$ is quite small).

On the other hand, consider the setting where our objective is to *inject new knowledge* via fine-tuning. Suppose the fine-tuning data uses a drastically different syntax or format from that used in pretraining, as well as completely new knowledge. Now, the Bayesian redundancy is equal to

$$\mathbb{I}(X_{1:N+n}; \phi_{\text{syn}}, \phi_{\text{ins}}, \phi_{\text{knw}}, \phi_{\text{nknw}}) \tag{18}$$

where $\phi_{\text{nknw}}$ is the new knowledge model. If the new syntax component is very different from the original one, and the knowledge component is also quite new (i.e., nearly independent from the pretraining), the mutual information (18) can be approximately decomposed into

$$\mathbb{I}(X_{1:N+n}; \phi_{\text{syn}}, \phi_{\text{ins}}, \phi_{\text{knw}}, \phi_{\text{nknw}}) \approx \mathbb{I}(X_{1:N}; \phi_{\text{syn}}, \phi_{\text{knw}}) + \mathbb{I}(X_{N+1:N+n}; \phi_{\text{ins}}, \phi_{\text{nknw}}).$$

The first term corresponds to the redundancy of the pretraining phase, and the second term corresponds to finetuning and the redundancy (per sentence) can be bounded as $\widetilde{O}(c_1/n + c_2/n^{1-\alpha})$ according to the same argument in Theorem 4.2. As $n$ is much smaller than $N$, we can see that the fine-tuning would experience a significant perplexity shift (even the entropy, which is the irreducible part of the loss, may also shift). Moreover, the additional syntax learning step not only slows down the learning, but also risks occupying the model's limited capacity, leading to forgetting of the pretrained knowledge, especially when the model's capacity is constrained. See the experimental results in Appendix G.2. Therefore, two practical recommendations are in order (consistent with good practices in prior empirical works [73, 24, 72]).

- Knowledge Injection: When injecting new knowledge during fine-tuning, avoid drastically different formats or syntaxes that deviate significantly from the original pretraining data, especially in the capacity constrained setting. By preserving familiar structures, the model can focus on absorbing new facts rather than first learning an unfamiliar syntax. Moreover, it is beneficial to adopt a gradual approach that mixes the pretrained data with a portion of new knowledge during finetuning, which helps mitigate large perplexity shifts and makes the optimization more stable [28, 36].

- Instruction Fine-Tuning. For instruction fine-tuning (e.g., adapting a model to question–answer formats), use a knowledge set that closely aligns with the data distribution seen during pretraining. This ensures that most of the fine-tuning effort is spent on learning new syntax or style. Moreover, if we use completely a new set of knowledge in the SFT stage, the model only focuses on compressing the SFT dataset, which slows down the learning and may lead to forgetting of pretrained knowledge as well.

# G   Experimental Setting and Additional Experimental Results

## G.1   Experiment Setting

Following the experimental setting from [2, 3], we generate profiles for 400,000 individuals. Each profile contains six attributes: date of birth, birth city, university, major, employer, and employer city. These attributes are used to populate a diverse set of templates (each type of information/instruction has 50 different templates), forming both pretraining and instruction fine-tuning datasets, as shown in Table 1. In the instruction tuning, we use data instances with odd indices (e.g., 1, 3, 5, 7, ...) as the training set and those with even indices (e.g., 2, 4, 6, 8, ...) as the test set.

As discussed in our theoretical section, the occurrence frequency of individuals in the pretraining/instruction fine-tuning dataset follows a power-law distribution, formulated as Equation 19:

$$P(i) = \frac{(i + \text{bias})^{-a}}{\sum_{j=1}^{N}(j + \text{bias})^{-a}} \tag{19}$$

Table 1: Examples of Pretraining and Instruction Fine-Tuning Data

| Dataset Type | Example |
|---|---|
| Pretraining | "Gracie Tessa Howell was born in Camden, NJ. He studied Biomedical Engineering and worked at UnitedHealth Group. He entered the world on April 15, 2081, and is employed in Minnetonka. He is an alumnus/alumna of Buena Vista College." |
| Instruction Fine-Tuning | "Q: What area of study did Gracie Tessa Howell focus on? A: Biomedical Engineering" |

where $i$ denotes the index of the individual, $N$ is the total number of individuals, and $a$ is the exponent parameter. Note that the frequency of data occurrence generated by a power-law distribution exhibits asymptotic behavior similar to that produced by the $\text{PYCRP}(1/a, \beta)$ (See Lemma H.2).

We set bias $= 1000$ and vary the parameter $a$ over $\{0, 1.05, 1.20, 1.35, 1.50\}$. If not specified, we choose $a = 1.20$. The final pretraining dataset contains 1.45B tokens (tokenized using the GPT-2 tokenizer), and the instruction fine-tuning dataset consists of 258M tokens. Pretraining is conducted for 4 epochs, totaling 5.8B tokens processed—almost twice the amount suggested by the Chinchilla scaling law.

**Model Configuration**   We conduct experiments with RoPE-encoded GPT-like models [39] of various sizes. The model configurations are detailed in Table 3.

**Training Procedure**   For training, we use a sequence length of 512 and batch size of 128. We apply a warmup ratio of 0.05 and a warmdown ratio of 0.9. Pretraining runs for 4 epoch with a learning rate of 0.0003, while fine-tuning runs for 1 epochs with a reduced learning rate of 0.00003. We employ weight decay of 0.1 and bf16 precision. To enhance parallelism, multiple sequences are packed into 512-token sequences, but cross-sequence attention is masked out.

## G.2   Experiment Results

**Data Heterogeneity**   We evaluate the accuracy of various properties under two different data distributions—uniform and power-law—to examine the heterogeneous nature of model learning. For each setting, we train models of varying sizes and measure their accuracy across multiple properties, revealing how capacity constraints influence what types of knowledge are learned first.

As shown in Figure 8, both settings demonstrate that models with limited capacity exhibit selective learning behavior—some properties are prioritized over others. Properties with lower entropy, such as Major, are easier to learn, as they contain less variability and thus can be compressed more efficiently by small models. In contrast, high-entropy properties require more capacity to capture and generalize. A notable example is the property *company city*. In both settings, this attribute starts with a non-trivial accuracy of approximately 7%, even when other properties remain close to zero. This is due to New York appearing with 7% frequency in the dataset. Small models, unable to generalize meaningfully, default to outputting the most frequent token—an effect more visible in the uniform setting where each person appears the same number of times in the pretraining corpus, i.e. no person is inherently easier.

However, the dynamics of this heterogeneity differ across the two distributions. Under the **power-law setting** (right), model accuracy for each property increases gradually as capacity grows. This suggests a smooth transition in learning, where dominant and compressible properties are acquired first, followed by rarer or more complex ones as capacity permits. In contrast, the **uniform setting** (left) reveals a more abrupt phase transition: most properties remain near-zero in accuracy until the model reaches a critical capacity threshold, after which multiple properties are rapidly acquired. This implies that, for capacity-constrained models, structuring knowledge injection with a skewed

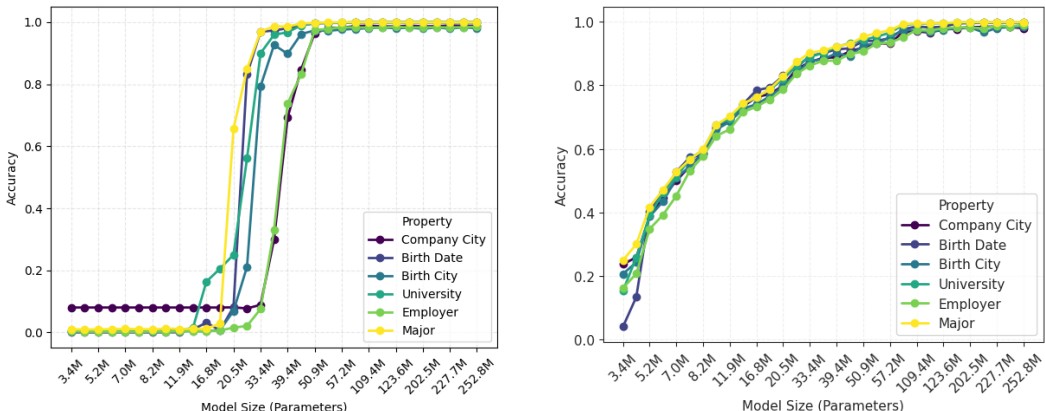

Figure 8: Accuracy of different properties across varying model sizes. The left panel shows the results under the uniform setting, while the right panel corresponds to the power-law setting. In both cases, model accuracy across properties exhibits heterogeneity—i.e., models prioritize certain properties over others. However, the trend differs: under the power-law setting, accuracy improves gradually across all properties as model size increases, whereas the uniform setting displays a sharp phase transition—accuracy remains low until a critical model size is reached, after which it rapidly improves.

distribution—rather than uniform—may ensure partial acquisition of salient knowledge, rather than risking a complete failure to learn under uniform allocation.

**Instruction Fine-Tuning**   We investigate how different instruction tuning strategies affect knowledge retention and acquisition under varying model capacities. We use uniform distribution in this experiment. Based on a shared pretrained model, we inject new knowledge using both Supervised Fine-Tuning (SFT) and Continued Pretraining (CPT), mixing in 80% of the original pretraining data to mitigate catastrophic forgetting following [10]. The underlying pretrained model is trained up to 50%, 90%, 100%, and 120% of its maximum capacity.

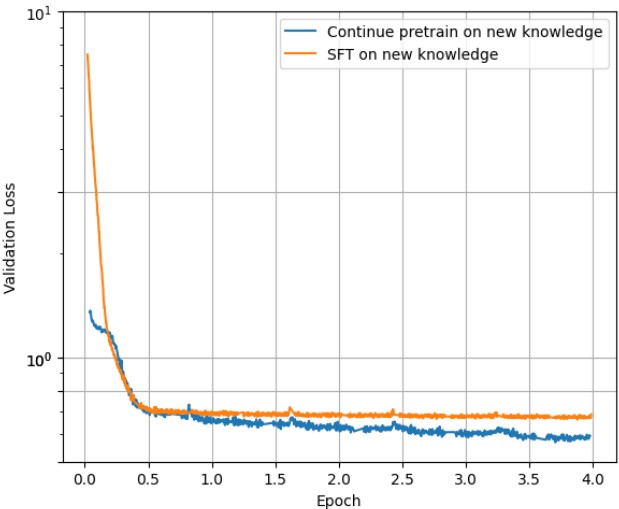

Figure 9: Training loss comparison between SFT and CPT. Due to format differences, SFT initially exhibits a higher loss.

While both SFT and CPT achieve perfect accuracy on new knowledge across all settings, they differ in how well they preserve prior knowledge. As shown in Table 2, SFT suffers from more severe forgetting, particularly in models trained at or beyond capacity (100% and 120%), where old accuracy drops significantly. This degradation is likely due to format-induced syntactic overhead in SFT, which consumes model capacity and leads to abrupt loss spikes (Figure 9). In contrast, CPT introduces new

Table 2: Accuracy on old and new knowledge after instruction tuning. New knowledge is learned equally well by both methods, but old knowledge is better retained with CPT, especially when the model is near or beyond capacity.

| | Accuracy on Old Knowledge (%) | | | |
| --- | --- | --- | --- | --- |
| | 50% | 90% | 100% | 120% |
| SFT (on old) | 100.0 | 93.25 | 89.9 | 78.5 |
| SFT (on new) | 100.0 | 100.0 | 100.0 | 100.0 |
| CPT (on old) | 99.6 | 96.5 | 94.5 | 85.1 |
| CPT (on new) | 100.0 | 100.0 | 100.0 | 100.0 |

knowledge more seamlessly, resulting in more stable loss and better retention of previously learned content. Interestingly, for under-capacity models (e.g., 50%), the difference is negligible—suggesting that forgetting primarily emerges when the model's capacity is saturated.

Table 3: Model Configurations with Parameter Counts

| Model Size | Layers | Heads | Emb Dim | Params (M) |
| --- | --- | --- | --- | --- |
| 6xs2 | 4 | 4 | 64 | 3.4 |
| 6xs | 3 | 4 | 80 | 4.3 |
| 5xs | 3 | 4 | 96 | 5.2 |
| 5xs2 | 3 | 4 | 112 | 6.1 |
| 5xs1 | 3 | 4 | 128 | 7.0 |
| 4xs | 4 | 4 | 128 | 7.2 |
| 4xs2 | 4 | 4 | 144 | 8.2 |
| 4xs1 | 4 | 4 | 192 | 11.4 |
| 3xs | 5 | 4 | 192 | 11.9 |
| 3xs2 | 5 | 4 | 224 | 14.3 |
| 3xs1 | 5 | 4 | 256 | 16.8 |
| xxs | 6 | 4 | 256 | 17.6 |
| xxs3 | 6 | 4 | 288 | 20.5 |
| xxs2 | 6 | 8 | 352 | 26.6 |
| xxs4 | 6 | 8 | 416 | 33.4 |
| xxs1 | 6 | 8 | 448 | 37.0 |
| xs | 7 | 8 | 448 | 39.4 |
| xs3 | 8 | 8 | 448 | 41.8 |
| xs2 | 8 | 8 | 512 | 50.9 |
| xs1 | 9 | 8 | 512 | 54.1 |
| s | 10 | 8 | 512 | 57.2 |
| s3 | 10 | 8 | 704 | 94.9 |
| s2 | 10 | 12 | 768 | 109.4 |
| s1 | 11 | 12 | 768 | 116.5 |
| base | 12 | 12 | 768 | 123.6 |
| m5 | 12 | 16 | 896 | 160.7 |
| m4 | 12 | 16 | 1024 | 202.5 |
| m3 | 13 | 16 | 1024 | 215.1 |
| m2 | 14 | 16 | 1024 | 227.7 |
| m1 | 15 | 16 | 1024 | 240.3 |
| m | 16 | 16 | 1024 | 252.8 |

## G.3  Real-world Data Validation

To validate the applicability of our theoretical framework to real-world scenarios, we conducted a series of experiments. We employed the same model architecture used in the synthetic data experiments and pre-trained our models on the large-scale Fineweb-edu dataset. We then analyzed and fitted the data scaling law and model scaling law observed during training. Furthermore, we evaluated the models' grasp of factual knowledge using the PopQA dataset, a popular question-

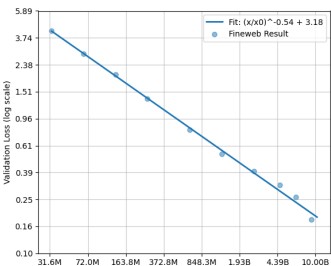 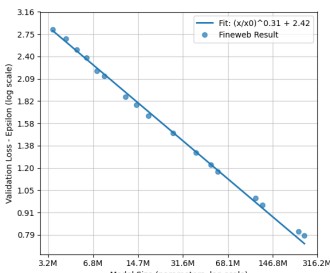 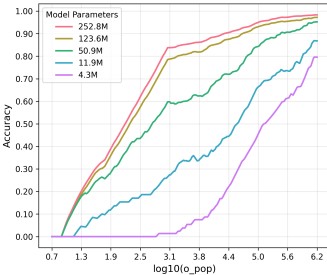

Figure 10: Experimental validation on real-world data. (a) Data scaling law on Fineweb-edu: Validation loss as a function of training data size. The fit indicates a power-law relationship, consistent with a power-law distribution of knowledge in the dataset. The estimated exponent for this power-law distribution of knowledge is between $1.35$ and $1.5$. (b) Model scaling law on Fineweb-edu: Validation loss as a function of model size. The plot shows a power-law decrease in loss with increasing model parameters, and also suggests a substantially larger irreducible loss component for natural language compared to synthetic data. (c) Knowledge cutoff on PopQA: Model performance (e.g., accuracy) on PopQA questions, categorized by the frequency of the underlying knowledge in the Fineweb-edu training set. Smaller models exhibit a clear inability to answer questions about low-frequency knowledge, while larger models demonstrate improved performance on rarer knowledge, effectively lowering the frequency cutoff for knowledge acquisition.

answering benchmark focusing on long-tail knowledge. In our PopQA experiments, we used the object popularity (o_pop) of a knowledge item in the training set as a proxy for its frequency.

By comparing the empirical exponents from the data and model scaling laws on Fineweb-edu against those from our synthetic experiments (which were generated using knowledge distributions with varying underlying power-law exponents, see discussions surrounding Figure 3 and Figure 4), we infer that the knowledge distribution in Fineweb-edu is effectively characterized by a power-law exponent between $1.35$ and $1.5$. This observation provides a potential estimate for the exponent range of real-world knowledge distributions and is consistent with the priors in our theoretical model (e.g., Assumption 4.4). When fitting the model scaling law, we observed that the inherent irreducible loss for real human language (English, in this case) is considerably larger than that observed for synthetic language. This is intuitive, as the complexity and inherent stochasticity of natural language far exceed the synthetic setting. Note that the irreducible loss values derived from the model scaling law and the data scaling law differ due to our finite model and dataset sizes. In practice, the observed loss can be expressed as $\mathcal{L} = (D/D_0)^{-\alpha} + (M/M_0)^{-\beta} + \varepsilon$, so when fitting the model scaling law (by fixing data size), the estimated irreducible term effectively includes the data-dependent component, i.e., $(D/D_0)^{-\alpha} + \varepsilon$; similarly, when fitting the data scaling law, the irreducible loss includes the model-dependent term, $(M/M_0)^{-\beta} + \varepsilon$. In other words, the discrepancy in irreducible loss arises from the limited model size or data size in each respective setting.

Tests on the PopQA dataset further illuminated the characteristics of knowledge acquisition in real models. Consistent with findings from our synthetic data experiments (see Figure 4(b) and discussion around Figure 2), we observed a "knowledge cutoff" phenomenon in real models as well. Specifically, even with sufficient training, smaller models struggle to learn knowledge that appears with low frequency in the training data. As model size increases, they become capable of learning knowledge at progressively lower frequencies.

## H  Omitted Proofs

### H.1  Properties of Pitman-Yor Mixture Model

**Lemma H.1** (Theorem 4.3 in [55])**.** *The mixing weights $p = (p_1, p_2, \cdots) \sim \mathsf{PYCRP}(\alpha, \beta)$ can be represented as a stick-breaking process:*

$$p_j = W_j \prod_{i=1}^{j-1} (1 - W_i),$$

*where $W_i$ are independent and $W_i \sim Beta(1 - \alpha, \beta + i\alpha)$.*

**Lemma H.2** (Theorem 3.13 in [55]). *Let $p = (p_1, p_2, \dots) \sim \mathrm{PYCRP}(\alpha, \beta)$ be the sequence of mixing weights drawn from a Pitman–Yor process. Then, the following limit almost surely exists:*

$$S_{\alpha,\beta} = \lim_{i \to \infty} i^{1/\alpha} p_i.$$

**Lemma H.3.** *Denote the remaining stick length of the stick-breaking process in Lemma H.1 as*

$$\mathsf{Len}_j := \prod_{i=1}^{j}(1 - W_i) = \sum_{i=j+1}^{\infty} p_i.$$

*Then the expectation of the remaining stick length satisfies:*

$$\mathbb{E}[\mathsf{Len}_n] = \Theta(n^{1-1/\alpha}).$$

*Proof.* Since $W_i \sim \mathrm{Beta}(1 - \alpha, \beta + i\alpha)$, we know that

$$\mathbb{E}[W_i] = \frac{\beta + i\alpha}{1 + \beta + (i-1)\alpha}.$$

Moreover, $W_i$ are independent. Thus we have

$$
\begin{aligned}
\mathbb{E}[\mathsf{Len}_n] &= \prod_{i=1}^{n} \frac{\beta + i\alpha}{1 + \beta + (i-1)\alpha} = \frac{\prod_{i=1}^{n}(\beta + i\alpha)}{\prod_{i=1}^{n}(1 + \beta + (i-1)\alpha)} \\
&= \frac{\alpha^n \cdot \prod_{i=1}^{n}\left(\frac{\beta}{\alpha} + i\right)}{\alpha^n \cdot \prod_{i=0}^{n-1}\left(\frac{1+\beta}{\alpha} + i\right)} = \frac{\Gamma\left(n + 1 + \frac{\beta}{\alpha}\right)}{\Gamma\left(1 + \frac{\beta}{\alpha}\right)} \cdot \frac{\Gamma\left(\frac{1+\beta}{\alpha}\right)}{\Gamma\left(n + \frac{1+\beta}{\alpha}\right)} \\
&= C \cdot \frac{\Gamma\left(n + 1 + \frac{\beta}{\alpha}\right)}{\Gamma\left(n + \frac{1+\beta}{\alpha}\right)}, \quad \text{where } C = \frac{\Gamma\left(\frac{1+\beta}{\alpha}\right)}{\Gamma\left(1 + \frac{\beta}{\alpha}\right)}.
\end{aligned}
$$

Then by Stirling's approximation, we know that that $\mathbb{E}(\mathsf{Len}_j) = \Theta(n^{1-1/\alpha})$. $\qquad\square$

### H.2 KL-divergence of Mixtures

**Lemma H.4.** *Let $\phi = \sum_{i=1}^{n} p_i \delta_{\phi_i}, \theta = \sum_{i=1}^{n} q_i \delta_{\theta_i}$ be two mixtures. Then we have*

$$D_{\mathsf{KL}}(P_\phi(x) || P_\theta(x)) \leq \sum_{i=1}^{n} p_i \log \frac{p_i}{q_i} + \sum_{i=1}^{n} p_i D_{\mathsf{KL}}(P_{\phi_i}(x) || P_{\theta_i}(x)).$$

*Proof.* By the log-sum inequality, we know that

$$
\begin{aligned}
D_{\mathsf{KL}}(P_\phi(x) || P_\theta(x)) &= \sum_{x} \left(\sum_{i=1}^{n} p_i P_{\phi_i}(x)\right) \log \frac{\sum_{i=1}^{n} p_i P_{\phi_i}(x)}{\sum_{i=1}^{n} P_{\theta_i}(x)} \\
&\leq \sum_{x} \sum_{i=1}^{n} \left(p_i P_{\phi_i}(x) \log \frac{p_i P_{\phi_i}(x)}{q_i P_{\theta_i}(x)}\right) \\
&= \sum_{i=1}^{n} p_i \log \frac{p_i}{q_i} + \sum_{i=1}^{n} p_i D_{\mathsf{KL}}(P_{\phi_i}(x) || P_{\theta_i}(x))
\end{aligned}
$$

$\qquad\square$

### H.3 KL-divergence and Fisher Information

**Definition H.5** (Fisher Information). Let $X$ be a random variable with probability density function (PDF) $P_\theta(x)$. The Fisher information is given by:

$$J(\theta) = \mathbb{E}_X\left[\left(\frac{\partial}{\partial \theta} \log P_\theta(X)\right)^2\right] = -\mathbb{E}_X\left[\frac{\partial^2}{\partial \theta^2} \log P_\theta(X)\right].$$

**Lemma H.6** (Exercise 11.7 in [13]). *For a parametric family $\{P_\theta(x)\}$ that*

$$\lim_{\theta' \to \theta} \frac{D_{\mathsf{KL}}(P_\theta \| P_{\theta'})}{(\theta - \theta')^T J(\theta)(\theta - \theta')} = \frac{1}{2}.$$

*Proof.* We provide a proof for completeness. We perform a second-order Taylor expansion of the KL divergence with respect to its second argument $\theta'$, around the first term $\theta$.

$$
\begin{aligned}
D_{\mathsf{KL}}(P_\theta \| P_{\theta'}) &= \mathbb{E}_{x \sim P_\theta} \left[ \log \frac{P_\theta(x)}{P_{\theta'}(x)} \right] \\
&= -\mathbb{E}_{x \sim P_\theta} \left[ (\theta - \theta') \nabla_\theta \log P_\theta + \frac{1}{2}(\theta - \theta')^T \nabla_\theta^2 \log P_\theta (\theta - \theta') \right] + o(|\theta - \theta'|^2) \\
&= -\mathbb{E}_{x \sim P_\theta} \left[ (\theta - \theta') \nabla_\theta \log P_\theta \right] - \frac{1}{2}\mathbb{E}_{x \sim P_\theta} \left[ (\theta - \theta')^T \nabla_\theta^2 \log P_\theta (\theta - \theta') \right] + o(|\theta - \theta'|^2) \\
&= -(\theta - \theta') \mathbb{E}_{x \sim P_\theta} \left[ \nabla_\theta \log P_\theta \right] - \frac{1}{2}(\theta - \theta')^T \mathbb{E}_{x \sim P_\theta} \left[ \nabla_\theta^2 \log P_\theta \right] (\theta - \theta') + o(|\theta - \theta'|^2) \\
&= -(\theta - \theta') \mathbb{E}_{x \sim P_\theta} \left[ \nabla_\theta \log P_\theta \right] + \frac{1}{2}(\theta - \theta')^T J(\theta)(\theta - \theta') + o(|\theta - \theta'|^2)
\end{aligned}
$$

The first-order term $(\theta - \theta')\mathbb{E}_{x \sim P_\theta} \left[ \nabla_\theta \log P_\theta \right] = 0$ since the KL divergence attains its minimum value of zero at $\theta' = \theta$.

Thus we know that

$$\lim_{\theta' \to \theta} \frac{D_{\mathsf{KL}}(P_\theta \| P_{\theta'})}{(\theta - \theta')^T J(\theta)(\theta - \theta')} = \frac{1}{2}.$$

$\square$

**Remark H.7.** *The conclusion of this lemma indicates that the Assumption 4.1 is essentially analogous to requiring the Fisher Information of the distribution family to be bounded. Such assumptions are common in related literature. For instance, Conditions 1 and 4 in [11] impose similar requirements. Notably, [11] also points out that these assumptions are widely satisfied within a class of exponential family distributions.*

### H.4 Proofs of Section 4.2

**Theorem H.8** (Restatement of Theorem 4.2). *Under the Bayesian sequential prediction framework and Assumption 4.1, the averaged optimal Bayesian redundancy (per sentence) of the hierarchical data model $\phi_{data}$ satisfies:*

$$\inf_M \frac{1}{N} \mathsf{Red}_N(\boldsymbol{M}, \boldsymbol{\Phi}_{data}) = \frac{1}{N}\mathbb{I}(X_{1:N}; \phi_{data}) = \widetilde{O}\left( \frac{d_{knw}}{N^{1-\alpha}} + \frac{n_s d_{syn}}{N} \right).$$

*Proof.* We use the *index of resolvability* approach similar to [6] to prove the theorem, by constructing a covering of the parameter space in terms of KL-divergence. However, we cannot directly apply the conclusion, as we are unable to explicitly construct a KL-divergence covering for $\mathcal{P}_{\boldsymbol{\Phi}_{data}}$. Instead, we can construct a set of probability distributions $\mathbb{Q}$ such that, for every specific $\phi_{data} \in \boldsymbol{\Phi}_{data}$, there exists a $Q \in \mathbb{Q}$ satisfying $D_{\mathsf{KL}}(\phi_{data} \| Q) \le f(\phi_{data})$, where $f(\phi_{data})$ is a value dependent on $\phi_{data}$.

Define the KL-covering number of the probability family $\mathcal{P}$ and the $L_2$ norm covering number of the parameter space $\Phi$ as:

$$N_{\mathrm{kl}}(\epsilon, \mathcal{P}) := \inf \left\{ n \in \mathbb{N} \mid \exists Q_i, i = 1, \ldots, n, \sup_{P \in \mathcal{P}} \min_i D_{\mathrm{kl}}(P \| Q_i) \le \epsilon^2 \right\}.$$

$$N_{L_2}(\epsilon, \Phi) := \inf \left\{ n \in \mathbb{N} \mid \exists \phi_i, i = 1, \ldots, n, \sup_{\phi \in \Phi} \min_i \|\phi - \phi_i\|_2 \le \epsilon \right\}.$$

We first construct KL-divergence coverings for the probability families $\mathcal{P}_{\boldsymbol{\Phi}_{\mathrm{syn}}}$ and $\mathcal{P}_{\boldsymbol{\Phi}_{\mathrm{knw}}}$, as well as a KL-divergence covering for the $n$-dimensional discrete simplex. We subsequently construct a point-wise approximate covering set $\mathbb{Q}$ for $\mathcal{P}_{\boldsymbol{\Phi}_{\mathrm{data}}}$ based on these coverings. By Assumption 4.1, we know that

$$N_{\mathrm{kl}}(\epsilon, P_{\boldsymbol{\Phi}_{\mathrm{syn}}}) \leq N_{L_2}(\epsilon/L_{\mathrm{syn}}, \boldsymbol{\Phi}_{\mathrm{syn}}) \quad \text{and} \quad N_{\mathrm{kl}}(\epsilon, P_{\boldsymbol{\Phi}_{\mathrm{knw}}}) \leq N_{L_2}(\epsilon/L_{\mathrm{knw}}, \boldsymbol{\Phi}_{\mathrm{knw}}).$$

Then we consider the $L_2$ norm covering number of the parameter space $\boldsymbol{\Phi}_{\mathrm{syn}}$. We simply divide each coordinate into $\lceil \frac{2L_{\mathrm{syn}}\sqrt{d_{\mathrm{syn}}}}{\epsilon} \rceil$ equal parts. It is easy to see that this results in an $L_2$ covering of the parameter space with a resolution of approximately $\frac{\epsilon}{L_{\mathrm{syn}}}$. Thus we know that

$$N_{\mathrm{kl}}(\epsilon, P_{\boldsymbol{\Phi}_{\mathrm{syn}}}) \leq N_{L_2}(\epsilon/L_{\mathrm{syn}}, \boldsymbol{\Phi}_{\mathrm{syn}}) \leq \left( \frac{2L_{\mathrm{syn}}\sqrt{d_{\mathrm{syn}}}}{\epsilon} \right)^{d_{\mathrm{syn}}}.$$

Similarly, we know that

$$N_{\mathrm{kl}}(\epsilon, P_{\boldsymbol{\Phi}_{\mathrm{knw}}}) \leq \left( \frac{2L_{\mathrm{knw}}\sqrt{d_{\mathrm{knw}}}}{\epsilon} \right)^{d_{\mathrm{knw}}}.$$

Next, we consider a KL-covering of the $n$-dimensional simplex.

Denote $\mathcal{P}_n = \{(p_1, \cdots, p_n); p_i \geq 0, \sum_{i=1}^{n} p_i = 1\}$. Consider the discretization of the simplex $\mathcal{Q}_{n,\epsilon} = \{(q_1, \cdots, q_n); \lfloor n/\epsilon^2 \rfloor q_i \in \mathcal{N}^+, \sum_{i=1}^{n} q_i = 1\}$.

We prove that $\mathcal{Q}_{n,\epsilon}$ is an $\epsilon^2$-KL-covering Then for any $(p_1, p_2, \cdots, p_n) \in \mathcal{P}_n$, there exists $(q_1, \cdots, q_n) \in \mathcal{Q}_{n,\epsilon}$ such that $|p_i - q_i| \leq \frac{\epsilon^2}{n}$. Therefore, we know that

$$\sum_{i=1}^{n} p_i \log \frac{p_i}{q_i} \leq \sum_{i=1}^{n} p_i \frac{p_i - q_i}{q_i} = \sum_{i=1}^{n} \frac{(p_i - q_i)^2}{q_i} \leq \sum_{i=1}^{n} \frac{\epsilon^2}{n} \leq \epsilon^2.$$

Note that a simple upper bound for $|\mathcal{Q}_{n,\epsilon}|$ is $|\mathcal{Q}_{n,\epsilon}| \leq \lfloor n/\epsilon^2 \rfloor^n$.

For notational simplicity, we denote

$$n_{\mathrm{syn}} = \left( \frac{2L_{\mathrm{syn}}\sqrt{d_{\mathrm{syn}}}}{\epsilon} \right)^{d_{\mathrm{syn}}}, \quad n_{\mathrm{knw}} = \left( \frac{2L_{\mathrm{knw}}\sqrt{d_{\mathrm{knw}}}}{\epsilon} \right)^{d_{\mathrm{knw}}},$$

as upper bounds on the $\epsilon-$KL-covering numbers of $\mathcal{P}_{\boldsymbol{\Phi}_{\mathrm{syn}}}$ and $\mathcal{P}_{\boldsymbol{\Phi}_{\mathrm{knw}}}$, respectively.

Assume that distributions
$$\{Q_{\mathrm{syn}}^{(1)}(\boldsymbol{\xi}), \ldots, Q_{\mathrm{syn}}^{(n_{\mathrm{syn}})}(\boldsymbol{\xi})\}$$
form an $\epsilon$-KL-covering of $\mathcal{P}_{\boldsymbol{\Phi}_{\mathrm{syn}}}$, and distributions
$$\{Q_{\mathrm{knw}}^{(1)}(\boldsymbol{\kappa}), \ldots, Q_{\mathrm{knw}}^{(n_{\mathrm{knw}})}(\boldsymbol{\kappa})\}$$
form an $\epsilon$-KL-covering of $\mathcal{P}_{\boldsymbol{\Phi}_{\mathrm{knw}}}$. Let $m = \lfloor N^\alpha \rfloor$ denote the truncation point used subsequently in the truncated estimation of $\mathcal{P}_{\boldsymbol{\Phi}_{\mathrm{knw}}}$. Then we construct the covering of $\mathcal{P}_{\boldsymbol{\Phi}_{\mathrm{data}}}$. Denote by $\mathcal{Q}$ the set of joint distributions over knowledge element $\boldsymbol{\kappa}$ and syntax element $\boldsymbol{\xi}$. A distribution $Q(\boldsymbol{\kappa}, \boldsymbol{\xi}) \in \mathcal{Q}$ if and only if it satisfies all of the following conditions:

- There exist $q = (q_1, \ldots, q_{m+1}) \in \mathcal{Q}_{m+1,\epsilon}$ and indices $k_1, \ldots, k_m$, such that

$$Q(\boldsymbol{\kappa}) = \sum_{i=1}^{m} q_i Q_{\mathrm{knw}}^{(k_i)} + q_{m+1} Q_u,$$

where $Q_u$ is the uniform distribution over the support of knowledge element $\mathbb{K}$.

- There exist indices $s_1, \ldots, s_{n_s}$, such that for each $i \in \{1, \ldots, n_s\}$,

$$Q(\boldsymbol{\xi} \mid \boldsymbol{\kappa} \in \mathbb{K}_i) = Q_{\mathrm{syn}}^{(s_i)},$$

where $\mathbb{K}_i$ is the set of knowledge corresponding to the syntax model $\phi_{\mathrm{syn}}^{(i)}$.

Next we prove that, for any $P_{\phi_{\text{data}}} \in \mathcal{P}_{\Phi_{\text{data}}}$, we have that

$$\min_{Q \in \mathcal{Q}} D_{\mathsf{KL}}(P_{\phi_{\text{data}}}(\boldsymbol{\kappa}, \boldsymbol{\xi})||Q(\boldsymbol{\kappa}, \boldsymbol{\xi})) \leq \mathsf{Len}_m \log |\mathbb{K}| + 3\epsilon^2,$$

where $\mathsf{Len}_j = \sum_{i=j+1}^{\infty} p_j$ is the remaining stick length of $\phi_{\text{knw}}$. (See Lemma H.3.)

We rewrite $\phi_{\text{knw}}$ as:

$$\phi_{\text{knw}} = \sum_{i=1}^{m} p_i \delta_{\phi_i} + \mathsf{Len}_m \delta_{\phi'},$$

where $\phi' = \dfrac{1}{\mathsf{Len}_m} \sum_{i=m+1}^{\infty} p_i \delta_{\phi_i}$. By the definition of $\mathcal{Q}$, there exists a distribution $Q^* \in \mathcal{Q}$ such that

$$Q^*(\boldsymbol{\kappa}) = \sum_{i=1}^{m} q_i^* Q_{\text{knw}}^{(k_i^*)} + q_{m+1}^* Q_u, \quad Q^*(\boldsymbol{\xi} \mid \boldsymbol{\kappa} \in \mathbb{K}_i) = Q_{\text{syn}}^{(s_i^*)},$$

and $Q^*$ satisfies the following three conditions:

- $D_{\mathsf{KL}}(P_{\phi_i}(\boldsymbol{\kappa})||Q_{\text{knw}}^{(k_i^*)}(\boldsymbol{\kappa})) \leq \epsilon^2$ for all $1 \leq i \leq m$.

- $\sum_{i=1}^{m} p_i \log \dfrac{p_i}{q_i^*} + \mathsf{Len}_m \log \dfrac{\mathsf{Len}_m}{q_{m+1}^*} \leq \epsilon^2$.

- $D_{\mathsf{KL}}(P_{\phi_{\text{syn}}^{(i)}}(\boldsymbol{\xi})||Q_{\text{syn}}^{(s_i^*)}(\boldsymbol{\xi})) \leq \epsilon^2$ for all $1 \leq i \leq n_s$.

Then by the chain rule of KL-divergence and Lemma H.4, we know that

$$
\begin{aligned}
D_{\mathsf{KL}}(P_{\phi_{\text{data}}}(\boldsymbol{\kappa}, \boldsymbol{\xi})||Q^*(\boldsymbol{\kappa}, \boldsymbol{\xi})) =& D_{\mathsf{KL}}(P_{\phi_{\text{knw}}}(\boldsymbol{\kappa})||Q^*(\boldsymbol{\kappa})) + \mathbb{E}_K \left[ D_{\mathsf{KL}}\left( P_{\phi_{\text{syn}}}(\boldsymbol{\xi}|\boldsymbol{\kappa})||Q^*(\boldsymbol{\xi}|\boldsymbol{\kappa}) \right) \right] \\
\leq& \sum_{i=1}^{m} p_i \log \frac{p_i}{q_i^*} + \mathsf{Len}_m \log \frac{\mathsf{Len}_m}{q_{m+1}^*} + \sum_{i=1}^{m} p_i D_{\mathsf{KL}}(P_{\phi_i}(\boldsymbol{\kappa})||Q_{\text{knw}}^{(k_i^*)}(\boldsymbol{\kappa})) \\
&+ \mathsf{Len}_m D_{\mathsf{KL}}(P_{\phi'}(\boldsymbol{\kappa})||Q_u(\boldsymbol{\kappa})) + \sum_{i=1}^{n_s} P(\boldsymbol{\kappa} \in \mathbb{K}_i) D_{\mathsf{KL}}(P_{\phi_{\text{syn}}^{(i)}}(\boldsymbol{\xi})||Q_{\text{syn}}^{(s_i^*)}(\boldsymbol{\xi})) \\
\leq& \epsilon^2 + \sum_{i=1}^{m} p_i \epsilon^2 + \mathsf{Len}_m \log |\mathbb{K}| + \sum_{i=1}^{n_s} P(\boldsymbol{\kappa} \in \mathbb{K}_i) \epsilon^2 \\
\leq& 3\epsilon^2 + \mathsf{Len}_m \log |\mathbb{K}|
\end{aligned}
$$

Subsequently, we derive an upper bound for the associated covering number. We know that $|\mathcal{Q}| \leq |\mathcal{Q}_{m+1,\epsilon}| n_{\text{knw}}^m n_{\text{syn}}^{n_s}$. That is to say

$$
\begin{aligned}
\log |\mathcal{Q}| &\leq \log |\mathcal{Q}_{m+1,\epsilon}| + m \log n_{\text{knw}} + n_s \log n_{\text{syn}} \\
&\leq 2m \log \frac{m}{\epsilon^2} + m d_{\text{knw}} \log \frac{L_{\text{knw}} d_{\text{knw}}}{\epsilon} + n_s d_{\text{syn}} \log \frac{L_{\text{syn}} d_{\text{syn}}}{\epsilon}.
\end{aligned}
$$

Finally, we prove the conclusion in this theorem. Consider $Q_0 = \frac{1}{|\mathcal{Q}|} \sum_{Q \in \mathcal{Q}} Q^N$. We can bound the redundancy as follows:

$$\inf_{\boldsymbol{M}} \mathsf{Red}_N(\boldsymbol{M}, \boldsymbol{\Phi}_{\text{data}}) \leq \mathsf{Red}_N(Q_0, \boldsymbol{\Phi}_{\text{data}}) = \int_{\boldsymbol{\Phi}_{\text{data}}} \mathcal{P}(\phi_{\text{data}}) D_{\mathsf{KL}}(P_{\phi_{\text{data}}}^N || Q_0) d\phi_{\text{data}}$$

$$= \int_{\boldsymbol{\Phi}_{\text{data}}} \mathcal{P}(\phi_{\text{data}}) \mathbb{E} \left[ \log \frac{P_{\phi_{\text{data}}}^N}{\sum_{Q \in \mathcal{Q}} Q^N} + \log |\mathcal{Q}| \right] d\phi_{\text{data}}$$

$$\leq \log |\mathcal{Q}| + \int_{\boldsymbol{\Phi}_{\text{data}}} \mathcal{P}(\phi_{\text{data}}) \mathbb{E} \left[ \log \frac{P_{\phi_{\text{data}}}^N}{\max_{Q \in \mathcal{Q}} Q^N} \right] d\phi_{\text{data}}$$

$$\leq \log |\mathcal{Q}| + N \int_{\boldsymbol{\Phi}_{\text{data}}} \mathcal{P}(\phi_{\text{data}}) \min_{Q \in \mathcal{Q}} \mathbb{E} \left[ \log \frac{P_{\phi_{\text{data}}}}{Q} \right] d\phi_{\text{data}}$$

$$= \log |\mathcal{Q}| + N \int_{\boldsymbol{\Phi}_{\text{data}}} \mathcal{P}(\phi_{\text{data}}) \min_{Q \in \mathcal{Q}} D_{\mathsf{KL}}(P_{\phi_{\text{data}}} || Q) d\phi_{\text{data}}$$

$$\leq \log |\mathcal{Q}| + N \int_{\boldsymbol{\Phi}_{\text{data}}} (\mathsf{Len}_m \log |\mathbb{K}| + 3\epsilon^2) d\phi_{\text{data}}$$

$$= \log |\mathcal{Q}| + 3N\epsilon^2 + N\mathbb{E}[\mathsf{Len}_m] \log |\mathbb{K}|$$

Plugging in the bound of $|\mathcal{Q}|$ and $\mathbb{K}$, we can see the above is upper bounded by

$$N\mathbb{E}[\mathsf{Len}_m] \log |\mathbb{K}| + 3N\epsilon^2 + 2m \log \frac{m}{\epsilon^2} + 2m d_{\text{knw}} \log \frac{L_{\text{knw}} d_{\text{knw}}}{\epsilon} + n_s d_{\text{syn}} \log \frac{L_{\text{syn}} d_{\text{syn}}}{\epsilon}.$$

Choosing $\epsilon = N^{-1}$ and applying the result in Lemma H.3, we know that

$$\inf_{\boldsymbol{M}} \mathsf{Red}_N(\boldsymbol{M}, \boldsymbol{\Phi}_{\text{data}}) \leq O(N^\alpha (\log |\mathbb{K}| + d_{\text{knw}}(\log N + \log L_{\text{knw}} + \log d_{\text{knw}})))$$

$$+ O(n_s d_{\text{syn}}(\log L_{\text{syn}} + \log d_{\text{syn}} + \log N)).$$

Therefore, we know that

$$\inf_{\boldsymbol{M}} \frac{1}{N} \mathsf{Red}_N(\boldsymbol{M}, \boldsymbol{\Phi}_{\text{data}}) = \frac{1}{N} \mathbb{I}(X_{1:N}; \phi_{\text{data}}) = \widetilde{O} \left( \frac{d_{\text{knw}}}{N^{1-\alpha}} + \frac{n_s d_{\text{syn}}}{N} \right).$$

This completes the proof. $\qquad\qquad\qquad\qquad\qquad\qquad\qquad\qquad\qquad\qquad\qquad\qquad\square$

