# OpenReview forum: "Understanding LLM Behaviors via Compression: Data Generation, Knowledge Acquisition and Scaling Laws"
_NeurIPS.cc/2025/Conference — NeurIPS 2025 spotlight_

### Official Review · Reviewer_wrDk · 2025-06-26

**Clarity:** 2
**Significance:** 3
**Originality:** 3
**Rating:** 5
**Confidence:** 3

**Summary:**

This work proposes an interpretation of LLM compression as a two-part coding process. Grounded with information theory concepts, the authors propose a syntax-knowledge model as a synthetic data generation model, and derive that the lower bound of the averaged optimal Bayesian loss follows a power-law scaling with respect to the data size. The authors also provide a theoretical analysis of model scaling laws, providing insights into LLM behaviors on factual knowledge acquisition.

**Questions:**

Please see the questions in the weakness section.

**Ethical Concerns:**

["NO or VERY MINOR ethics concerns only"]

**Final Justification:**

The authors have clarified most of my questions and concerns, including (1) the clarification of this work's main contribution (2) the choice of PYMM for modeling the factual knowledge aspects in natural language, and (3) technical sides on paper presentation. I believe this work provides a valuable theoretical foundation for understanding the dynamics of factual knowledge acquisition in LLMs, which have not been fully explored in the field yet. I am happy to increase the score by 1, recommending acceptance.

**Limitations:**

Yes

**Paper Formatting Concerns:**

There are no formatting concerns.

**Quality:**

3

**Strengths And Weaknesses:**

Strengths
- S1: The newly proposed syntax-knowledge model provides a synthetic testbed for understanding factual knowledge acquisition dynamics, which can be adopted for other studies
- S2: The framework explains several widely observed phenomena very well, and the predicted trends are verified with solid and large-scale experiments. The insights on the connection between information theory concepts and power-law scaling is impressive
- S3: The core claims are supported with mathematical rigor, and the concepts are explained in great detail (although I'm not very familiar with advanced information theory concepts, and I couldn't verify the correctness of the derivations and proofs)

Weaknesses
- W1: Although natural language is compositional, and the proposed syntax model itself doesn't inform how different syntactic structures are correlated or interact. As shown in Table 1, the generated synthetic dataset consists mainly of 'simple' statements, which can be essentially reduced to the (entity1, relation, entity2) format. Hence, the logical connection between the theoretical explanation of the scaling law and practical observations is limited to the acquisition of simple factual knowledge. Another point is that PYMM is one of many possible algorithms that can lead to a Zipfian pattern, which might significantly differ from natural language.
- W2: The central theme of 'LLMs are compressors' itself is not completely novel, although I believe the novel conceptual contributions are clear (e.g., the prediction of scaling laws). I think this can be more sharply discussed in the main text (e.g., comparison with Tishby's information bottleneck analysis on NNs and how this framework distinguishes from former analyses).
- W3: While the high-level flow is clear, I think some details should be more clarified in the main text (although some of them are resolved in the appendix): (1) $d_{knw}$ and $d_{syn}$ are used without definition in L215-217. What do they mean? (2) In Figure 3a (4a) and 3b (4b), the x- and y-axis labels and ticks are confusing. Although all axes seem to be on a log scale, it is specified in the axes label only in 3a (4a). (3) $\zeta$ is not defined in L262. (4) It is unclear exactly how the syntax model is constructed to give the actual dataset in Table 1. (5) Although the Kolmogorov structure function perspective is highlighted in the introduction section as one of the main contributions, the intuition is not discussed in the main text.
- W4: The trendline for Fig.3(a) does not seem to be a linear fit on the log-log plane (which should be linear, either the authors fitted the raw data with or without linearization), although the authors claim the experimental results follow a power-law trend.

---

> ### Author Rebuttal · Authors · 2025-07-31
>
> We sincerely thank the reviewer for the positive feedback and valuable suggestions! We are very glad to address the questions and suggestions raised by the reviewer, which we believe will help further refine our work. Below are our responses to the questions and suggestions raised by the reviewer.
>
> [**W1**] Although natural language is compositional, and the proposed syntax model itself doesn't inform how different syntactic structures are correlated or interact. As shown in Table 1, the generated synthetic dataset consists mainly of 'simple' statements, which can be essentially reduced to the (entity1, relation, entity2) format. Hence, the logical connection between the theoretical explanation of the scaling law and practical observations is limited to the acquisition of simple factual knowledge. Another point is that PYMM is one of many possible algorithms that can lead to a Zipfian pattern, which might significantly differ from natural language.
>
> [**R1**]We thank this critique regarding our modeling choices. We acknowledge that we focus exclusively on factual knowledge acquisition, and the syntax model represents some templates in our current synthetic data experiment. However, our syntax model is a general parametric framework capable of representing probabilistic context-free grammars and other syntactic structures. The whole data model is more general than the examples in the synthetic data experiment: it generates abstract knowledge elements, which a syntax model then encodes into sentences. Crucially, we only require a theoretical one-to-one mapping between each sentence and its (knowledge elements, syntax elements) pair to preserve information—without needing to specify the actual encoding process(Appendix D, Figure 7 and Example D.1). We also recognize the importance of extending this framework to handle compositional structures as a significant future direction(Lines 324-327). Our theoretical analysis intentionally employs the PYMM framework, and the direct Zipfian pattern assumption remains valid and methodologically efficient for studying data scaling. The choice of PYMM stems from PYMM's inherent "preferential attachment" mechanism for table(cluster) formation, which we believe is more representative of fundamental natural language phenomena than  Zipfian distributions. Actually, the fact that the data generated by PYMM exhibits Zipfian pattern properties is what we use to substantiate that PYMM conforms to natural language rules.
>
> [**W2**] The central theme of 'LLMs are compressors' itself is not completely novel, although I believe the novel conceptual contributions are clear (e.g., the prediction of scaling laws). I think this can be more sharply discussed in the main text (e.g., comparison with Tishby's information bottleneck analysis on NNs and how this framework distinguishes from former analyses).
>
> [**R2**] We thank the reviewer for the constructive suggestion regarding the discussion of additional related work. We acknowledge that the perspective of "LLMs as compressors" serves as our foundational framework, though our primary contributions reside in: (1) introducing a novel modeling approach for natural language data characterized by power-law frequency structures, and (2) utilizing information-theoretic tools to theoretically analyze LLM behaviors (particularly scaling laws), rather than claiming originality in the "LLMs as compressors"  argument itself. We have discussed related work on "LLMs as compressors"  in Appendix A   and provided corresponding citations. The main difference between our analysis and the former analysis is that we employ a data-centric framework. Specifically, our analysis focuses on our proposed power-law structured data framework. Consequently, it yields distinct theoretical implications compared to prior studies combining information theory with neural networks – a divergence empirically evidenced by the markedly different uniform distribution curves in Figures 3 and 4. Regarding the reviewer's reference to Tishby's information bottleneck analysis on NNs, we acknowledge that both approaches utilize similar information-theoretic tools like compression, MDL principle, etc.. We will add a detailed discussion in the camera-ready version. However, our work takes a data-centric perspective, which is novel in its focus on isolating how power-law data structures fundamentally drive scaling behaviors.
>
> [**W3**] While the high-level flow is clear, I think some details should be more clarified in the main text (although some of them are resolved in the appendix): (1)  and are used without definition in L215-217. What do they mean? (2) In Figure 3a (4a) and 3b (4b), the x- and y-axis labels and ticks are confusing. Although all axes seem to be on a log scale, it is specified in the axes label only in 3a (4a). (3)  is not defined in L262. (4) It is unclear exactly how the syntax model is constructed to give the actual dataset in Table 1. (5) Although the Kolmogorov structure function perspective is highlighted in the introduction section as one of the main contributions, the intuition is not discussed in the main text.
>
> [**R3**] We thank the reviewer's careful review of technical details. We moved certain elaborations to the appendix due to page constraints and will integrate them into the main text if space allows:
> (1) $d_{knw}$ and $d_{syn}$ were introduced in Lines 170 and 177 respectively. We will add the corresponding explanation in the theorem in the revised version.
> (2) We apologize for the inconsistent labeling. We will revise the axis labels in all relevant figures to explicitly state "(log scale)" in the camera-ready version to ensure clarity and consistency.
> (3) $\zeta$ was explicitly defined in Line 262.
> (4) In our experiments, the "knowledge model" is instantiated as a database of 400,000 individual profiles whose sampling follows a power-law distribution. The "syntax model" is then implemented as a large collection of sentence templates—50 for each type of information—that render these facts into varied linguistic expressions. For the example in Table 1, a biography template is populated with a person's profile for pre-training, while a question-answer template is used for instruction tuning. In this way, the syntax model provides the linguistic structure and variation for the factual content supplied by the knowledge model.
> (5) The full discussion of Kolmogorov structure functions appears in Appendix C due to strict page constraints. This perspective forms important high-level guidance for our theoritical framework. However, the non-computability of Kolmogorov complexity makes rigorous theoretical analysis intractable, and since explaining its abstract structure function requires significant space, we placed related insights and definition details in the appendix due to page constraints. Should space allow during revision, we will prioritize moving this content to the main text.
>
> [**W4**] The trendline for Fig.3(a) does not seem to be a linear fit on the log-log plane (which should be linear, either the authors fitted the raw data with or without linearization), although the authors claim the experimental results follow a power-law trend.
>
> [**R4**] We thank the reviewer for their careful analysis of Figure 3(a) and apologize if the visualization was misleading. The reviewer is correct that the trendlines are not perfectly linear on the log-log plot. This is because a power-law relationship of the form loss = $A  x^{-\alpha} + \epsilon$ only becomes linear on a log-log scale when plotting the loss after subtracting the irreducible error, $\epsilon$. Our figure plots the raw validation loss, and as the fitted equations in the legend show, the irreducible loss. $\epsilon$ is different for each of the power-law distributions. Because these varying baselines are not subtracted in the plot, the curves appear non-linear. However, we can confirm that the power-law model is an excellent statistical fit, with R-squared values for our regressions all exceeding 0.98. This is in stark contrast to the uniform setting, which cannot be reasonably fitted to a power-law model, highlighting the fundamental difference in scaling behavior.

---

> > ### Comment · Reviewer_wrDk · 2025-08-01
> >
> > The authors have clarified most of my questions and concerns, including (1) the clarification of this work's main contribution (2) the choice of PYMM for modeling the factual knowledge aspects in natural language, and (3) technical sides on paper presentation. I believe this work provides a valuable theoretical foundation for understanding the dynamics of factual knowledge acquisition in LLMs, which have not been fully explored in the field yet. I am happy to increase the score by 1, recommending acceptance.

---

> > > ### Author Response · Authors · 2025-08-01
> > >
> > > Thank you very much for your kind review, and we are glad that you enjoyed our paper!

---

### Official Review · Reviewer_dW7Z · 2025-07-03

**Clarity:** 3
**Significance:** 3
**Originality:** 3
**Rating:** 5
**Confidence:** 3

**Summary:**

Motivated by classical theory results, the paper presents an abstract model for language models that can be used to explain how and what models learn, why scaling laws exist, and hallucinations. They provide both theory (with various assumptions, including on how the data is sampled) and experiments to verify the theory.

**Questions:**

1. In figure 3a, I worry a little that the uniform data seems to follow a (admittedly steep) power law curve for all but the two largest data points, so could it be the case that the power law predictions for the other data will break for large enough models in a similar fashion?
2. In figure 4a, I noticed that the blue data seems to almost be subtly following a similar trend as the data for the uniform setting, have a slight plateau at the end. Is there a sense in which the uniform setting is the limit as the power goes to 1 (or is this just me reading too much into noise)? If so, could this be used to gain an understanding of the uniform setting?
3. Could this framework be applied to non-language models? Should the results extend to, say, image models?

**Ethical Concerns:**

["NO or VERY MINOR ethics concerns only"]

**Final Justification:**

The authors rebuttals are clear and address all of my questions. The paper provides more theoretical understanding of an important phenomena (like scaling laws) with experimental validation, a good contribution for deep learning theory.

**Limitations:**

The paper could spend a little more time discussing the limits of the assumptions on the data, and possibly discuss the limitations of abstracting away architecture and optimization.

**Quality:**

4

**Strengths And Weaknesses:**

The paper gives a good balance of technical explanation and discussion of the results. It has robust theory and toy experiments to support its theory. The experiments do an adequate job at supporting the theory, validating predictions while showing that when assumptions are broken those predictions fail, which is decent evidence that the theory is meaningful.

The paper is rather dense on definitions and notation, but its unclear if this is possible to substantially improve.

The results apply in a somewhat limited setting, where the data must be constructed in a fairly artificial manner, and inductive biases/details of optimization are not considered. But considering the difficulty of theory in deep learning, these weaknesses are acceptable.

---

> ### Author Rebuttal · Authors · 2025-07-31
>
> We sincerely thank the reviewer for the positive feedback and valuable suggestions! We are very glad to address the questions and suggestions raised by the reviewer, which we believe will help further refine our work. Below are our responses to the questions and suggestions raised by the reviewer.
>
> [**Q1**]In figure 3a, I worry a little that the uniform data seems to follow a (admittedly steep) power law curve for all but the two largest data points, so could it be the case that the power law predictions for the other data will break for large enough models in a similar fashion?
>
> [**A1**]  We thank the reviewer for the sharp observation and apologize for the potential misunderstanding caused by the visualization in Figure 3a. A power-law relationship of the form loss $= A x^{-\alpha} + \epsilon$ appears as a straight line on a log-log plot only after the irreducible loss $\epsilon$ is subtracted. The plot shows raw validation loss, and as the fitted equations in the legend indicate, the irreducible loss ε is different for each power-law distribution. This is why the curves are not perfectly linear. We can confirm the fits are statistically strong, with R² values exceeding 0.98 for all power-law regressions. In contrast, the data from the uniform distribution fundamentally deviates from this scaling behavior and cannot be fitted to a power law, which is the key distinction we aimed to illustrate.
>
> [**Q2**] In figure 4a, I noticed that the blue data seems to almost be subtly following a similar trend as the data for the uniform setting, have a slight plateau at the end. Is there a sense in which the uniform setting is the limit as the power goes to 1 (or is this just me reading too much into noise)? If so, could this be used to gain an understanding of the uniform setting?
>
> [**A2**] We thank the reviewer for the insightful question. The reviewer is correct to point out the subtle plateauing for the Power=1.20 data at the largest model sizes in Figure 4a. This effect, however, is an artifact of our experimental setting having a finite dataset, rather than an indication that the power-law scaling is converging to the uniform case. In this experiment, we fix the total data while increasing model size. As the model becomes sufficiently large relative to the fixed amount of knowledge in the dataset, it begins to saturate its ability to learn from that data, causing the loss reduction to plateau. In real-world, large-scale scenarios, the dataset is typically vast and scaled with the model, meaning the model rarely runs out of new information to learn from the data's long tail. Therefore, the uniform setting should be viewed as a qualitatively different regime, not as a limit of the power-law case in our framework.
>
> [**Q3**] Could this framework be applied to non-language models? Should the results extend to, say, image models?
>
> [**A3**] We thank the reviewer for the insightful question. Our theoretical framework's high-level guidance – the compression-prediction duality (Appendix B, C) – is model-agnostic. We focus on language models due to (i) empirically validated power-law structures (e.g., Zipf's and Heap's laws) in natural language, and (ii) this domain's established scaling law research foundation. If image data exhibits power-law structures resembling those in natural language, similar scaling behaviors may emerge. However, modeling these structures in visual domains currently poses significant challenges beyond our theoretical scope. This possibility remains an interesting direction for future work.
>
>
> [**L1**] The paper could spend a little more time discussing the limits of the assumptions on the data, and possibly discuss the limitations of abstracting away architecture and optimization.
>
> [**R1**] We thank the reviewer for the constructive suggestion regarding the discussion of limitations. As acknowledged in ​​the” Concluding Remarks“ section, we have stated the limitations of our data model and outlined potential extensions like "compositional reasoning and inference mechanisms" (Lines 324-327).  ​​Our work adopts a data-centric perspective, using information-theoretic tools to characterize the power-law properties of data distributions. Within this framework, we analyze the redundancy bound of the Bayesian optimal predictor—abstracting away model architectures and optimization algorithms to isolate how data structure fundamentally drives scaling behaviors. We will add a discussion of this data-centric perspective in the camera-ready version to further clarify its implications.

---

> > ### Comment · Reviewer_dW7Z · 2025-08-03
> >
> > The authors have effectively answered my questions, and I have increased my confidence in my score of accept. I enjoyed the paper, and I wish the authors good luck!

---

> > > ### Author Response · Authors · 2025-08-04
> > >
> > > Thank you very much for your kind review, and we are glad that you enjoyed our paper!

---

### Official Review · Reviewer_VRwY · 2025-07-03

**Clarity:** 3
**Significance:** 3
**Originality:** 3
**Rating:** 5
**Confidence:** 4

**Summary:**

This paper provides a theoretical framework for understanding LLM behaviors through the lens of compression and Kolmogorov complexity. The authors interpret LLM training as a two-part process using the Kolmogorov Structure Function, where models first learn frequent syntactic patterns then gradually acquire rare knowledge. They introduce the Syntax-Knowledge model, a hierarchical data generation framework that separates the language syntax (ie. rules, parametric model) from factual knowledge (non-parametric Pitman-Yor Chinese Restaurant Process). Under Bayesian assumptions, the paper derives both data and model scaling laws: data scaling follows O(N^(α-1)) for knowledge and O(N^(-1)) for syntax, while model scaling shows that larger models progressively learn lower-frequency knowledge. The theory predicts that models will hallucinate knowledge below frequency thresholds due to capacity constraints. Experiments on power-law distributed synthetic datasets validate these predictions, and the authors make predictions on adequate data distributions.

**Questions:**

Q1: Is it possible to literally show the scaling law plots (loss vs flops(compute) to see if the scaling laws actually work?
Q2: The paper doesn't rely much on the transformer architecture. *imaging* llms were just dense mlps, when would these findings hold? Does the finding assume that the architecture is "good enough" to "implement" this two part coding process?

**Ethical Concerns:**

["NO or VERY MINOR ethics concerns only"]

**Final Justification:**

The authors rebuttals were satisfactory!

I will maintain my positive score of 5.

**Limitations:**

Yes.

**Paper Formatting Concerns:**

None.

**Quality:**

4

**Strengths And Weaknesses:**

S1: This paper is very well written. Despite the theoretical complexity of the data and their model, everything is very clearly described and easy to follow.


S2: This paper demonstrates a very clean model explaining neural scaling laws established in language models. There have been past efforts to model neural scaling laws with synthetic data, but most of them didn’t fit the single epoch online nature, or didn’t explicitly include complexities from syntax and knowledge. This paper beautifully integrates both parts and makes clean predictions.


S3: Nice juxtaposition of theory and experiments. The paper cleanly displays theoretical findings and experimental validation.


S4: The paper makes a prediction based on their analysis that “it can be advantageous to have power law distributed data” for a well justified reason.

---

W1: While neural scaling laws are discussed, it would be nice to literally show the scaling law plots of compute vs loss predicted by theory and ran by experiments. This, for me, is the main asset missing.


W2: The data generation process can be better illustrated visually to help understanding.


W3: It would be good if the paper could be further grounded to the real LLM world, explicitly mention what kind of data distributional choices can be made: e.g. English vs other languages in pretraining, etc.

---

> ### Author Rebuttal · Authors · 2025-07-31
>
> We sincerely thank the reviewer for the positive feedback and valuable suggestions! We are very glad to address the questions and suggestions raised by the reviewer, which we believe will help further refine our work. Below are our responses to the questions and suggestions raised by the reviewer.
>
> [**W1**]While neural scaling laws are discussed, it would be nice to literally show the scaling law plots of compute vs loss predicted by theory and ran by experiments. This, for me, is the main asset missing.
>
> [**R1**]We thank the reviewer's suggestion regarding compute-optimal scaling plots. First, we would like to recall that the scaling law takes the form $L=L_0+AN^{-\alpha}+BM^{-\beta}$. Because our theoretical analysis is asymptotic, the multiplicative constants A and B cannot be precisely quantified. While we can derive approximate compute-optimal configurations of data size N and model size M in the asymptotic regime, ignoring the practical significance of constants A and B would risk substantial prediction errors. We therefore find empirical characterization more meaningful for studying compute-optimal scaling laws. Moreover, we include the experimental part of this question in our answer to Q1.
>
> [**W2**] The data generation process can be better illustrated visually to help understanding.
>
> [**R2**] We thank the reviewer for the valuable suggestion. We would like to highlight that Figure 7 in Appendix D provides a schematic visualization of the data generation process, supplemented by concrete implementation examples (Example D.1). Space constraints required moving the schematic visualization to the appendix, and we will integrate it into the main text if revision space becomes available and refine the visualization to include more details.
>
> [**W3**]It would be good if the paper could be further grounded to the real LLM world, explicitly mention what kind of data distributional choices can be made: e.g. English vs other languages in pretraining, etc.
>
> [**R3**]We thank the reviewer for this valuable suggestion, as our framework offers direct insights into data curation. Our findings imply two strategies for optimizing pre-training: First, for fixed model capacity, data with frequency below a model-size-dependent threshold is fundamentally unlearnable(Figure 2). Our theoretical analysis proves such patterns cannot be learned regardless of training duration(Theorem 4.6), making this computationally wasteful. Second, for high-frequency data sources, strategic downsampling can improve computational efficiency, as high-frequency patterns are learned quickly and provide diminishing returns, freeing up FLOPs for a broader range of knowledge.
>
> [**Q1**] Is it possible to literally show the scaling law plots (loss vs flops(compute) to see if the scaling laws actually work?
>
> [**A1**] We thank the reviewer for this suggestion. While the current rebuttal rules preclude adding new figures, we will gladly include a direct loss versus compute (FLOPs) plot in the camera-ready version to make this relationship explicit. For now, we wish to clarify that the necessary components of the scaling law are already presented. Figures 3 (loss vs. tokens trained) and 4 (loss vs. model parameters) are generated from the same set of model training runs. Since the number of tokens trained is directly proportional to computational flops, these two figures jointly confirm that our experimental results align with established data and model scaling laws.
>
> [**Q2**] The paper doesn't rely much on the transformer architecture. imaging llms were just dense mlps, when would these findings hold? Does the finding assume that the architecture is "good enough" to "implement" this two part coding process?
>
> [**A2**] We thank the reviewer for the thoughtful question about architecture-agnostic implications in our methodology. Our theoretical analysis operates under the optimal predictor framework,  abstracting away from implementational constraints like specific architectures (Transformer/MLP) or optimizers (Adam/SGD). Analyzing training dynamics under fully realistic settings(incorporating data distributions, model architectures, and optimizers) is intractable. We posit that data's power-law structures fundamentally drive scaling laws' behaviors, as empirically substantiated in Figures 3 and 4. Consequently, we employ a data-centric theoretical framework, analyzing the redundancy bound of the optimal predictor. We hope that future work can systematically incorporate these implementation-level considerations.

---

> > ### Comment · Reviewer_VRwY · 2025-08-05
> >
> > R1: I see. Thank you!
> >
> > R2: Acknowledged.
> >
> > R3: Thank you
> >
> > thank you for the questions answered as well.
> >
> > I think I will maintain my positive score!

---

> > > ### Author Response · Authors · 2025-08-06
> > >
> > > Thank you very much for your kind review, and we are glad that you enjoyed our paper!

---

### Official Review · Reviewer_Da6R · 2025-07-05

**Clarity:** 3
**Significance:** 3
**Originality:** 2
**Rating:** 4
**Confidence:** 1

**Summary:**

This paper presents a theoretical framework to understand the behavior of Large Language Models (LLMs) under computational constraints. It introduces a Bayesian model that separates syntax and factual knowledge learning, explaining how LLMs scale with data size and model capacity. The paper also derives scaling laws for knowledge acquisition, redundancy reduction, and loss decomposition, supported by empirical experiments on synthetic and real-world datasets.

**Questions:**

1. How realistic are the assumptions of conditional independence between knowledge clusters given the model? Are there empirical studies or experiments supporting this?
2. While the paper includes experiments on real-world data (e.g., PopQA), how well do the theoretical predictions hold when applied to more diverse or complex tasks beyond fact-based QA?
3. Given the focus on large-scale models and data, what are the computational implications of the proposed framework? Can it be efficiently implemented or approximated in practice?

**Ethical Concerns:**

["NO or VERY MINOR ethics concerns only"]

**Limitations:**

yes

**Paper Formatting Concerns:**

No formatting issues

**Quality:**

3

**Strengths And Weaknesses:**

Strength：
1. The paper offers a novel Bayesian perspective on LLM learning dynamics.
2. The theoretical results are backed by extensive experiments on both synthetic and real-world datasets (e.g., PopQA).
3. The paper is well written and organized.

Weakness:
1. While theoretically insightful, the paper lacks concrete guidance on how these findings can be applied to improve practical LLM training or deployment strategies.

---

> ### Author Rebuttal · Authors · 2025-07-31
>
> We sincerely thank the reviewer for the positive feedback and valuable suggestions! We are very glad to address the questions and suggestions raised by the reviewer, which we believe will help further refine our work. Below are our responses to the questions and suggestions raised by the reviewer.
>
> [**W1**] While theoretically insightful, the paper lacks concrete guidance on how these findings can be applied to improve practical LLM training or deployment strategies.
>
> [**R1**] We thank the reviewer for the constructive feedback regarding the practical applicability of our theoretical insights. Our theoretical framework provides a more nuanced characterization of scaling behaviors than existing scaling laws (e.g., Chinchilla). Specifically, we establish frequency-specific loss predictions across model scales and reveal size-dependent learning bottlenecks related to knowledge frequency(Figure 1 and Figure 2). These theoretical contributions can yield implications for model size-aware data curation strategies. More specifically, our findings imply two strategies for optimizing pre-training data curation: First, for fixed model capacity, data with frequency below a model-size-dependent threshold is fundamentally unlearnable. Our theoretical analysis proves such patterns cannot be learned regardless of training duration, making this computationally wasteful. Second, for high-frequency data sources, strategic downsampling can improve computational efficiency, as high-frequency patterns are learned quickly and provide diminishing returns, freeing up FLOPs for a broader range of knowledge. However, consistent with our paper's primary focus on fundamental insights, we defer operational aspects of training improvement to future work.
>
> [**Q1**] How realistic are the assumptions of conditional independence between knowledge clusters given the model? Are there empirical studies or experiments supporting this?
>
> [**A1**] We thank the reviewer for the rigorous examination of our data model's conditional independence assumption. We would like to clarify that the conditional independence assumption pertains specifically to our knowledge model, which generates factual knowledge (such as personal information like birthdates, birthplaces, etc. in our synthetic data experiments). We maintain that this assumption is well-justified for factual knowledge representations. As acknowledged in the ”Concluding Remarks“ section, we also have stated the limitations of factual knowledge representations and outlined potential extensions like "compositional reasoning and inference mechanisms" (Lines 324-327).
>
> [**Q2**] While the paper includes experiments on real-world data (e.g., PopQA), how well do the theoretical predictions hold when applied to more diverse or complex tasks beyond fact-based QA?
>
> [**A2**] We thank the reviewer for the excellent question regarding the generalizability of our theory. We hypothesize that our core findings would extend to other knowledge-intensive tasks, such as those in benchmarks like MMLU, where the acquisition of concepts can also be modeled as a process of compressing information with power-law frequency distributions. However, for tasks that rely heavily on complex, multi-step reasoning or mathematical logic, our current model would likely be insufficient. These domains involve procedural and compositional knowledge that cannot be captured by our current assumptions. As we state in our conclusion, integrating more complex structures like "compositional reasoning and inference mechanisms"  into our theoretical model is a crucial and exciting direction for future work.
>
> [**Q3**] Given the focus on large-scale models and data, what are the computational implications of the proposed framework? Can it be efficiently implemented or approximated in practice?
>
> [**A3**] We thank the reviewer for the inquiry regarding computational implications. To clarify, our work does not propose a new training framework. Rather, our work adopts a data-centric perspective, using information-theoretic tools  to characterize the power-law properties of data distributions. Within this framework, we analyze the redundancy bound of the Bayesian optimal predictor—abstracting away model architectures and optimization algorithms to isolate how data structure fundamentally drives scaling behaviors. Our insights into computational implications are also based on data curation, which we have already presented in R1.

---

> > ### Author Response · Authors · 2025-08-09
> >
> > Dear Reviewer Da6R,
> >
> > I hope this message finds you well. As the discussion phase is approaching its end, we would like to kindly confirm whether we have sufficiently addressed all of your concerns. Should there be any remaining questions requiring further clarification, please do not hesitate to let us know. Your insights are in invaluable to us, and we are eager to address any remaining issues to improve our work.
> >
> > Thank you again for your time and effort in reviewing our paper. We sincerely look forward to your feedback.

---

### Note · Authors · 2025-08-15

We thank the reviewers for their constructive feedback and valuable discussion. To clarify our main contribution, we emphasize that our work introduces a new data-centric theoretical framework explicitly grounded in the power-law frequency structure of natural language. This data-centric perspective, leveraging information-theoretic tools, allows us to derive novel theoretical predictions—most notably, the emergence of scaling laws from data properties alone. This finding is also empirically validated by our experiments. Beyond its theoretical contributions, our framework offers direct, actionable guidance for data curation. It suggests two key strategies for optimizing pre-training: 1) Data Filtering, where removing data below a model-size-dependent frequency threshold prevents computational waste on patterns the model cannot learn, as proven by our analysis; and 2) Data Downsampling, which strategically reduces rapidly learned, high-frequency data to free up resources for a broader spectrum of knowledge. We believe this work offers a valuable new lens on how data shapes LLM capabilities and provides a principled foundation for future data-centric AI.

---

### Decision · Program_Chairs · 2025-09-17

**Decision:**

Accept (spotlight)

**Comment:**

This paper presents a theoretical framework for understanding LLM behaviors through the lens of compression and Kolmogorov complexity. To study this formally, the authors introduce a Syntax-Knowledge model, a hierarchical data-generation framework that separates language syntax (i.e., rules, parametric model) from factual knowledge (non-parametric Pitman–Yor Chinese Restaurant Process), allowing the derivation of both data and model scaling laws. The theory further predicts that models will hallucinate knowledge below frequency thresholds due to capacity constraints.

Overall, the paper is well written and all reviewers recommend acceptance. The introduction of data modeling enables a formal characterization of power-law scaling, providing new theoretical insights into the view of “LLMs as compressors.” While questions were raised regarding the contribution relative to prior work on this theme, the authors have addressed them during the rebuttal.